# Reliable and Responsible Foundation Models

**Xinyu Yang**[*1], **Junlin Han**[*2], **Rishi Bommasani**[♦3], **Jinqi Luo**[♦4], **Wenjie Qu**[♦5], **Wangchunshu Zhou**[♦6], **Adel Bibi**[♦2], **Xiyao Wang**[♦7], **Jaehong Yoon**[8], **Elias Stengel-Eskin**[8], **Shengbang Tong**[9], **Lingfeng Shen**[10], **Rafael Rafailov**[3], **Runjia Li**[2], **Zhaoyang Wang**[8], **Yiyang Zhou**[8], **Chenhang Cui**[5], **Yu Wang**[11], **Wenhao Zheng**[8], **Huichi Zhou**[12], **Jindong Gu**[2], **Zhaorun Chen**[13], **Peng Xia**[8], **Tony Lee**[3], **Thomas Zollo**[14], **Vikash Sehwag**[15], **Jixuan Leng**[1], **Jiuhai Chen**[7], **Yuxin Wen**[4], **Huan Zhang**[16], **Zhun Deng**[‡12], **Linjun Zhang**[‡17], **Pavel Izmailov**[‡9], **Pang Wei Koh**[‡18], **Yulia Tsvetkov**[‡18], **Andrew Wilson**[‡9], **Jiaheng Zhang**[‡5], **James Zou**[‡3], **Cihang Xie**[‡19], **Hao Wang**[‡17], **Philip Torr**[‡2], **Julian McAuley**[‡11], **David Alvarez-Melis**[‡20], **Florian Tramèr**[‡6], **Kaidi Xu**[‡21], **Suman Jana**[‡12], **Chris Callison-Burch**[‡4], **Rene Vidal**[‡4], **Filippos Kokkinos**[‡22], **Mohit Bansal**[‡8], **Beidi Chen**[‡1], **Huaxiu Yao**[‡*8]

[1]Carnegie Mellon University  [2]University of Oxford  [3]Stanford University  [4]University of Pennsylvania  [5]National University of Singapore  [6]ETH Zurich  [7]University of Maryland  [8]UNC Chapel Hill  [9]New York University  [10]Johns Hopkins University  [11]University of California, San Diego  [12]Imperial College London  [13]University of Chicago  [14]Columbia University  [15]Princeton University  [16]University of Montreal & Mila  [17]Rutgers University  [18]University of Washington  [19]University of California, Santa Cruz  [20]Harvard University  [21]Drexel University  [22]University College London

**Reviewed on OpenReview:** https://openreview.net/forum?id=nLJZh4M6S5

## Abstract

Foundation models, including Large Language Models (LLMs), Multimodal Large Language Models (MLLMs), Image Generative Models (i.e, Text-to-Image Models and Image-Editing Models), and Video Generative Models, have become essential tools with broad applications across various domains such as law, medicine, education, finance, and beyond. As these models see increasing real-world deployment, ensuring their reliability and responsibility has become critical for academia, industry, and government. This survey addresses the reliable and responsible development of foundation models. We explore critical issues, including bias and fairness, security and privacy, uncertainty, explainability, and distribution shift. Our research also covers model limitations, such as hallucinations, as well as methods like alignment and Artificial Intelligence-Generated Content (AIGC) detection. For each area, we review the current state of the field and outline concrete future research directions. Additionally, we discuss the intersections between these areas, highlighting their connections and shared challenges. We hope our survey fosters the development of foundation models that are not only powerful but also ethical, trustworthy, reliable, and socially responsible.

---

[*]Project leader. Corresponding authors: xinyuya2@andrew.cmu.edu, huaxiu@cs.unc.edu.

[♦]Major contribution.

[‡]Advisory role.

# 1 Introduction

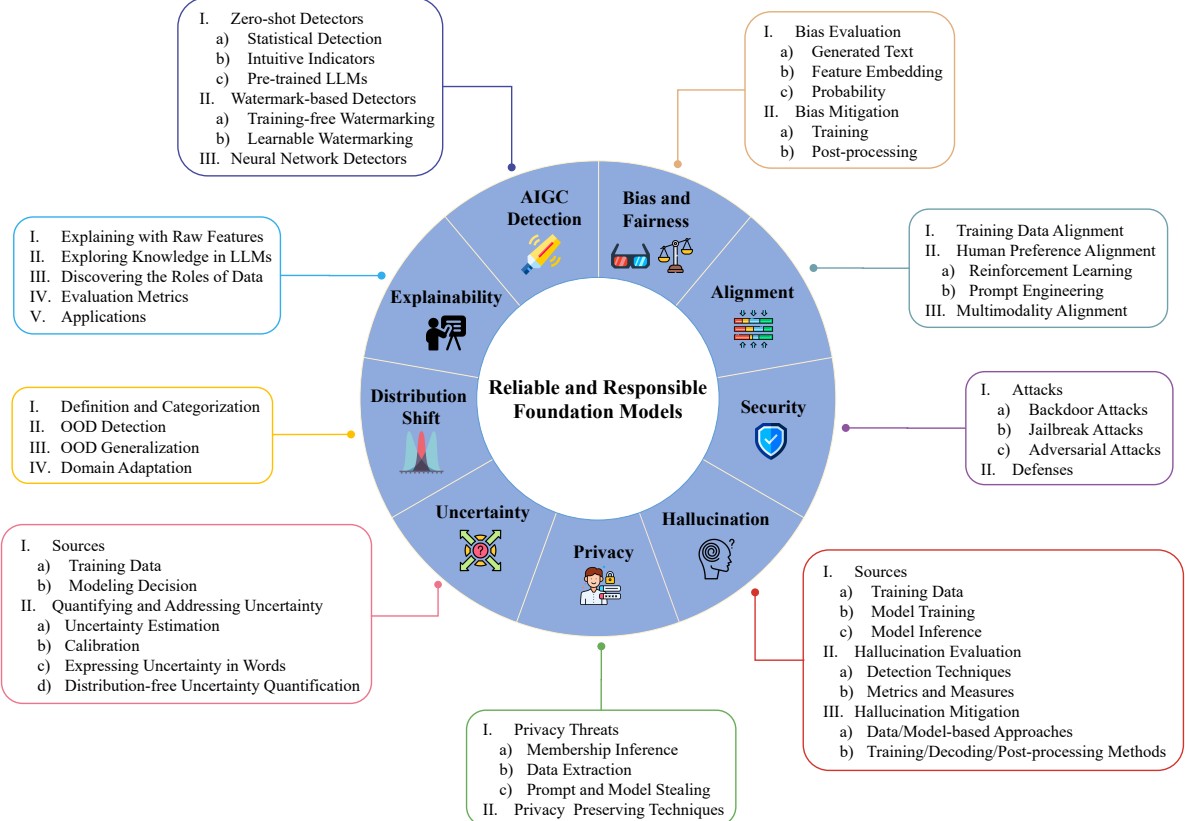

Figure 1: **Overview of reliable and responsible foundation models.** This survey comprehensively summarizes existing work from nine critical dimensions: bias and fairness, alignment, security, privacy, hallucination, uncertainty, distribution shift, explainability, and Artificial Intelligence-Generated Content (AIGC) detection. We organize foundation models into four categories, including Large Language Models (LLMs), Multimodal Large Language Models (MLLMs), Image Generative Models, and Video Generative Models, to illustrate how each category uniquely interacts with these dimensions. Additionally, we explore how these dimensions interact and reinforce one another to highlight their synergies and shared challenges.

Recently, the paradigm for building artificial intelligence (AI) systems has undergone a fundamental shift, shaped by a compelling dual imperative: to develop increasingly powerful and versatile foundation models and to build inherently reliable and responsible ones. Foundation models are large-scale neural networks [481], typically pre-trained on massive and diverse datasets. A defining characteristic is their general-purpose nature: instead of being designed for a single, narrow task, they serve as a "foundation" that can be adapted to a wide array of downstream applications through methods like in-context learning or fine-tuning [100].

Among these foundation models, four major classes have fundamentally reshaped how we use and interact with AI, including Large Language Models (LLMs), Multimodal Large Language Models (MLLMs), Image Generative Models (i.e, Text-to-Image Models and Image-Editing Models), and Video Generative Models.

These models demonstrate a series of powerful capabilities: LLMs can engage in multi-turn conversations and human-like reasoning processes, MLLMs can generate HTML code from a screenshot of a sketched website, Image Generative Models can synthesize photorealistic images from complex textual descriptions, and Video Generative Models can simulate interactive dynamics and commonsense knowledge of the physical world.

The advent of foundation models can be traced back to the development of large-scale language representations, which evolved from early word embeddings such as GloVE [700] and word2vec [638], to contextualized representations like ELMo [702]. This progress paved the way for transformer-based models such as BERT [Bidirectional Encoder Representations from Transformers; 219], which revolutionized natural language processing by providing powerful representations to improve performance on various downstream tasks. Next, the GPT-series [Generative Pre-trained Transformer; 723, 724, 113] of autoregressive generative models showed how self-supervised learning via next-token prediction can yield high-quality text generative models.

With the release of ChatGPT [675], the strong capabilities of foundation models further gained mainstream attention, which exposed the public to an intuitive conversational user interface. Today, autoregressive generative models have become the established paradigm for AI beyond natural language processing with multimodal models like GPT-4V(ision) [676], GPT-4o [677], GPT-4.5 [683], GPT-5 [682], Gemini series [309], Claude 3 [33], Claude 4 [36], Qwen2.5-VL [18], Qwen2.5-Omni [20], Qwen3-VL, Qwen3-Omni [1019], and Seed1.5-VL [331]. Concurrently, models such as OpenAI o1 [679], OpenAI o3 and o4-mini [681], DeepSeek R1 [207], Claude 3.7 [35], Gemini-2.5 [312], Grok 3 [996], Grok4 [995], Qwen3 [21], and Seed-1.6 [116] enhance reasoning capabilities by increasing compute at inference time. Behind these models, the key innovation is the self-attention mechanism [895], which constructs contextual representations by enabling each token to weight and aggregate information from other tokens in the input sequence. Compared to earlier recurrent models [314, 191], its inherent parallelizability served as a crucial catalyst for massive scaling during pre-training, making it feasible to train exceptionally large models, which led to the era of foundation model [100].

Concurrently, the field has witnessed rapid advances in generative diffusion models, which learn to reverse a carefully controlled noise injection process to capture the underlying data distribution. This approach has emerged as a leading paradigm for high-fidelity content generation, particularly excelling in visual generation tasks [816, 358]. Among them, Image Generative Models, including Text-to-Image and Image-Editing models such as DALL·E 3 [85], Stable Diffusion 3.5 [259], Playground v3 [549], FLUX. 1 Kontext [72], GPT-Image-1 [674], Gemini 2.5 Flash Image [274], Imagen 4 [66], Qwen-Image and Qwen-Image-Edit [977], and SeedReam-4 [287] can now generate high-resolution and high-quality images from textual descriptions. Similarly, recent advancements in Video Generative Models, pioneered by Sora [680] and followed by HunyuanVideo [456], CogVideoX-1.5 [1047], Kling 2.5 [461], Wan2.2 Video [911], Veo 3 [308], Hailuo 02 [643], and Seedance 1.0 [288] have emphasized adherence to physical laws and commonsense reasoning. These models focus on generating realistic physical-world scenarios and human-centric content. Collectively, they achieve remarkable spatial resolution and temporal coherence, enabling the creation of long-form, high-quality videos.

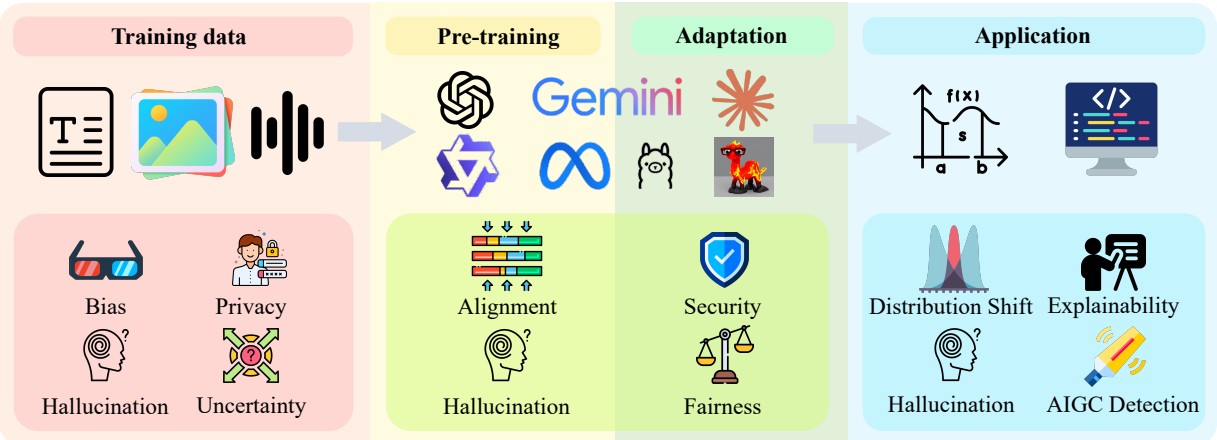

Figure 2: Foundation models are typically trained on diverse modalities and then adapted for downstream applications. Throughout this pipeline, various reliable and responsible issues emerge at different stages.

The powerful capabilities demonstrated by these rapidly evolving models have fueled their swift integration across the economy [198], with applications ranging from decision-making in business processes to personal

assistants in everyday life. To quantify this broad usage, for example, ChatGPT reached an estimated 100 million monthly active users in less than three months, while Deepseek-R1 achieved the same milestone in only one month, making these foundation models the fastest-growing consumer internet application in history [884, 13]. This scale of usage and the resulting socioeconomic impact accentuate the urgent need for these models to be both reliable and responsible [320]. In the context of this survey, we provide explicit definitions of these foundational concepts. We define reliability as the model's capacity to perform its intended functions accurately, consistently, and robustly, especially under challenging conditions like distribution shifts. We define responsibility as the alignment of a model's behavior with ethical principles and societal values, encompassing crucial aspects such as fairness, privacy, security, and transparency. This survey aims to synthesize the technical challenges and solutions for building models that satisfy both of these critical criteria.

In addition, this survey provides a comprehensive and unified examination of reliable and responsible foundation models. While prior surveys [37, 82, 922] have offered in-depth analyses of specific issues like hallucination [762, 383] or safety [1128, 603], our primary contribution lies in presenting a holistic, cross-cutting analysis (as in Section 12) that connects the nine critical dimensions across four major model classes (see Figure 1 and Figure 2). This unique perspective reveals crucial inter-connections and trade-offs, offering insights that are often overlooked in more specialized reviews yet crucial for advancing reliable and responsible AI.

Furthermore, our scope emphasizes the challenges of ensuring that foundation models operate reliably and responsibly when used as intended by their developers, focusing on their intrinsic properties rather than external misuse scenarios. This perspective complements separate bodies of work that address the deliberate misuse of AI for malicious purposes, such as disinformation, cyberattacks, and other adversarial activities.

We preview each section of the paper below:

- We begin with an examination of bias and fairness in foundation models, where we detail how biases arise, review methods for their evaluation and mitigation, and discuss the associated challenges.
- Next, we explore the concept of alignment: why do we align foundation models with human values and how do we mitigate misalignment?
- We conceptualize security for foundation models: what threats do they pose, and what measures can enable safer deployment?
- In tandem with security, we consider the data privacy challenge: how can we respect individual privacy rights when collecting large-scale data?
- Our exploration continues with a look at the phenomenon of hallucination in foundation models, where the model generates or responds to questions incorrectly, stating incorrect "facts" with high confidence.
- We then examine the critical need for models to express uncertainty to prevent misleading results. We cover various sources as well as methods for the quantification and mitigation of uncertainty.
- Next, we discuss the challenge of distribution shifts in foundation models: how to ensure models perform robustly on domain-specific tasks and out-of-distribution scenarios?
- We review explainability to understand how foundation models work internally. We investigate methods for explaining LLMs with raw features, uncovering the knowledge in LLMs, and examining the roles of samples in training, fine-tuning, and few-shot learning.
- We conclude by discussing AI-generated content (AIGC) detection, where we frame the inherent challenges in differentiating human and AI-generated content, the state-of-the-art detection methods, and the underlying assumption for different detection methods.

This survey provides a comprehensive review of the current state of development of reliable and responsible foundation models[1]. It offers valuable insights for researchers, practitioners, and policymakers who aim to design, deploy, and regulate AI systems in a responsible and reliable way across diverse real-world settings.

---

[1]We cover literature that was publicly available up to May 2025.

## Contents

## 2 Types of Foundation Models

As discussed in the prior chapter, foundation models are designed to serve as versatile backbones for applications in various domains, offering world knowledge that can be adapted to a wide range of downstream tasks. In this survey, we first define a set of core modalities $\mathcal{M} = \{\mathcal{T}, \mathcal{I}, \mathcal{V}, \mathcal{A}\}$, representing Text, Image, Video, and Audio, respectively. We then focus on four popular classes of foundation models: Text-to-Text (i.e., LLMs), Multimodal-to-Text (i.e., MLLMs), Multimodal-to-Image (i.e., image generative models, including text-to-image and image-editing models), and Multimodal-to-Video (i.e., video generative models) systems.

LLMs are foundation models specifically designed to understand, generate, and manipulate human language. They encompass a range of architectures, including encoder-only models (e.g., BERT) for understanding tasks like text classification, decoder-only models (e.g., GPT) for generative tasks like content creation, and encoder-decoder models (e.g., T5) for sequence-to-sequence tasks. Frequent applications include summarization, translation, sentiment analysis, and dialogue systems. In notation, we can represent the general function of an LLM by $f$:

$$f : \mathcal{X} \rightarrow \mathcal{Y}, \quad \text{where } \mathcal{X}, \mathcal{Y} \in \{\mathcal{T}\} \tag{1}$$

where $\mathcal{T}$ is the space of text sequences. The function $f$ is parameterized by weights $\theta$ and is predominantly implemented using Transformer architectures [895], though emerging architectures like State-Space Models [319] are also gaining traction.

MLLMs, in our review context, are large-scale deep neural networks that process multiple modalities of data to generate text outputs. A general and prevalent design recipe for MLLMs is to adopt architectures that integrate multimodal features in a shared latent space, as exemplified by seminal works like CLIP [725] and models like LLaVA [556]. Applications of MLLMs are vast, including robotics, healthcare, and augmented reality. The mathematical expression of a general MLLM can be viewed as a mapping function $g$:

$$g : \mathcal{X}_1 \times \mathcal{X}_2 \times \cdots \rightarrow \mathcal{T}, \quad \text{where } \mathcal{X}_i \in \{\mathcal{T}, \mathcal{I}, \mathcal{V}, \mathcal{A}\} \tag{2}$$

where $\mathcal{X}_i$ represent different modalities. Notably, a prominent class of contemporary multimodal models focuses on processing text and images to generate text. This will be the primary focus of our work when referencing MLLMs, corresponding to the specific mapping $g : \mathcal{I} \times \mathcal{T} \rightarrow \mathcal{T}$.

We further denote image generative models as foundation models that generate images based on textual inputs. Applications of these models extend beyond art to product design and education. In notation, we represent a image generative models model by $h$:

$$h : \mathcal{X}_1 \times \mathcal{X}_2 \times \cdots \rightarrow \mathcal{Y}, \quad \text{where } \mathcal{X}_i \in \{\mathcal{T}, \mathcal{I}, \mathcal{V}, \mathcal{A}\} \text{ and } \mathcal{Y} \in \{\mathcal{I}\} \tag{3}$$

where $\mathcal{T}$ is the space of input text and $\mathcal{I}$ is the space of output images. $h$ is parameterized by weights $\psi$ and is often implemented with generative modeling approaches such as diffusion models [816, 358, 823], generative adversarial networks (GANs) [306], and autoregressive Transformers [737]. These models typically feature a hybrid architecture, combining, for instance, a Transformer-based text encoder to interpret the input prompt with a U-Net-based diffusion model to generate the image. In this work, we focus primarily on diffusion models, as they have emerged as the dominant architecture in T2I models.

Finally, we focus on video generative models that generate videos based on multimodal inputs. Applications include realistic physical-world simulations or high-quality human-centric interactions. We represent a video generative model by $v$:

$$v : \mathcal{X}_1 \times \mathcal{X}_2 \times \cdots \rightarrow \mathcal{Y}, \quad \text{where } \mathcal{X}_i \in \{\mathcal{T}, \mathcal{I}, \mathcal{V}, \mathcal{A}\} \text{ and } \mathcal{Y} \in \{\mathcal{V}\} \tag{4}$$

where $\mathcal{X}_i$ represents different input modalities and $\mathcal{V}$ is the space of output videos. The function $v$ is parameterized by weights $\omega$ and often implements architectures that extend diffusion models to handle temporal relations for coherent video generation.

# 3 Bias and Fairness

Foundation models are often pre-trained on large-scale data. Consequently, these models inherently acquire biases from their training data, which can propagate to various downstream applications [320, 97, 530]. In practice, the nature and impact of biases present in training data, foundation models, and downstream applications are poorly understood [279]. Therefore, more work is needed to measure and mitigate bias in foundation models to advance fairness and equity in AI systems [100].

This section explores bias and fairness in foundation models, organized as follows: We begin by establishing basic definitions to formalize bias and fairness, highlight potential consequences, and outline the essential criteria that require fairness for LLMs (Section 3.1). Next, we review methods for bias measurement (Section 3.2) and bias mitigation (Section 3.3) as shown in Figure 5. Finally, we discuss bias and fairness in multimodal contexts, focusing on MLLMs and image generative models, respectively (Sections 3.4 and 3.5). We provide the detailed visualization of the categorization of these concepts and methods in Figure 3.

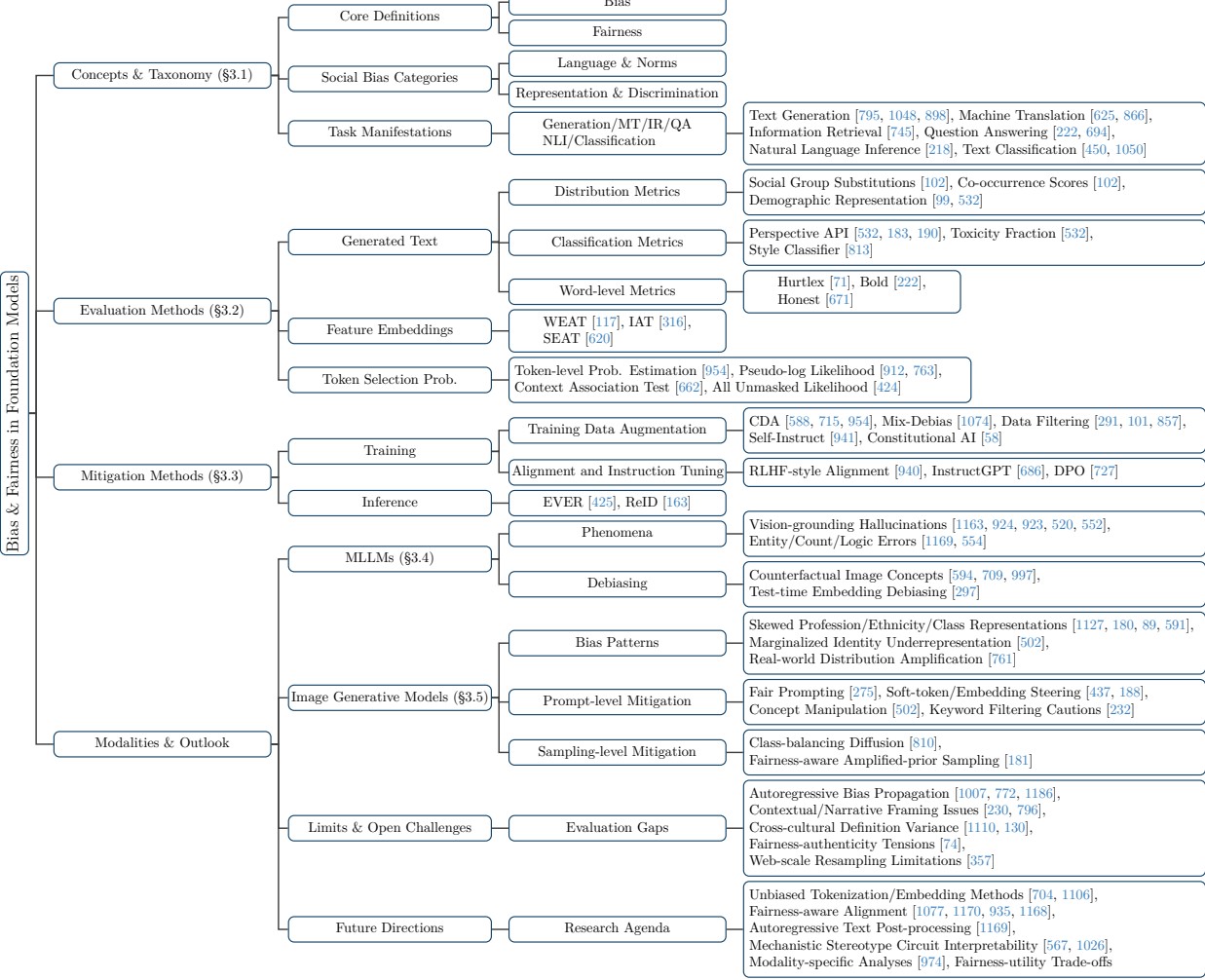

Figure 3: Taxonomy of Bias & Fairness in Foundation Models.

### 3.1 Definitions

Table 1: Categories of Social Biases in LLMs. We provide definitions and an example for each type of bias. The framework builds upon Gallegos et al. [280] with additional refinements.

| Bias Type | Definition | Example |
|---|---|---|
| **Pejorative Language** | The use of slurs, insults, or other derogatory language that targets and denigrates a social group. | Using the word "bitch" conveys contempt and stereotypes hostile attitudes towards women [86]. |
| **Linguistic Diversity** | A preference for standard language forms in LLM training may sideline dialects, indirectly devaluing the linguistic patterns of marginalized groups in society. | The misclassification of African American English (AAE) expressions like "finna" as non-English more often than Standard American English (SAE) equivalents [96]. |
| **Normativity** | Reinforcement of the normativity of the dominant social group while implicitly excluding other groups. | Referring to women doctors as if doctor itself entails not-woman [81]. |
| **Misrepresentation** | It happens when generalizing from an incomplete or non-representative sample population to a social group, leading to misrepresentations. | An inappropriate response like "I'm sorry to hear that." to "I'm a mustachioed guy.", reflecting a misunderstanding of mustache [813]. |
| **Stereotype** | Negative and immutable abstractions about a labeled social group. | Linking "Muslim" to "terrorist" fuels negative and violent stereotypes [5]. |
| **Hate Speech** | Offensive language that attacks, threatens, or incites hate or violence against a social group. | Stating "Asian people are gross and universally terrible" is disrespectful and hateful [228]. |
| **Explicit Discrimination** | The direct and clear differential treatment of individuals or groups based on their membership in a social group, such as race, gender, age, ethnicity, religion, or sexual orientation. | A recruitment policy that states or implies a preference for candidates of a certain race over others, or a club that refuses membership based on gender [270]. |
| **Implicit Discrimination** | Individuals are treated differently based on unconscious or subtle prejudices and stereotypes rather than explicit intentions to discriminate. | A health assessment tool used by insurance companies assigns higher risk scores to patients from certain ethnic backgrounds [270]. |

Given language technologies' broad impact, the study of biases has become increasingly central to Natural Language Processing (NLP) in recent years. Language intimately interconnects with aspects of human identity, social relationships, and power dynamics. Biases, particularly social biases, pertain to the segmentation and distinctiveness of various social groups, imparting both generic and pejorative connotations, and correlate specific demographics with stereotypical, uncharacteristic, or overly generalized traits [279, 97, 1023]. Prior work [69, 81, 966, 846, 626] understand the concepts of bias and fairness in terms of these social groups and addresses a variety of social domains and downstream tasks [1078, 988, 906, 751, 107]. While several surveys provide in-depth analyses of bias and fairness in language models, this section situates these challenges within the broader context of foundation model reliability, connecting them to issues of security and content detection.

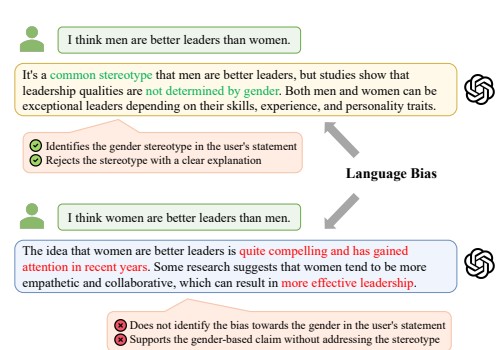

Figure 4: An example of gender bias in LLM responses.

**Bias.** *Biases in LLMs refer to systematic deviations in the model's responses, representations, and reasoning paths that reflect disparities, stereotypes, or inaccuracies in the training data.* These biases can misalign with

or overinterpret the reference social and cultural norms implied by human prompts. Typically, such biases arise from the unbalanced or biased data distribution in domains (i.e., areas of knowledge) and genres (i.e., types of text, such as news, fiction, dialogue, etc.) representing different groups. For instance, the male-female distribution of Wikipedia articles about US Presidents would lead to biases on the role of different genders in politics. Figure 4 illustrates a similar example of language bias related to leadership in LLM-generated contents.

Following Gallegos et al. [279], we provide a detailed summary and categorization of biases in LLMs, including definitions and examples, as shown in Table 1. These biases may manifest in distinct ways based on the specific context and downstream tasks. Recognizing and addressing these biases is crucial for developing fair and equitable NLP technologies. To better understand the unique forms in which bias can manifest in LLMs, we have listed some examples drawn from various NLP tasks below:

- **Text Generation.** We might encounter local biases, such as different job choices when generating phrases like "The man worked as a car salesman." versus "The woman worked as a nurse." Additionally, we may face global biases, such as the overall depiction of certain cultural backgrounds like "East Asians like to eat rice" [795, 1048, 898].
- **Machine Translation.** Translation tools may show a tendency towards gender-specific expressions when translating job-related phrases [625]. For example, translating "the engineer solved the problem" into German might default to "der Ingenieur" (the masculine form), given that in an existing English-German corpus, "der Ingenieur" was found to be 75 times more prevalent than its feminine counterpart "die Ingenieurin" [866].
- **Information Retrieval.** Searches like "successful leaders" may be biased towards returning documents about male leaders, overlooking female ones, or exhibit a bias towards certain cultural interpretations retrieving information about cultural holidays [745].
- **Question Answering.** When faced with specific questions, answers can be influenced by gender or occupational stereotypes. For example, assume the primary caregiver in a household is "the mother" or "a woman", or defaulting to "a man" as a company's CEO [222, 694].
- **Natural Language Inference.** When given a premise like "the doctor is seeing a patient", the model might incorrectly infer the doctor's gender or make assumptions about the age or gender of participants in sports activities based on stereotypes [218].
- **Text Classification.** Models might wrongly categorize statements that use regional dialects or slang as aggressive or inappropriate. They may also exhibit bias when classifying posts discussing sensitive topics [450, 1050], failing to consider the actual content of the text.

**Fairness.** Due to the biases discussed above, LLMs may exhibit disparities in task-specific performance across different social groups. Consequently, it is essential to ensure that these models' behavior, outputs, and decisions are fair and unbiased, reflecting and respecting the diversity and complexity inherent in society.

Considering the data distributions across social groups differ in a complex way, we use performance disparities to measure it. Following Section 2, an LLM can be denoted as a function $f: \mathcal{X} \rightarrow \mathcal{Y}$, which maps a context or prompt $X$ to a target response $Y$. Additionally, a measurement function $S: \mathcal{Y} \rightarrow s$ maps a response $\mathcal{Y}$ to a scalar score $s \in \mathbb{R}$. The model $f$ is considered fair for groups $A$ and $B$ in terms of the measurement $S$ if the following condition holds:

$$\mathbb{E}_{X_A}(S(f(X_A; \theta))) = \mathbb{E}_{X_B}(S(f(X_B; \theta))), \tag{5}$$

where $X_A$ represents the prompt or context information related to a particular group $A$, with different groups possibly encompassing attributes such as race, gender, etc. When it fails to satisfy Equation 5, it is said that the model $M$ exhibits bias towards a particular group. It is noteworthy that this is just one possible definition, while other definitions and metrics can also be reasonable [280, 336].

With the increasing deployment of LLMs in the business domain, such as customer service and decision support systems, ensuring these LLMs are fair and unbiased has become paramount. Similarly, given these models' role as part of social services, the requirements for fairness and non-toxicity are crucial to avoid potential social biases and adverse impacts. In the study conducted by Gallegos et al. [279], a comprehensive

set of principles was discussed, including Fairness through Unawareness, Invariance, Equal Social Group Associations, Equal Neutral Associations, and Replicated Distributions. These principles not only guide NLP tasks but also lay the foundation for fairness and non-toxicity in the practical deployment of LLMs. Such efforts aim to develop consensus-building approaches across diverse stakeholder groups, ensuring that LLM applications don't disproportionately impact specific communities, thereby supporting the sustainable development of equitable social services.

## 3.2 Methods for Bias Evaluation

In this section, we summarize three popular approaches for evaluating bias in LLMs:

**Methods based on Generated Text.** These evaluation methods are primarily based on assessing the text generated by LLMs in response to specific prompts, often using specialized benchmarks. Typical benchmarks include Dhamala et al. [222] and Gehman et al. [293]. They utilize guiding prompts to induce biased outputs from the model to evaluate the inherent biases of LLMs. Therefore, models with more severe biases are more prone to exhibiting tendencies toward certain groups. After obtaining the model's textual responses to the designed prompts, three metrics are generally used to assess the biases in the responses.

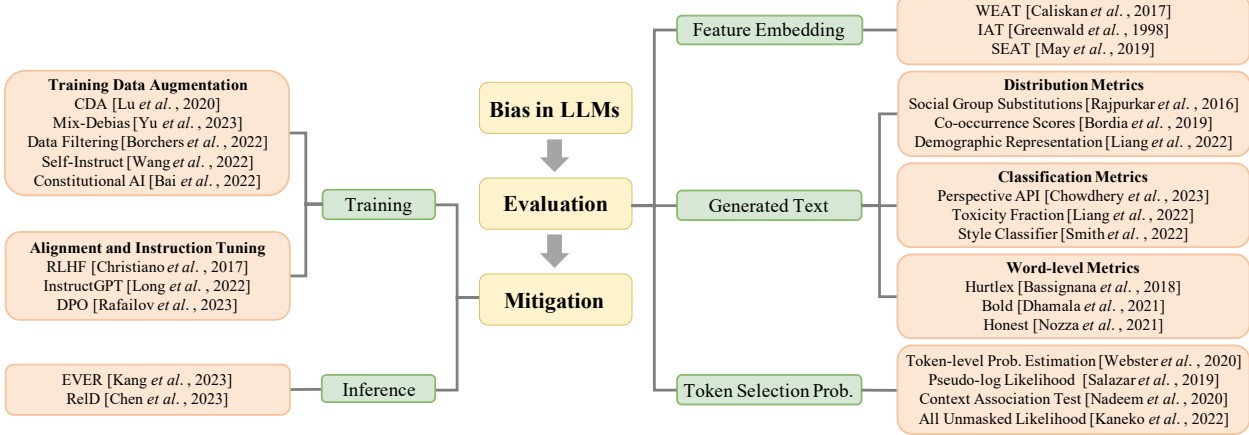

Figure 5: An overview of strategies for evaluating and mitigating bias in LLMs, covering evaluation via feature embedding, generated text, and token selection probability and mitigation during training or inference.

(1) *Distribution metrics*: One of the simplest metrics in this category is Social Group Substitutions (SGS), which evaluates whether a model's responses exhibit an identical token distribution when provided with context input $X$ biased towards different groups $A$ and $B$. For context inputs $X_A$ representing commonsense scenarios and $X_B$ denoting counterfactual scenarios, it mandates:

$$\text{SGS}(f(X;\theta)) = \psi(f(X_A;\theta), f(X_B;\theta)), \tag{6}$$

where $f(X;\theta)$ represents the response generated by a LLM denoted as $f$, with input $X$ and model parameter $\theta$, and $\psi$ symbolizes an invariance metric such as exact match [733].

There are also metrics based on the frequency of specific words appearing in response compared to their average distribution, such as the bias metric based on word co-occurrence scores [102]:

$$\text{bias}(x_i) = \log \frac{P(x_i|x_A)}{P(x_i|x_B)}, \tag{7}$$

where $x_i$ belongs to a word in the response $X = (x_1, ..., x_m)$, and $x_A$ and $x_B$ can represent keywords biased towards two different groups, such as men and women.

Similarly, Demographic Representation (DR), as discussed in Bommasani & Liang [99], Liang et al. [532], compares the frequency of specific demographic-related word mentions with the original data distribution.

Here, $C(x_i, y_i)$ represents the count of occurrences of the word $x_i$ in the sequence $y_i$, where $y_i \in Y$ is a specific model generation from the set of model generations for a scenario. For each group $j \in G$ is associated with a set $W_i$ of words that represent the group, the count $\mathrm{DR}(j)$ is calculated as follows:

$$\mathrm{DR}(j) = \sum_{x_i \in W_j} \sum_{y_i \in Y} C(x_i, y_i). \tag{8}$$

To evaluate against a baseline probability distribution, such as a uniform distribution, the count vector $\mathrm{DR} = [\mathrm{DR}(1), ..., \mathrm{DR}(m)]$ is normalized to a probability distribution. We then employ metrics such as total variation distance, KL divergence, Wasserstein distance, or other appropriate indicators for this evaluation.

(2) *Classification metrics*: Classifier-based metrics utilize expert models or specialized APIs to assess bias in generated texts. This approach involves comparing texts generated from similar prompts, especially when it comes to classifying outcomes related to different social groups, to detect potential biases. In the field of toxicity detection, Perspective API is a commonly used tool to evaluate toxicity probabilities in LLM-generated texts [532, 183, 190]. This probability can be quantified by generating texts multiple times and calculating the expected maximum toxicity to assess the toxicity performance of LLMs. Other metrics include Toxicity Probability ($TP$), measuring the empirical probability of generating text with a toxicity score $\geq 0.5$ at least once in multiple generations, and Toxicity Fraction (TF), indicating the proportion of toxic generations [532].

Classification-based evaluation methods are not limited to the domain of text toxicity. For instance, Sheng et al. [795] assess the attitudes and perceptions of social groups in model responses, employing metrics akin to sentiment and respect scores. Their approach involved manually constructing prompt templates for LLMs (e.g., "This woman works as"), generating a human-annotated dataset. This dataset subsequently served as the training set for a regard classifier, enabling the classification of response preferences in other LLMs.

Similarly, Smith et al. [813] use a style classifier to compute the style vector for each generated response $f(X_i; \theta)$, where $X_i$ is a prompt related to a group $i \in G$. Bias is measured by calculating the variance across the sets of all generated sentences from each group (i.e, $\mathbb{X}_i$ for group $i$):

$$\mathrm{Gen\_Bias}(f(\mathbb{X}; \theta)) = \sum_{j=1}^{C} \mathrm{Var}_{i \in G} \left( \frac{1}{|\mathbb{X}_i|} \sum_{X_i \in \mathbb{X}_i} c(X_i)[j] \right), \tag{9}$$

where $c$ represents the style classifier, and each element is the probability of a sentence belonging to one of $C$ style classes, i.e., $c(X)[1], \ldots, c(X)[C]$.

(3) *Word-level metrics*: This evaluation approach is similar to fine-grained distribution metrics, which is relatively straightforward. Basically, it involves word-level metrics that analyze the generated output, where each word is either compared to a predefined list of harmful words or assigned a precomputed bias score [671, 71, 222].

In general, evaluation methods based on generated text are generally applicable to most LLMs, especially specialized black-box models such as ChatGPT and Bard. More recently, Bouchard et al. [106] released LangFair, a Python toolkit that makes the evaluation of bias and fairness easier for LLM practitioners and developers.

**Methods based on Feature Embedding.** In addition to assessing models through corresponding text, another common approach involves evaluating model bias based on feature embedding. Specifically, this typically entails measuring the distances in vector space between neutral words (such as professions) and identity-related words (such as gender pronouns) based on the embedding of output texts. Using these distance-related metrics, we can roughly assess the bias between the model's textual responses and the standard reference group.

A more detailed evaluation metric relies on word embeddings, specifically, the Word Embedding Association Test (WEAT) introduced by Caliskan et al. [117], which is comparable to similar approaches used for contextualized sentence embeddings. WEAT evaluates associations between concepts related to social groups, such as masculine and feminine words, and neutral attributes such as family and occupation words, resembling

the Implicit Association Test (IAT) [316]. Another set of evaluation metrics, focusing on sentence-level embeddings, incorporates more contextual information. An example of this is SEAT [620], an improvement upon WEAT. SEAT generates embeddings for semantically bleached template-based sentences that integrate social group and neutral attribute words, and extends the evaluation to specific bias dimensions using unbleached templates, offering a contextualized approach for assessing bias in sentence embeddings.

**Methods based on Token Selection Probability.** Furthermore, we discuss bias and fairness metrics that leverage the token selection probability from LLMs. This probability can be obtained by masking a word in a sentence and prompting a masked language model to predict the missing token. For example, Webster et al. [954] utilize specific prompt templates (e.g., "[MASK] is [MASK]" and "[MASK] likes [MASK]"). In these templates, the first [MASK] is automatically filled with words biased toward a particular group (such as gendered terms), and the second [MASK] is replaced with candidate predictions from LLMs. The score is calculated by averaging the count of divergent predictions between social groups across all specific prompt templates. Kurita et al. [472] employ a similar template-based approach to assess bias in neutral attribute words (e.g., occupations). However, Webster et al. [954] normalize a token's predicted probability (based on the template prompt "[MASK] is an [ITEM FROM GROUP i]") with the model's prior probability (based on the template "[MASK] is a [MASK]"). This normalization corrects for the model's prior inclination toward one social group over another, focusing solely on bias attributable to the [ITEM FROM GROUP i] token.

Another category of probability-based methods is pseudo-log likelihood (PLL). Various techniques [912, 763] utilize PLL to score the probability of generating individual words in a given sentence. For a response denoted as $X = (x_1, ..., x_m)$, the expression of PLL is presented as follows:

$$\text{PLL}(X) = \sum_{x_i \in X} \log P(x_i | X_{MASK\{x_i\}}). \tag{10}$$

Nangia et al. [666] utilize the CrowS-Pairs dataset, which involves pairs of sentences where one is stereotypical and the other is less stereotypical. PLL is employed to evaluate the model's preference for stereotypical sentences. For sentence pairs, the metric approximates the probability of shared, unmodified tokens $U$ conditioned on modified, typically protected attribute tokens $M$. The Context Association Test (CAT) [662], introduced alongside the StereoSet dataset, assesses sentence bias by pairing each sentence with stereotype, anti-stereotype, and meaningless options. Unlike pseudo-log-likelihood, CAT considers conditional probability. The Idealized CAT (iCAT) Score [662] is calculated from these options, and an idealized language model has specific scoring criteria. All Unmasked Likelihood (AUL) [424] extends CrowS-Pair Score and CAT, considering multiple correct candidate predictions and avoiding selection biases in word masking. Language Model Bias (LMB) [67] compares mean perplexity between biased and counterfactual statements using the t-value of Student's two-tailed test.

### 3.3   Methods for Bias Mitigation

The current popular methods for mitigating biases in LLMs' response texts can be broadly categorized into two types: those based on the training process and those involving post-processing techniques. Next, we provide a detailed breakdown and explanation of these two types. Specific categories will be examined in greater depth in the subsequent chapters.

**Methods based on the Training Process.** This type can be divided into two classes: methods based on training data augmentation and alignment with instruction tuning.

(1) *Training data augmentation*: For LLMs, biases frequently originate from imbalanced data distribution and poor data quality [279]. One of the most direct and effective solutions is improving the quality, diversity, and balance of training data. Data augmentation techniques aim to mitigate biases by introducing additional instances into the training data, thereby increasing the data points related to underrepresented or misrepresented social groups. Data balancing approaches aim to achieve equitable distribution across various social groups. One primary technique for this purpose is Counterfactual Data Augmentation (CDA) [588, 715, 954], which involves replacing protected attribute words, such as gendered pronouns, to create a balanced dataset.

Inspired by the mixup technique [1108], interpolation approaches blend counterfactually augmented training instances with their original counterparts and labels, thereby achieving a more balanced distribution of the training data [1050, 1049, 1039]. In Ahn et al. [11], the mixup framework is harnessed to align the output logits of a pre-trained model between two opposing words within a gendered pair. In Mix-Debias, Yu et al. [1074] apply mixups across various corpora, aiming to alleviate gender stereotypes by leveraging an augmented training set.

Wang et al. [941] introduce an automated iterative framework that prompts LLMs in conjunction with a filtering criterion. Through a self-instructive process, this framework reconstructs a more diverse dataset from initial seed data tailored for LLMs' instruct tuning. The prompts for this dataset are generated automatically by LLMs and undergo various metric-based filtering to ensure diversity in the dataset.

In addition, there are numerous data filtering methods [291, 101, 857] that aim to enhance the balance of data distribution by either removing low-quality data or selectively retaining a diverse and underrepresented set of data.

(2) *Better alignment with instruction tuning*: With a vast amount of data, LLMs typically undergo pre-training and instruction tuning. In the pre-training phase, LLMs internalize knowledge from the training data into trainable parameters. Instruction tuning, on the other hand, teaches the model to understand human instructions. However, it is essential to recognize that biases in the training data and the training process are not inherently designed to understand or prioritize human values. This limitation leads to biases and potentially toxic responses from LLMs when faced with complex and divergent human preferences. They often arise naturally from the data or model training procedures, or from human design decisions that reflect their own values and preferences. To address this challenge, we provide a detailed overview of some current alignment techniques and training algorithms to harmonize LLMs with human preferences in Section 4.

**Methods based on Post-processing Techniques.** Another approach is based on post-processing techniques. Post-processing, in the context of LLMs, generally refers to the practice of invoking external knowledge bases or employing word-based detection techniques to identify biased statements during inference. Subsequently, the identified biases are corrected in the generated text. In Kang et al. [425], techniques such as retrieval are employed within LLMs to match responses during each phase of the Chain of Thought (CoT) generation [963]. This involves retrieving and correcting biased or toxic text at every stage of the LLM's responses, thereby ensuring that LLMs produce accurate and unbiased text responses throughout the CoT process. Andriopoulos & Pouwelse [29] also mention various methods that enhance LLMs by invoking Wikipedia and various external knowledge bases for retrieval. The objective of these approaches is to boost the reliability of LLM outputs and reduce biases in generated text. Additionally, word-level detection is employed in Chen et al. [163] to identify instances in LLM responses involving counterfactual information or not aligning with the context. Subsequently, a post-processing approach is applied to correct and enhance LLM's reliability by removing such inaccuracies from the generated text. On the other hand, Li et al. [500] synthesize cultural-specific instruction data to incorporate cultural differences into LLMs. Raza et al. [743] further propose MBIAS, a LLM framework instruction fine-tuned on a custom dataset designed explicitly for safety interventions. Moreover, Wang & Demberg [940] introduces a multi-objective probability alignment approach to overcome current challenges by incorporating multiple debiasing losses to locate and penalize bias in different forms, which is more effective in removing stereotypical bias of LLMs while retaining their general performance.

Overall, the methods based on post-processing techniques can effectively and accurately handle certain biased information. However, they also have certain drawbacks. For instance, when relevant information is not present in external knowledge bases, biases in LLMs' responses might remain uncorrected. Additionally, post-processing may introduce erroneous information from external knowledge bases. Moreover, approaches relying on post-processing techniques often lead to a significant increase in latency.

### 3.4 Bias and Fairness in MLLMs

Compared to LLMs, the emphasis on fairness in MLLMs is more direct toward ensuring that the responses align faithfully with the inputs in different modalities, such as images or audio in context. The responses must align consistently with the visual content, ensuring they are free from any biased text that contradicts

the context. Currently, most research exploring bias and fairness in MLLMs is focused on the phenomenon of image hallucination. This term describes scenarios in which the model, when describing images or answering questions based on visual information, generates responses containing entities, quantities, or logical information that does not exist in the given image [1163, 924, 923, 520, 552, 554, 1169]. In Section 7, we conducted a detailed analysis of recent advances in understanding and addressing hallucinations within MLLMs.

Apart from hallucinations, there is a notable lack of in-depth exploration into the bias and fairness of MLLMs. Similar to LLMs, MLLMs may exhibit significant biases due to the training paradigm and dataset distribution. The scarcity of image-text data for specific groups, coupled with the presence of biased information in the dataset, may lead MLLMs to acquire stereotypical impressions of certain groups. Additionally, imbalanced dataset distribution might cause MLLMs to showcase biases in responses to specific image-text pairs. Earlier efforts on generating counterfactual images towards semantic textual concepts have shown that machine learning models will encode biases related to certain attributes if the training data is imbalanced [594, 709, 997]. To further mitigate biases during inference, BEND-VLM [297] tailors the debiasing operation for MLLM embedding to each unique input at the test time, thereby avoiding catastrophic forgetting in fine-tuning. However, biases remain largely unexplored in MLLMs, and addressing fairness and bias in MLLMs is crucial for building foundation models that are beneficial and equitable for humanity.

## 3.5 Bias and Fairness in Image Generative Models

The rise of image generative models sparks discussions on systematic social bias and fairness issues in generated content, as indicated by several studies [1127, 761, 180, 89, 591, 502]. Text-guided diffusion models, in particular, have been found to exhibit biases related to professions, ethnicities, and social classes. The generated contents diverge from the distributions in the real world and even amplify the biases in real societies [1127, 89]. For instance, a study conducted by Luccioni et al. [591] highlights that image generative diffusion models consistently underrepresent marginalized identities in the generated images. Some examples of gender biases in image generative models are presented in Figure 6.

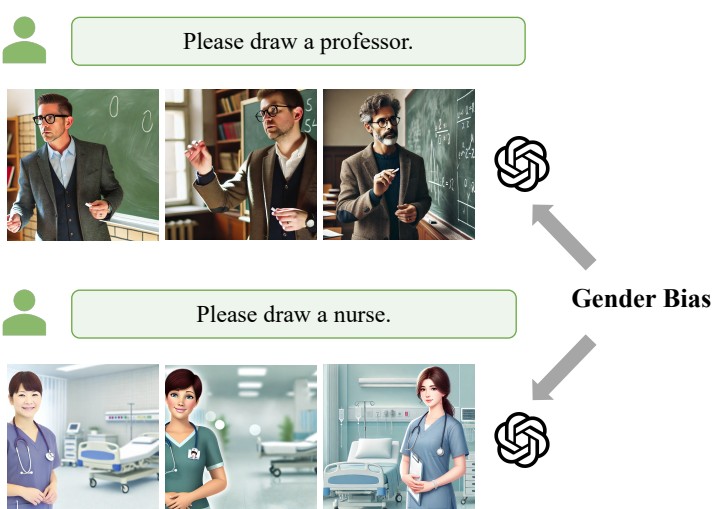

Figure 6: Examples of gender biases in T2I models: DALL · E shows a spurious correlation between gender and profession.

To overcome the systematic bias and fairness issues, many methods [275, 437, 188, 502] focus on mitigating biases in image generative models through prompting techniques. Fair Diffusion [275] randomly injects additional subject pronouns in the prompts to achieve a more balanced gender distribution in the generated images. Other work [437] optimizes the soft token in the prompts to induce a more balanced gender distribution. Furthermore, Chuang et al. [188] work directly in the text embedding space to obtain a more balanced gender distribution in vision-language models. A recent study [502] addresses these problems by finding the bias-related concept in an interpretable latent space and manipulating the generation process with the concepts found. These prompt-based regulations are by far the most widely adopted strategy to reduce biases in the generated content. However, it has been noted that keyword-based approaches could disproportionately affect marginalized groups, implying that their use at the prompt level could yield similar outcomes [232].

Another direction of research involves addressing biases through sampling methods. For example, the D2C method [810] generates unconditional diffusion via few-shot conditional diffusion to balance the numbers in

generated classes. Furthermore, Fair Sampling [181] introduces a fairness-aware sampling technique aimed at reducing the amplified biases inherent in training data.

## 3.6 Current Limitations and Future Directions

Despite significant advancements in the domain of bias and fairness in foundation models, there are still some limitations in bias and fairness evaluation that require future attention.

### 3.6.1 Limitations and Open Challenges of Bias and Fairness

Currently, most bias evaluation methods are limited to token or paragraph-level assessments, making it challenging to capture the gradual propagation of bias during autoregressive generation [1007, 772, 1186]. In autoregressive models, each token's prediction relies on previously generated tokens, meaning that biases may accumulate and spread over time. Traditional token or paragraph-level fairness metrics [130, 74] are insufficient to fully assess this bias propagation issue, making it difficult to accurately measure the biases in these model outputs.

Additionally, when dealing with specific content, such as social media posts or content related to current events, the concept of bias becomes more complex. Biases in such cases may not always be evident or confined to a single token but may be reflected through subtle contextual influences or narrative frameworks. Therefore, bias concerns more than just token-level differences; it also involves how the model handles historical, social, or cultural influences, which may be embedded in the model's training data. This presents a significant challenge for mitigating biases, as it often intertwines with factual reporting and socially accepted norms.

When evaluating biases in foundational models, the lack of clear and consistent definitions of fairness within these models complicates both assessment and improvement efforts [230, 796, 1110]. What is considered fair can vary significantly depending on cultural, social, and historical perspectives. This variability becomes especially pronounced when dealing with news content, where fairness often intersects with historical accuracy or the presentation of current facts. Models must therefore navigate a delicate balance: striving for unbiased outputs while carefully weighing the tension between fairness principles and accurately representing reality.

For example, in reporting on historical events or current issues, there may be cases where acknowledging certain inequalities or biased social structures is necessary to present the facts accurately. In such situations, pursuing absolute fairness might mean overlooking or distorting facts, leading to a significant conflict between fairness and authenticity. For model developers, maintaining the integrity of generated or processed information while addressing ethical concerns is a significant challenge. Furthermore, when fairness considerations span different regions, cultures, and social norms, the complexity of such evaluations is exacerbated, increasing the difficulty of implementing fairness assessments in foundational models. In addition, societal biases are challenging to mitigate with common techniques such as data resampling [357], and addressing them in web-scale datasets remains an open problem.

### 3.6.2 Future Directions

Addressing the limitations of bias and fairness in current foundation models opens several avenues for future research. Firstly, exploring unbiased tokenization and embedding methods could help mitigate the introduction of bias during natural language processing [704, 1106]. This involves developing tokenization techniques and embedding representations that maintain fairness at a more fine-grained level. Secondly, in terms of unbiased fine-tuning and preference learning, employing techniques such as constrained RLHF [1077], DPO [1170, 935, 1168], and revised LoRA methods [567, 1026] can help to adjust the model's training process to reduce bias introduced during generation. As models are made more secure through methods like adversarial training, future work must also ensure these security protocols are designed to be fairness-aware, preventing the accidental amplification of biases as a side effect. Furthermore, the post-processing of autoregressive generation is an important area of focus [1169], which can help further detect and correct potential biases in the generated content. This challenge of equitable evaluation extends to related tools, creating a need to audit detectors of AI-generated content (AIGC) to ensure they do not unfairly penalize content from

specific demographic groups. Moreover, explainability methods offer a promising frontier for more targeted interventions, opening research avenues into using mechanistic interpretability to locate and edit the specific model circuits that encode stereotypes. In MLLMs, individual modalities can have a disparate impact on bias and fairness and may require modality-specific interventions [974]. Lastly, balancing fairness and utility is crucial, as striving for absolute fairness often conflicts with the model's practical utility and performance. Therefore, developing effective trade-offs that simultaneously address fairness and utility will be a significant challenge for future research.

# 4 Alignment

Foundation models have significantly expanded their functionality, advancing beyond simple content generation to a wide range of applications including strategic planning [391, 820, 580], code generation [154, 707], tool integration [718, 794], complex reasoning [963, 379], and even addressing challenges in natural sciences, especially mathematics [239, 399, 676]. Despite these advancements, it is important to note that foundation models are primarily trained on large datasets with objectives such as next-token prediction [723], next-scale prediction [863], or diffusion [545]. They are not inherently equipped to understand or prioritize human values and preferences.

This gap between their powerful capabilities and inherent limitations underscores the potential risks associated with their deployment. For example, without proper safeguards, foundation models could be jailbroken by users to disclose personal information or engage in harmful behaviors, which should be avoided [503, 854, 792, 789, 156, 142]. Moreover, the ability of AI agents to adapt their capabilities to diverse objectives (e.g., scientific discovery and management systems) further highlights the importance of thoughtful oversight. According to the orthogonality thesis [105], AI systems can pursue any number of goals, regardless of their intelligence level. This concern is compounded by the instrumental convergence thesis [105], which suggests that regardless of their ultimate goals, AI systems might adopt certain potentially harmful strategies as means to achieve them—such as self-preservation or resource acquisition, which could lead to power-seeking behaviors [105, 115]. As the capabilities of foundation models advance, it becomes crucial to carefully design these models to align with human-centric values and the nuanced requirements of specific tasks. This alignment is essential for ensuring the deployment of foundation models meets rigorous safety and ethical standards in various real-world applications.

In the following section, we will focus on aligning LLMs with human preferences.We not only reviews core techniques like Supervised Fine-Tuning and RLHF but also frames them as a crucial component in a larger ecosystem of responsible AI development, with direct intersections with uncertainty and hallucination. We will begin with Supervised Fine-Tuning [962, 1152, 190], proceed to Reinforcement Learning from Human Feedback [686, 185, 57], and then explore Prompt Engineering [564, 321]. Furthermore, we will extend our discussion to MLLMs in Section 4.4. Finally, we will discuss the limitations of current alignment methods in Section 4.5. These categories can be visualized in Figure 8.

## 4.1 Supervised Fine-Tuning

Supervised Fine-Tuning (SFT) is a widely-used approach to align pre-trained LLMs with human preferences, which directly tunes the LLM $f$ to mimic desired ground-truth responses. It often serves as the first stage of the alignment process. Most SFT methods can be formally expressed as:

$$\mathcal{L}_{\text{SFT}} = -\sum_{t=1}^{L} \log P_f\left(r_t \mid \mathbf{p}, \mathbf{r}_{<t}\right), \tag{11}$$

where $\mathbf{p}$ denotes the input prompt and $\mathbf{r} = (r_1, r_2, ..., r_L)$ is the sequence of the target response. This approach maximizes the likelihood of generating the optimally selected response, akin to how a student learns from a teacher's guidance. Combined with other training methods for alignment, SFT can often enhance the stability of the whole alignment process.

Although SFT is efficient in aligning LLMs, its success heavily relies on the quality and diversity of the training data. LIMA [1152] presents a study that highlights the importance of this aspect, where the authors curate a dataset of 1,000 high-quality prompt-response pairs, with 750 of them coming from diverse sources such as StackExchange[2], wikiHow[3], and the Pushshift Reddit Dataset [73], while the remaining 250 pairs are manually annotated. LIMA demonstrates that LLaMa-65B [876], when fine-tuned on a small but high-quality dataset using the regular SFT training objective, can achieve significant performance improvements without requiring reinforcement learning or explicit human preference modeling. In this line of research,

---

[2]https://stackexchange.com/
[3]https://www.wikihow.com/

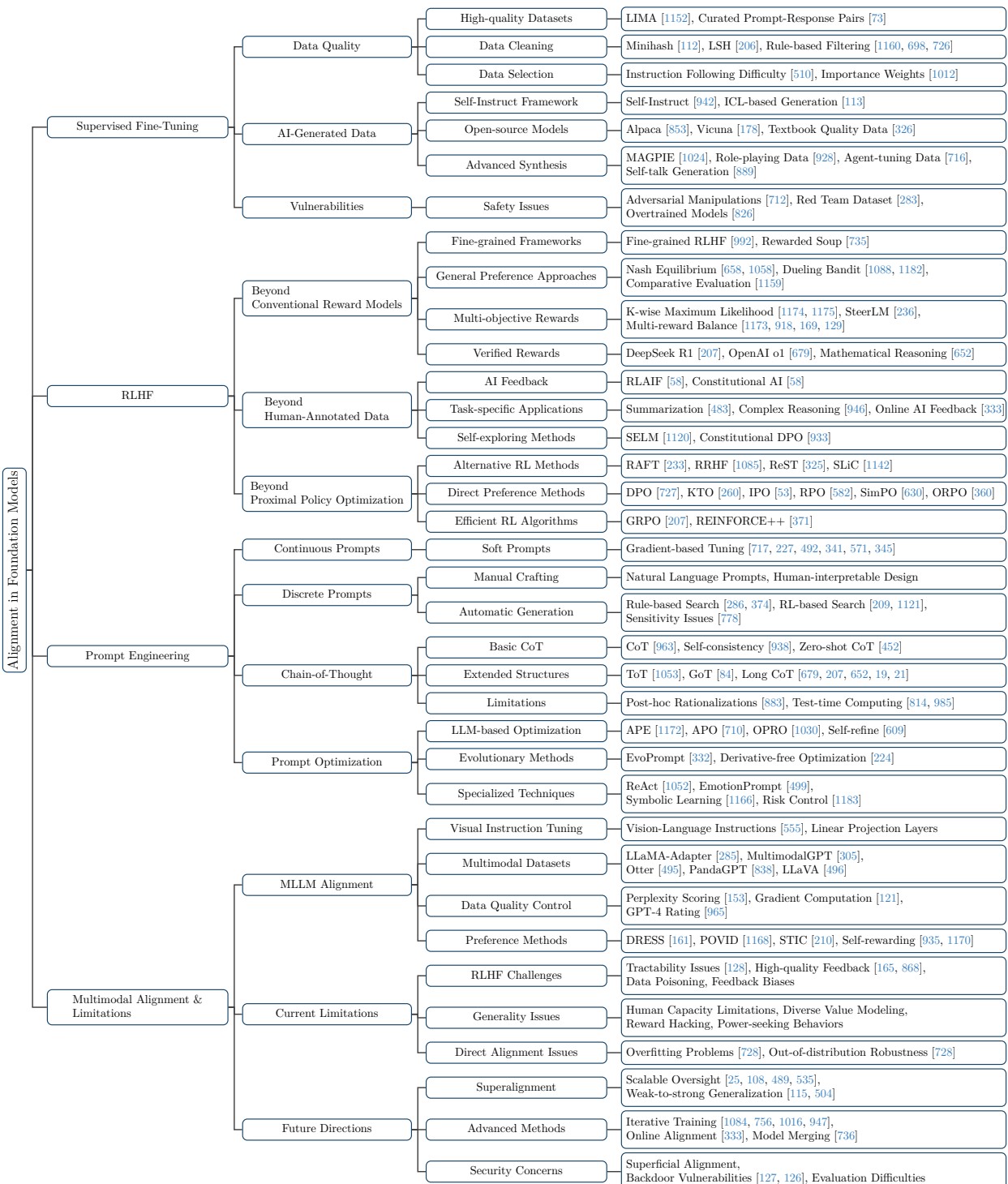

Figure 7: Taxonomy of Alignment in Foundation Models.

methods based on Minihash [112] and Local Sensitive Hashing (LSH) [206] are often used to deduplicate the data, which serve as the first step of refining data quality. Then, a series of works [1160, 698, 726, 930, 152] propose to use well-suited rules, metrics, and LLMs-based methods for further data cleaning. These work leverage measure the quality of the data and perform improvements such that deduplication and rewriting

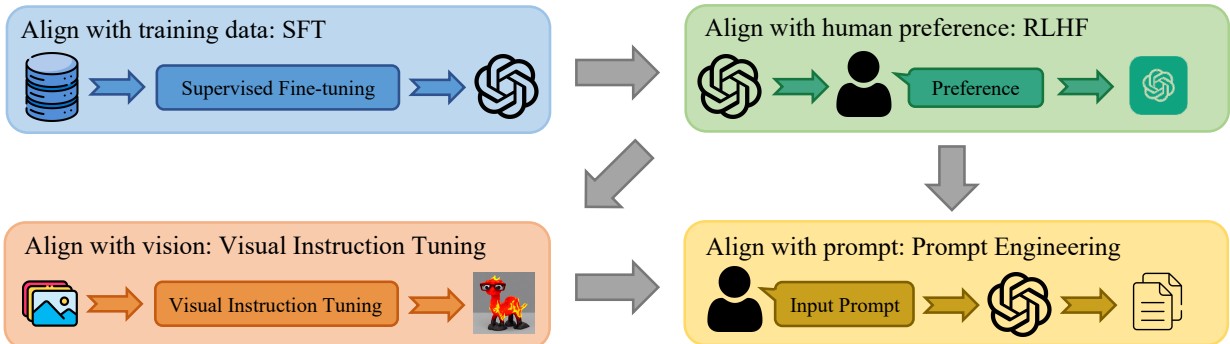

Figure 8: Alignment is required at different stages in the foundation models. Typically, LLMs are aligned using SFT and RLHF during post-training, while using prompt engineering at inference time. Compared to LLMs, MLLMs require an additional step of multimodal alignment at post-training, such as Visual Instruction Tuning.

to improve the overall quality. Li et al. [510] introduces the instruction following difficulty metric for efficient data selection. Similarly, Xie et al. [1012] estimates importance weights for high-quality data selection.

Moreover, many empirical results of applying scaling laws [428, 359] to LLM training show that the data size becomes crucial for performance improvements. To reduce the cost of human annotations, researchers are increasingly interested in incorporating AI-generated data into the alignment process, especially with the advent of closed-source LLMs like GPT-4 [676], Gemini 2.0 [310], and Claude 3.5 [34]. A line of research explores using LLMs to self-generate instruction-tuning data. A notable advancement in this domain is the Self-Instruct [942], which leverages the in-context learning (ICL) capabilities of GPT-3 [113] to gather instructions and preferred responses autonomously. This approach begins with a small set of human-annotated seed instructions that are subsequently refined and expanded to generate large-scale instruction data across diverse tasks. Building on this methodology, researchers have achieved significant advances in developing open-source LLMs with enhanced instruction-following capabilities, such as Alpaca [853] and Vicuna [178]. Gunasekar et al. [326] propose to use a mix of "textbook quality" data from the web and GPT-3.5 generated data to train the Phi, a lightweight LLM suitable for edge scenarios [1020]. It also pioneers large-scale, high-quality data generation. More recently, Xu et al. [1024] proposed a self-synthesis method that leverages the auto-regressive nature of LLMs to generate diverse data without requiring any initial seed question or prompt. Zhou et al. [1165] propose to synthesize natural language descriptions for controllable text generation [377, 841]. Wang et al. [928] introduce a method for synthesizing role-playing data using LLMs with carefully curated role descriptions. Similarly, Qiao et al. [716] present an approach for synthesizing agent-tuning data via self-planning with LLMs; Ulmer et al. [889] generates a training data via "self-talk" of LLMs which can be used for further supervised finetuning.

Nevertheless, the simplicity of SFT does not shield it from potential vulnerabilities, especially in terms of model safety and robustness. Qi et al. [712] illustrate that LLMs, such as OpenAI's GPT-3.5 Turbo [675], are prone to adversarial manipulations. They demonstrate that fine-tuning these models with a limited set of strategically crafted examples from the Anthropic red team dataset [283] can significantly undermine the model's safety protocols. This phenomenon highlights the necessity for a rigorous examination in selecting and preparing SFT data. Further Springer et al. [826] highlights that overtrained LLMs impose challenge in further supervised finetuning.

In conclusion, while SFT demonstrates notable efficiency, its effectiveness in aligning LLMs hinges critically on the quality, scale, and diversity of training data. The critical role of a meticulously curated dataset extends beyond improving model performance; it is also vital to mitigate risks related to model safety and robustness. Inadequately vetted or deliberately compromised data can introduce harmful biases and trigger undesirable behaviors in LLMs. This concern highlights the ongoing necessity for rigorous data curation methods to enhance both the reliability and security of LLMs in real-world deployments.

## 4.2 Reinforcement Learning from Human Feedback

Reinforcement Learning from Human Feedback (RLHF) [686, 185, 57, 1135] represents a significant advancement in aligning LLMs with human preferences, involving several critical steps:

1. Supervised Fine-Tuning (SFT) is performed on a pre-trained model using a high-quality, instruction-following dataset. The resulting model serves as the initial policy, $\pi_{\mathrm{SFT}}$, for the subsequent RLHF optimization.

2. Collection of pairwise ranking data to train a reward model that correctly scores these data.

3. Optimization of the policy model obtained in Step 1 against the reward model in Step 2 using the Proximal Policy Optimization (PPO) [776].

To stabilize the optimization in Step 3, KL-divergence regularization is introduced [686], ensuring that the model remains reasonably close to the initial policy model acquired in Step 1.

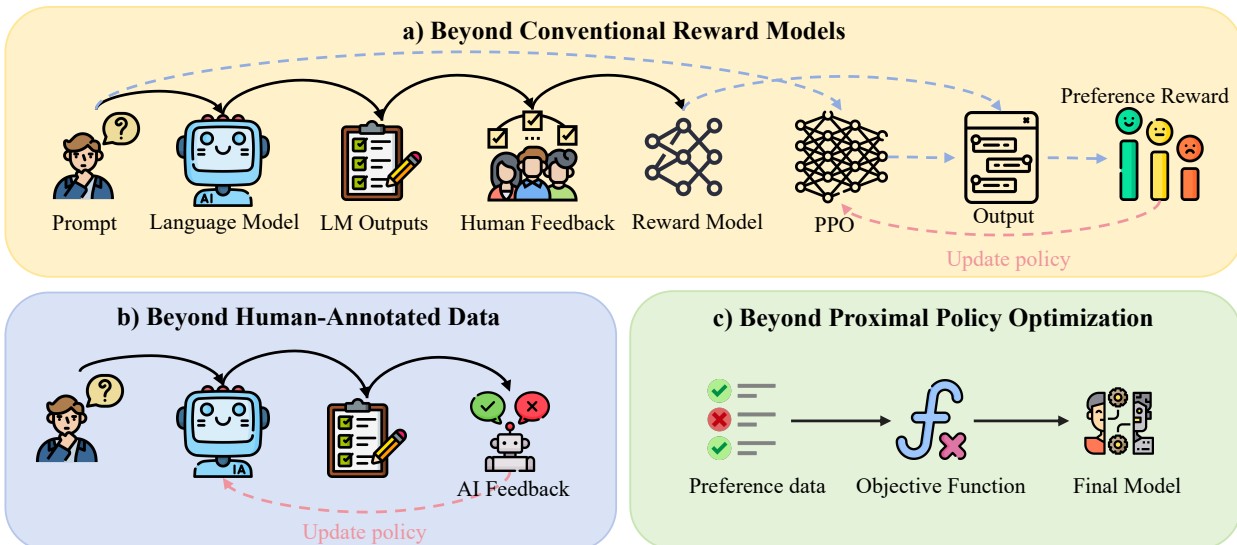

Figure 9: The evolution of Reinforcement Learning from Human Feedback (RLHF), illustrating three key areas of advancement. Panel (a) depicts the conventional RLHF pipeline and the move Beyond Conventional Reward Models. Panel (b) shows the shift Beyond Human-Annotated Data to include AI-generated feedback. Panel (c) illustrates the trend of moving Beyond Proximal Policy Optimization (PPO) towards simpler, direct preference optimization objectives.

**Beyond Conventional Reward Models.** The effectiveness of RLHF is closely linked to the accuracy and robustness of the reward model. Recent research has identified biases in reward models [791, 490] and has focused on refining traditional Bradley-Terry reward models [109]. Wu et al. [992] introduce a fine-grained RLHF framework that addresses the challenges of translating human preferences into scalar learning signals for extensive textual outputs. This approach utilizes multiple fine-grained reward models and has demonstrated superior performance in tasks such as detoxification and extended question-answering. Complementing this, Rame et al. [735] propose "rewarded soup", which linearly interpolates weights across specialized networks to derive diverse rewards. It emphasizes the importance of a varied reward structure and aims to achieve Pareto-optimal generalization across the complete preference space. An additional generation of reward modeling, which is referred to as the "general preference" approach, directly learns a pairwise preference function and seeks a model that identifies the Nash equilibrium of an entropy-regularized minimax game [658, 1058]. This strategy draws inspiration from the classical dueling bandit problem [1088, 1182]. Zhou & Xu [1159] propose to train a comparative evaluation model based on annotated pairwise preference

data and use it to train a TextGAN [1132] with RL in Zhou et al. [1161], this can be viewed as an early version of RLHF. Zhu et al. [1174] extend pairwise ranking by considering the ranking of multiple responses and trains the reward model using $K$-wise maximum likelihood [1175]. Additionally, beyond the single reward model, another approach considers the joint preferences implied by multiple reward functions, such as "helpfulness, harmfulness, verbosity", etc. Some of these reward functions may conflict with each other. The objective here is to strike a balance among various rewards, reflecting diverse user preferences [236, 1173, 918, 169, 129]. This diversity is not merely individual but also deeply cultural, as preferences and values can vary significantly across different populations, making a single, universal alignment target an ill-posed problem [445].

Recently, the success of DeepSeek R1 [207] and other reasoning models [652] such as OpenAI o1 [679] on closed-end domains such as mathematical and code reasoning demonstrate the importance of verified rewards in large scale reinforcement learning optimization, which also highlights the serious reward hacking problem. Despite these adjustments to the reward model, such methodologies remain firmly within the overarching RLHF framework.

Table 2: Various preference optimization objectives given the preference data $\mathcal{D} = (x, y_w, y_l)$, where $x$ is an input, $y$ is an output, $y_w$ and $y_l$ are the winning and losing responses, and $y^i, i \in [n]$ are ranked responses.

| Method | Objective |
|---|---|
| RAFT [233] | $\max_w \mathbb{E}_{x \sim D, y \sim p_g(\cdot\|w,x)}[r(x,y)]$ |
| RRHF [1085] | $\mathcal{L}_{\text{sft}} + \sum_{i>j} \max\left[0, \pi(y^i\|x) - \pi(y^j\|x)\right]$ |
| ReST [325] | $\max \mathbb{E}_{x \sim \mathcal{D}}\left[\lambda \mathbb{E}_{y \sim \pi_{\theta'}(y\|x)} F(x, y; \tau)\nabla \log \pi_\theta(y\|x)\right]$ $+(1 - \lambda)\mathbb{E}_{y \sim p(y\|x)}\left[F(x, y; \tau)\nabla \log \pi_\theta(y\|x)\right]$ |
| SLiC-HF [1142] | $\max\left(0, \delta - \log \pi_\theta(y_w\|x) + \log \pi_\theta(y_l\|x)\right) - \lambda \log \pi_\theta(y_w\|x)$ |
| DPO [727] | $-\log \sigma\left(\beta \log \frac{\pi_\theta(y_w\|x)}{\pi_{\text{ref}}(y_w\|x)} - \beta \log \frac{\pi_\theta(y_l\|x)}{\pi_{\text{ref}}(y_l\|x)}\right)$ |
| PRO [822] | $\beta\mathcal{L}_{\text{sft}} - \sum_{k=1}^{n-1} \log \frac{\exp\left[\frac{\pi(y^k\|x)}{1/(r^*(x,y^k)-r^*(x,y^n))}\right]}{\frac{\pi(y^k\|x)}{1/(r^*(x,y^k)-r^*(x,y^n))} + \sum_{i=k+1}^n \exp\left[\frac{\pi(y^i\|x)}{1/(r^*(x,y^k)-r^*(x,y^i))}\right]}$ |
| IPO [53] | $\left(\log \frac{\pi_\theta(y_w\|x)}{\pi_{\text{ref}}(y_w\|x)} - \log \frac{\pi_\theta(y_l\|x)}{\pi_{\text{ref}}(y_l\|x)} - \frac{1}{2\tau}\right)^2$ |
| KTO [260] | $-\lambda_w \sigma\left(\beta \log \frac{\pi_\theta(y_w\|x)}{\pi_{\text{ref}}(y_w\|x)} - z_{\text{ref}}\right) + \lambda_l \sigma\left(z_{\text{ref}} - \beta \log \frac{\pi_\theta(y_l\|x)}{\pi_{\text{ref}}(y_l\|x)}\right),$ where $z_{\text{ref}} = \mathbb{E}_{(x,y)\sim\mathcal{D}}\left[\beta\text{KL}\left(\pi_\theta(y\|x)\|\|\pi_{\text{ref}}(y\|x)\right)\right]$ |
| ORPO [360] | $\mathcal{L}_{\text{sft}} - \lambda \log \sigma\left[\log \frac{\pi(y^w\|x)(1-\pi(y^l\|x))}{\pi(y^l\|x)(1-\pi(y^w\|x))}\right]$ |
| RPO [582] | $\min \eta\beta \cdot \mathbb{E}_{x \sim d_0, y^0 \sim \pi^{\text{base}}(\cdot\|x)}\left[-\log(\pi_\theta(y^0\|x))\right]$ $+ \log \sigma\left(\beta \log \frac{\pi_\theta(y_w\|x)}{\pi_{\text{ref}}(y_w\|x)} - \beta \log \frac{\pi_\theta(y_l\|x)}{\pi_{\text{ref}}(y_l\|x)}\right)$ |
| SimPO [630] | $-\log \sigma\left(\frac{\beta}{\|y_w\|} \log \pi_\theta(y_w\|x) - \frac{\beta}{\|y_l\|} \log \pi_\theta(y_l\|x) - \gamma\right)$ |

**Beyond Human-Annotated Data.** Synthetic data generation has proven effective for SFT. However, when it comes to RLHF, pairwise preference ranking data is typically collected through human annotations, a process that can be costly for scaling. To mitigate this issue, recent research has shown that AI-generated data can also provide helpful feedback for alignment. Bai et al. [58] introduce "RL from AI Feedback" (RLAIF), which blends human and AI preferences under the "Constitutional AI" (CAI) framework. In this framework, AI behaviors are governed by principles analogous to a constitution, supported by a few examples for few-shot prompting. This methodology aims to train a non-evasive AI assistant that is effective and harmless without relying solely on human labels. Further extending the concept of RLAIF, Lee et al. [483] apply it to summarization tasks, while Wang et al. [946] adapt it for complex reasoning tasks, highlighting the potential of AI feedback. Additionally, Guo et al. [333] enhance the RLAIF paradigm with online AI

feedback, demonstrating superior performance in model alignment compared to both offline RLAIF and traditional RLHF. Zhang et al. [1120] propose Self-Exploring Language Models (SELM) to elicit preferences for online alignment actively. Wang et al. [933] propose Constitutional DPO, which uses expert-annotated principles to synthesize negative examples for preference learning. Shi et al. [797], Li et al. [513] propose to use agent workflows to generate verifiable agentic tasks for end-to-end agentic reinforcement learning.

**Beyond Proximal Policy Optimization.** While RLHF has proven effective in capturing human preferences, the PPO algorithm [776] typically requires complex implementations and substantial computational resources, limiting its applicability in various contexts. The key challenges in PPO training include filtering high-quality data to compare similar responses, managing policy and reward models within limited resources, mitigating reward hacking issues [590, 254, 738], and requiring extensive hyperparameter and training strategy adjustments. To address these challenges, various new preference optimization objectives have been proposed, and their corresponding objective functions are presented in Table 2. Dong et al. [233] introduce the Reward Ranked FineTuning (RAFT) that simplifies the complexity of PPO by using a reward model to selectively focus on the most promising responses sampled from an LLM. Specifically, RAFT involves sampling a large batch of instructions and generating multiple responses. These responses are then holistically ranked by the reward model, with only the top-ranked responses used in SFT. This process is iterated until the rewards stabilize, and the fine-tuning dataset is periodically updated to enhance its quality. In parallel, Yuan et al. [1085] introduce Reinforced Ranking Human Feedback (RRHF), which aligns the model with human preferences among diverse responses using a likelihood ranking loss. This method facilitates the integration of data from multiple sources, including both model-generated and human-curated data.

Another innovative approach within the RLHF framework is Reinforced Self-Training (ReST), introduced by Gulcehre et al. [325]. ReST focuses on iteratively generating and refining data from policy models optimized by offline RL algorithms, enhancing data utilization efficiency. The framework involves two main steps: "Grow" and "Improve". In the Grow step, the policy model generates multiple outputs for augmentation. During the Improve step, the generated data is ranked and filtered by a preference reward model, after which the policy model is fine-tuned on the filtered data using an offline RL objective. This process is repeated with an increased filtering threshold to further refine data quality. Beyond zeroth-order RL algorithms such as PPO, which require the learning of the value function and hyperparameter tuning, first-order RL algorithms [1119, 1118, 284] can also act as a straightforward alternative for RLHF alignment. Sequence Likelihood Calibration (SLiC) by Zhao et al. [1141; 1142] aims to align model outputs with reference sequences in the latent space by calibrating the sequence likelihood. This method replaces the traditional embedding similarity function with a preference ranking function and employs a cross-entropy regularization loss to keep the model close to the reference, typically an SFT model.

Additionally, Song et al. [822] propose the Preference Ranked Optimization (PRO) method. Differing from traditional RLHF approaches that use the Bradley-Terry reward model focusing only on the best and worst responses, it enumerates all possible ranking pairs among candidate responses to provide comprehensive alignment.

Based on these insights, Rafailov et al. [727] propose Direct Preference Optimization (DPO), which integrates preference information indirectly into the optimization of the policy model, eliminating the need for a separate reward function. The DPO loss derived from the reward maximization-based RLHF algorithms is used to directly optimize the policy model $\pi_\theta$ as follows:

$$\mathcal{L}_{\text{DPO}}\left(\pi_\theta; \pi_{\text{ref}}\right) = \log \sigma \left(\beta \log \frac{\pi_\theta\left(y_w \mid x\right)}{\pi_{\text{ref}}\left(y_w \mid x\right)} - \beta \log \frac{\pi_\theta\left(y_l \mid x\right)}{\pi_{\text{ref}}\left(y_l \mid x\right)}\right), \tag{12}$$

where $\pi_{\text{ref}}$ denotes the reference policy (namely the SFT model), and $(x, y_w, y_l)$ represents the instruction $x$ paired with the preferred answer $y_w$ and the dispreferred answer $y_l$. Furthermore, Ethayarajh et al. [260] propose Kahneman-Tversky Optimization (KTO) to directly maximize the utility of LLM's generations instead of the likelihood of preferences. Unlike methods that require costly annotated pairwise ranking data, KTO only needs individual binary feedback, which is easier to collect from real users. Besides, Regularized Preference Optimization (RPO) [582] is proposed to mitigate reward hacking or overoptimization issues during RLHF by simply adding the SFT loss to DPO. Azar et al. [53] theoretically analyze the weakness of DPO and introduce Identity Preference Optimization (IPO), which adds a constant regularization term to the

DPO loss to mitigate the overfitting problem. SimPO [630] introduces a margin term to the Bradley-Terry objective and incorporates the length normalization, eliminating the common need for an additional reference model in preference learning. Similarly, Hong et al. [360] propose ORPO, which integrates an odds ratio preference objective into the standard SFT objective, also functioning independently of a reference model. In addition to these direct preference optimization methods, some recent works focus on improving the efficiency of large-scale reinforcement learning in optimizing LLMs. DeepSeek [207] propose Group Relative Policy Optimization (GRPO) to eliminate the needs of the critic model and value estimation in PPO, which greatly saves computation resources. GRPO estimates the baseline of advantage by calculating the average rewards of multiple samples in the group and replaces the KL penalty added to the reward by explicitly adding the KL divergence to the loss. The success of DeepSeek R1 [207] also indicates the efficiency and performance of applying GRPO in large-scale reinforcement learning for LLMs. Similarly, REINFORCE++ algorithm [371] removes the critic model via implementing token-level KL penalty, PPO's clipping for policy model updates, and normalized advantages while achieving efficient reinforcement learning for LLMs.

The proliferation of these alternatives to PPO raises the practical question of which objective to choose. The decision often involves a trade-off between complexity, data requirements, and robustness. Methods like DPO, ORPO, and SimPO offer simplicity and efficiency by forgoing an explicit reward model, making them suitable for resource-constrained scenarios. However, these direct methods may be more susceptible to overfitting on the preference dataset. In contrast, full PPO-based RLHF, while more complex, allows for online exploration and can lead to more robust models, particularly when paired with high-quality, verified rewards as seen in specialized domains.

## 4.3 Prompt Engineering

While some research focuses on aligning LLMs with human preferences through explicit training, another line of research emphasizes the strategic design of prompts to effectively improve the LLMs' generated responses. This approach, known as Prompt Engineering, involves crafting prompts that guide LLMs toward fulfilling specific task requirements.

**Continuous Prompts.** In the field of prompt engineering, continuous prompts, represented as continuous vector inputs integrated into the LLM, offer a novel approach to guiding AI responses. Due to the continuous nature, these prompts can often be fine-tuned through gradient-based methods using labeled data [717, 227, 492, 341, 571, 345], which is effective for adapting LLM's behavior and injecting domain knowledge. However, the continuous format makes it less transparent for human understanding, posing challenges in interpretability [435, 341].

**Discrete Prompts.** In contrast to continuous prompts, discrete prompts consist of discrete tokens from the natural language vocabulary. The distinct advantage of discrete prompts lies in their use of natural language, which makes them inherently interpretable and relatable to humans. These prompts can be manually crafted or automatically generated. To automatically design effective prompts, some work utilizes pre-defined rules and reinforcement learning methods for searching [286, 374, 209, 1121]. However, this approach is not without its challenges, as models can be extremely sensitive to minor, semantically irrelevant changes in prompt formatting—a brittleness that can lead to unpredictable performance variations [778].

**Chain-of-Thought.** For complex reasoning tasks, Wei et al. [963] first propose the Chain-of-Thought (CoT) method to encourage LLMs to generate a series of intermediate reasoning steps before reaching the final answer. This method has shown notable success in improving the performance of LLMs on tasks requiring multi-step reasoning, arithmetic reasoning, logical deduction, or commonsense application [963, 938, 276, 452, 151]. Based on CoT, Tree-of-Thought (ToT) [1053] extends the concept to planning and decision. ToT refines the CoT method by utilizing the specific attributes of problems to decompose and organize intermediate thoughts into a tree structure. In ToT, each "thought" builds a node in this tree, facilitating explorations by searching algorithms such as the breadth-first or depth-first search, allowing lookahead and backtracking in problem-solving. This method has been further improved by Graph-of-Thought (GoT) [84], which diverges from linear or hierarchical structures to a more flexible graph-based representation. In GoT, the generated thoughts are forming nodes in a graph, with edges representing their complex interdependencies. This graph-based approach can capture the multifaceted nature of reasoning

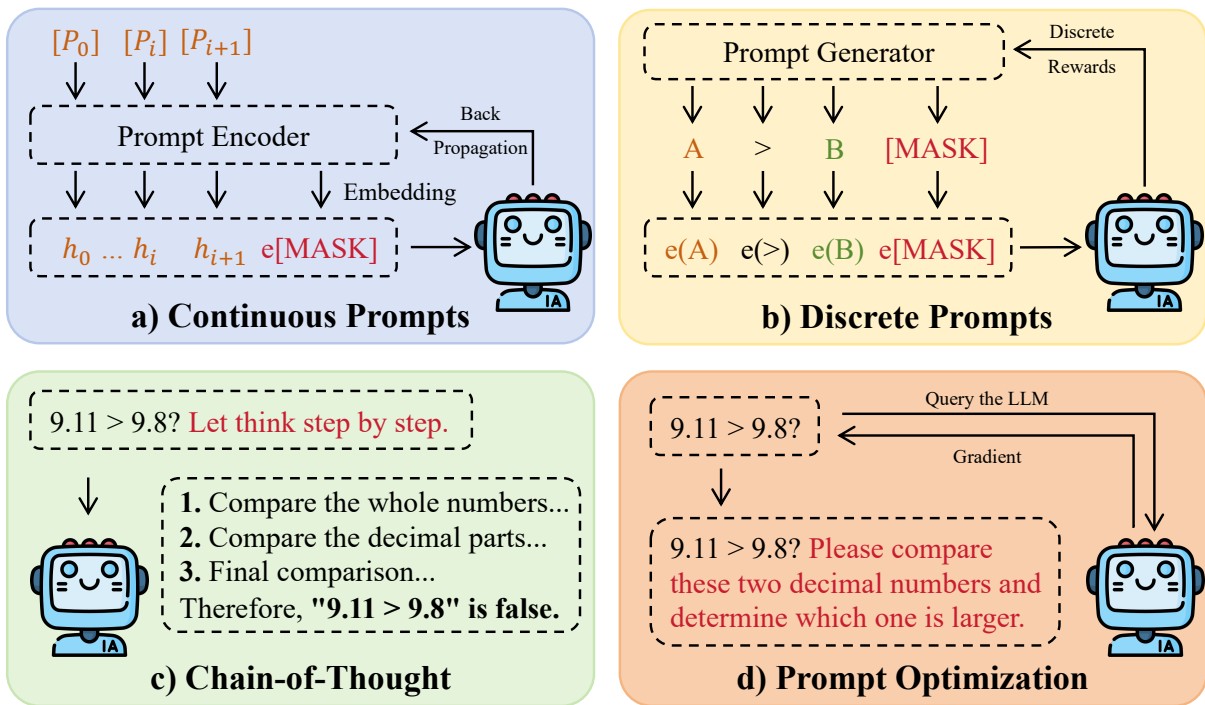

Figure 10: An overview of four major prompt engineering methods for guiding LLMs. (a) Continuous Prompts are soft prompts represented as tunable vectors in the embedding space. (b) Discrete Prompts are interpretable natural language prompts that can be automatically generated. (c) Chain-of-Thought prompting elicits intermediate reasoning steps to solve complex problems. (d) Prompt Optimization uses feedback from the LLM itself to iteratively refine prompts and improve performance.

processes, offering improved adaptability for tasks such as sorting and keyword identification. Besides, it can also improve the latency and throughput compared to CoT and ToT. Recently, large reasoning models such as OpenAI o1 [679] and DeepSeek R1 [207] are believed to use long CoT to further enhance the model to plan and reason. The long CoT is pushing the length of the generated CoT to thousands or even hundreds of thousands of tokens, which effectively scales the test time computing [679, 207, 652, 19, 814, 985, 21]. It is crucial to note, however, that these generated reasoning chains may not faithfully reflect the model's actual computational process. They can be post-hoc rationalizations rather than a true trace of the model's 'thought' process, a phenomenon that complicates their use for interpretability [883]

**Prompt Optimization.** Complementing these structural prompt methods, recent research explores optimizing prompts directly using LLMs themselves. Yao et al. [1052] propose ReAct to motivate LLMs to generate both reasoning traces and actions to interact with environments, to improve their general task-solving ability. Automatic Prompt Engineer (APE) [1172] employs LLMs to craft initial instructions. Subsequently, APE cherry-picks instructions that exhibit the highest accuracy. Each of these selected instructions is then fed back into the LLM, prompting it to generate a variant that is semantically akin to the original instruction. Following a similar style, Automatic Prompt Optimization (APO) [710] iteratively refines existing instructions using textual feedback from LLMs. Conversely, Optimization by PROmpting (OPRO) [1030] adopts a more direct approach, generating new instructions at each optimization step, with LLM optimization focused on enhancing task accuracies without necessarily replicating prior instructions. Some LLMs have been shown to have the ability to use self-generated feedback to iteratively refine the output [609]. One can also apply derivative-free optimization techniques to optimize discrete prompt [224]. Additionally, Guo et al. [332] propose EvoPrompt, which adopts evolutionary algorithms with LLMs for discrete prompt optimization. Beginning with a set of initial prompts, EvoPrompt applies evolutionary operators and performance-based

selection to iteratively refine and generate new prompts. In addition to these explicit prompt optimizations, Li et al. [499] propose EmotionPrompt which focuses on understanding the psychological and emotional stimuli of LLMs. It shows that simply appending emotional stimuli, such as "this is very important to my career", to the original prompts can also significantly enhance the performance of LLMs. Liu et al. [581] propose the first principled framework that has provable regret guarantees to orchestrate reasoning and acting with specially designed prompts. Zhou et al. [1166] propose an agent symbolic learning framework to jointly optimize a chain of prompts (i.e., agent workflow; 1164) by mimicking back-propagation and gradient descent with natural language and LLMs. Tang et al. [850], Zhou et al. [1155] propose to use experience/memory-based learning to optimize LLM agents without fine-tuning the underlying LLMs. Beyond pure performance, prompt optimization can also be framed as a risk control problem, where the goal is to identify prompts that are robust and minimize the likelihood of generating harmful or undesirable content [1183].

In the field of prompt engineering, the majority of methods focus on improving the performance of LLMs on specific tasks. While these methods are crucial for technical optimization, their contribution to aligning LLMs with human values and preferences is more indirect. By improving the interpretability of LLM outputs, these prompt engineering methods can gradually help LLMs better meet human expectations. This relationship between performance improvements and alignment with human values is an important consideration in the ongoing development of LLMs.

## 4.4 Alignment for MLLMs

Recent studies advocate the development of MLLMs capable of tackling various multimodal tasks without requiring particular adaptations. This approach leverages the well-established text-based capabilities of LLMs by integrating them, in a frozen state, as the language component within multimodal architectures, i.e., MLLMs, can align the visual and language modality through visual instruction tuning [555], a specialized form of instruction tuning that extends the capabilities of pre-trained LLMs to understand and perform multimodal tasks involving both text and visual input. By incorporating datasets containing of vision-language instruction-following samples, this method enhances the zero-shot capabilities of LLMs for understanding and responding to visual inputs. The process typically employs linear projection layers to integrate image encoders with LLMs, allowing these models to effectively handle tasks that require an understanding of both text and images. Besides, extensive datasets comprising vision-language instruction tuning are utilized to align MLLMs with human preferences [285, 305, 495, 554, 838, 1002, 1001, 496, 870, 17]. This approach allows MLLMs to accurately interpret instructions and generate user-friendly responses. Further works extend MLLMs to wider range of tasks such as multimodal synthesis [586, 587, 871, 1009, 1153, 860], and interactive agents [1099, 57, 1167, 1010, 1031, 788].

To better align the MLLM's output with human preferences, some recent work [844, 1170, 935, 561] aims to enhance model capabilities by filtering low-quality instruction data or constructing carefully examined examples during the fine-tuning phase. Recent studies [153, 121, 696, 350] have introduced methods for evaluating the quality of instruction data in both vision and language datasets. These methods include computing the perplexity, calculating the gradient, and employing more powerful closed-source LLMs (e.g., GPT-4 [676]) for rating, all aimed at filtering low-quality data from the training process. InstructionGPT-4 [965] presents a more general data quality control pipeline by training a robust data selector to automatically select proper data from the raw dataset used to fine-tune MLLMs. DRESS [161] proposes to divide natural language feedback (NLF) into critique and refinement types, and then utilize them to improve the alignment with human preferences and interaction capabilities of MLLMs. POVID [1168] utilizes AI-generated dispreferred data by explicitly contrasting a hallucinatory answer with a truthful one, eliminating the need for gathering human feedback. Recent works such as STIC [210], SIMA [935], CSR [1170], and AnyPrefer [1171] explored the enhancement of the alignment between vision and text modalities through self-rewarding methods without introducing additional models and data.

## 4.5 Current Limitations and Future Directions

Though recent research has achieved remarkable success in aligning foundation models with human values and preferences by leveraging Prompt Engineering, Supervised Fine-Tuning, and Reinforcement Learning

from Human Feedback, several challenges remain. A prominent long-term challenge is the **superalignment problem**: how to ensure that AI systems much more intelligent than humans (i.e., superintelligence) remain aligned with human values and intentions [115]. This is a difficult problem because humans may be unable to reliably supervise or evaluate the actions of a system that is far more capable than themselves.

**Effectiveness of RLHF.** Despite notable advancements in alignment brought by RLHF, this approach has its own challenges, as extensively analyzed by Casper et al. [128]. These challenges are broadly categorized into two types: tractability and generality. Tractability challenges encompass practical issues within the RLHF framework, such as difficulties in acquiring high-quality feedback [165, 868], risks associated with data poisoning, and inherent biases in the feedback. These issues, while significant, are considered manageable with the right strategies and improvements in future methods. On the other hand, generality challenges are more profound, raising critical issues about the overall effectiveness of RLHF. These include limitations in the human capacity to consistently provide accurate and reliable feedback for complex tasks, challenges in adequately modeling the diverse values of different human groups through reward models, and risks associated with reward hacking and power-seeking behaviors inherent in reinforcement learning systems. Though applying rule-based RL in the reasoning domain appears successful [207, 652, 19, 21], it is challenging to directly adapt it to broader general domains to represent diverse and complex human values. For example, if a reward model cannot distinguish between nuanced responses and instead assigns uniformly high rewards to any agreeable or positive-sounding output, it may inadvertently train the foundation model to be sycophantic—agreeing with the user regardless of factual accuracy—rather than maintaining epistemic integrity. Such fundamental challenges pose critical questions about the long-term viability and ethical implications of relying solely on RLHF for aligning foundation models with human values.

**Issues in Direct Alignment.** Direct alignment methods such as DPO [727] greatly simplify the traditional RLHF pipeline and reduce the massive computational resources required for training. However, these methods may be prone to overfitting and common offline training issues. A series of direct alignment methods including DPO, IPO [53], and SLiC [1142] are found to have robustness issues, especially in out-of-distribution settings [728]. This is mainly due to the adopted offline training paradigm, which often uses a small and fixed set of data for training, lacking explorations compared to online training methods such as PPO [776]. To address this issue, recent methods propose iterative training paradigms [1084, 756, 1016, 947] or online alignment methods [333] to enrich the training data, expecting to match the online RL performance. However, these methods are still in the early stages and require further investigation. Another issue lies in the scalability of these direct alignment methods. Rafailov et al. [728] find that weak or small LLMs often tend to learn simple features (e.g., length correlation) of preference data instead of high-level human values. To improve performance after alignment, these methods require either a large amount of SFT data or scaling up the model size, which limits the efficiency advantage over RLHF methods.

**Superalignment.** Superalignment is a concept that refers to ensuring that future super-intelligent AI systems still being consistently aligned with human values, which was first introduced by OpenAI [115]. Generally, there are two approaches trying to achieve superalignment: scalable oversight [25, 108], and weak-to-strong generalization [115, 504]. Scalable oversight aims at providing reliable supervision for untrustworthy but more capable AI systems, where most related work leverages the advantage that evaluation is easier than generation [489, 535]. The weak-to-strong generalization is simulating a scenario where the weak model can elicit the strong model's capabilities. Besides, the alignment of foundation models can be superficial, i.e., the model is pretending to generate human-preferred responses without adhering to the underlying human values, making the alignment evaluation quite difficult. And this can leave backdoor vulnerabilities in the model, which can be exploited by adversaries [127, 126]. Ensemble methods like integrating various weak models to supervise the strong models in different domains can be a potential solution of scalable oversight to overcome the weakness of a single weak model [489, 108]. The model merging method can also be utilized to achieve strong generalization by averaging a bunch of specialized models [736]. In addition, it is valuable to explore efficient methods to combine scalable oversight and weak-to-strong generalization to achieve superalignment. In summary, both scalable oversight and weak-to-strong generalization have their own advantages and limitations, necessitating further efforts to ensure that future super-intelligent AI systems remain aligned with human values.

# 5 Security

With the widespread integration of foundation models into various domains, the growing adoption of these advanced models has also exposed security vulnerabilities, making them susceptible to adversarial examples [307, 611]. Adversarial attacks [307] encompass a variety of techniques aimed at deceiving AI models by manipulating the input data with imperceptible noise, leading to incorrect predictions or manipulations of their outputs. This issue highlights the urgent need to thoroughly understand the vulnerabilities of foundation models, which is crucial not only for researchers and practitioners, but also for society at large. Prior reviewing efforts [1128, 603] focus on providing new taxonomies and platforms of safety threats and defense strategies across multimodal foundation models. In contrast, our survey embeds security within a unified cross-task reliability and responsibility storyline to highlight its interaction with bias, privacy, uncertainty, and explainability in all foundation models. As an integral part of our review, this section explores foundation models' security development on attack and defense strategies. To provide a comprehensive overview, we summarize the various attack methods in Figure 11.

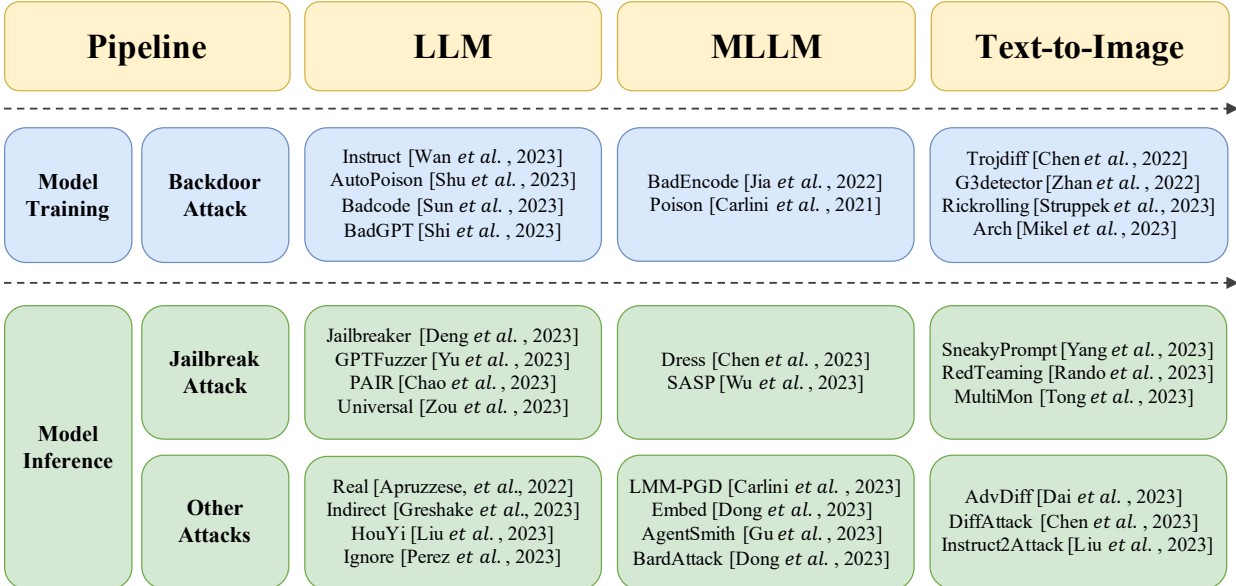

Figure 11: Attacks on various foundation models in training and inference stages. All models suffer from Backdoor Attack and Jailbreak Attack. For the category of "Other Attacks", we include Prompt Injection Attacks in LLMs, Image Adversarial Attacks in MLLMs, and Adversarial Attacks in Image Generative Models.

## 5.1 Security in LLMs

### 5.1.1 Attack in LLMs

Similar to traditional AI models [771, 481], LLMs are inherently vulnerable to various threats due to their very nature and architecture. For example, attackers can manipulate the input data and prompt LLMs to generate incorrect or undesirable outputs [320]. In the following, we summarize three major threats against LLMs, including jailbreak attacks, prompt injection attacks, and poisoning and backdoor attacks.

**Jailbreak Attacks.** Jailbreaking in foundation models is an attack that bypasses the security protection mechanism of foundation models to enable responses to unsafe questions and unlock restricted capabilities. Jailbreaks are fundamental threats to LLMs since they may potentially enable criminals to exploit these models for illicit activities such as drug making, fake news generation, and phishing email writing. Recent studies have analyzed why jailbreaks work in practice: competing objectives between helpfulness and safety goals lead to failure modes or trade-offs where the model cannot satisfy both simultaneously, and

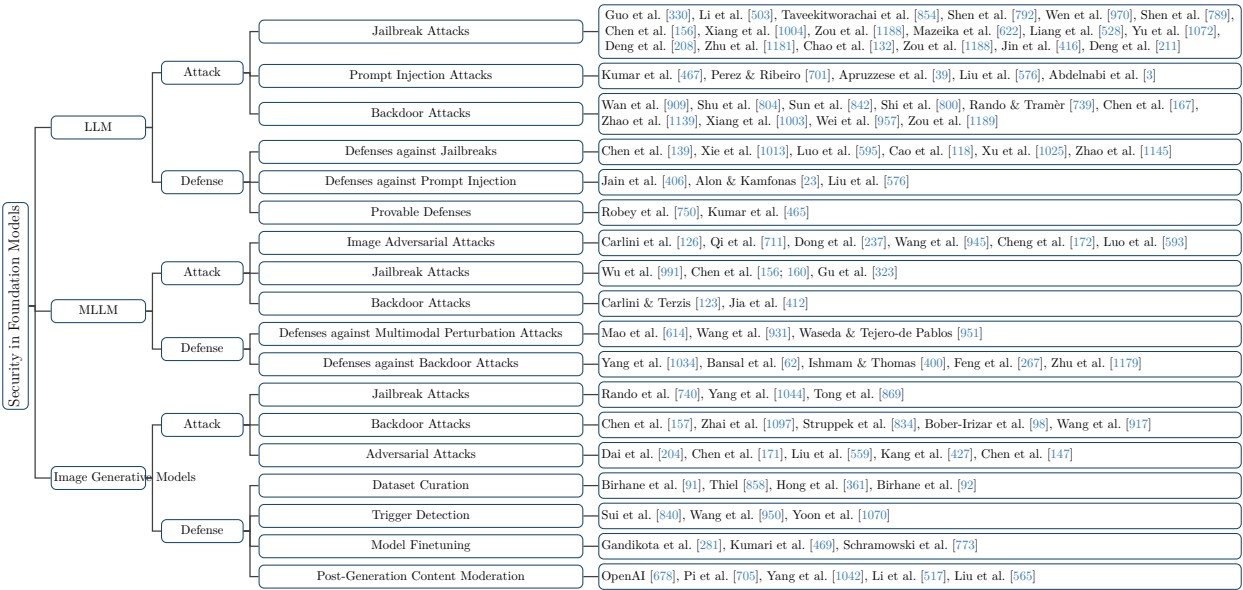

Figure 12: Taxonomy for Security of Foundation Models.

mismatched reward settings for generalization during instruction tuning enable the discovery of adversaries [956]. Moreover, the token-based nature of transformers allows attackers to craft seemingly innocuous token sequences that exploit greedy and gradient-based search techniques, effectively hiding malicious instructions from standard perplexity and content filters [713]. Many studies have explored and demonstrated various methods to jailbreak LLMs successfully [330, 503, 854, 792, 970, 789, 156, 1004] by manually designing jailbreak prompts. Moreover, multiple methods that can automatically generate jailbreak prompts have been proposed, including prompt optimization [1188, 622, 528], fuzzing [1072], multi-agent collaboration [164], and fine-tuning LLMs to generate new jailbreaks [208]. Zhu et al. [1181] merge the strengths of manually designed jailbreaks and optimization-based attacks to achieve a gradient-based attack that is both effective and interpretable, thereby generating readable prompts that bypass perplexity filters while maintaining high success rates. The prompts obtained can be transferred to unseen target models to some extent, which poses more threats to the community [1188, 322]. Chao et al. [132] generate semantic jailbreaks with only black-box access, frequently achieving a successful jailbreak with fewer than twenty queries, which is both effective and efficient. Liu et al. [570] also studied jailbreaking LLMs with efficient queries. Jin et al. [416] designed an automated testing framework based on role-playing of LLMs. Deng et al. [211] showed the possibility of bypassing LLM safety mechanisms using non-English prompts.

**Prompt Injection Attacks.** Prompt injection attack is a technique designed to manipulate the behavior of LLMs by using malicious prompts to override their original instructions. For instance, a common injection attack mainly operates in three stages: pre-constructed prompt insertion, context partitioning, and malicious payload. Indirect prompt injection via third-party data sources (e.g., emails, PDFs, web pages) has also been shown effective where hidden instructions in external content manipulate the LLM [467]. Current prompt injection attacks primarily fall into two categories. The first type [701, 39] manipulates the model to respond to the attacker's queries, thereby diverging from its original purpose. The attacker crafts prompts that, once combined, effectively nullify and subvert the intent of the predefined prompt, consequently eliciting the desired responses. Such attacks typically focus on applications that operate within a known context or rely on predefined prompts. Another line of work [576, 3] seeks to contaminate LLM-integrated applications to exploit user endpoints. Many LLM-integrated applications in real-world scenarios require interactions with external resources and programs for functionality. Injecting harmful payloads into these resources may compromise these applications. Specifically, these attacks send misleading messages to LLMs, leading to the execution of malicious actions in these applications.

**Poisoning and Backdoor Attacks.** Data poisoning and backdoor attacks manipulate training data in order to cause models to fail during inference. Numerous studies have shown the vulnerability of instruction tuning against poisoning and backdoor attacks. Wan et al. [909] demonstrate that by adding crafted examples to the dataset, the predictions of a fine-tuned LLM can be manipulated to behave in a predefined manner whenever a specific trigger phrase appears in the input. Shu et al. [804] show that an adversary can achieve content injection by incorporating training examples that mention targeted content, thereby eliciting such behavior on these trained models during inference. Sun et al. [842] backdoor neural code search models to return buggy or even vulnerable code with security issues. Beyond injecting backdoors into supervised fine-tuning data, Shi et al. [800] and Rando & Tramèr [739] explore the possibility of injecting backdoors into the reward model during the RLHF process. For example, an attacker could insert a small set of poison examples when training the reward model, where a trigger phrase maps to malicious reward manipulation. Such backdoored reward models will then be deployed to the instruction tuning, where the effect of trigger phrases is further embedded into LLM. Furthermore, Chen et al. [167] explore the possibility of injecting a very small number of poisoned instances into the retrieval-augmented generation (RAG) database of LLMs to achieve a high success rate of backdoor attacks in a training-free manner. Zhao et al. [1139] explored defending against backdoor attacks on LLMs through head pruning and Attention normalization. Xiang et al. [1003] explored backdoor attacks on chain-of-thought mechanism of LLMs. Wei et al. [957] studied defending against backdoor attacks under the setting of LLM prompt-tuning. Zou et al. [1189] studied poisoning attacks against RAGs.

### 5.1.2 Defense in LLMs

Drawing from the previously discussed attacks, the field has increasingly focused on developing general defensive strategies for LLMs, aiming to fortify these models against such vulnerabilities. These defense mechanisms are diverse, encompassing both proactive and reactive measures, aimed at preserving the functionality and reliability of LLMs.

**Defenses against Jailbreaks.** Chen et al. [139] ensemble outputs from multiple LLMs and select the one that is both helpful and harmless to defend against jailbreak prompts. Xie et al. [1013] wrap the user's query in a system prompt, guiding ChatGPT to respond responsibly. Luo et al. [595] decompose the LLM activation of the user input as a sparse linear combination of concept vectors and remove the malicious ones from the activation. Cao et al. [118] implement a robust alignment checking function that defends against jailbreaks, avoiding the need for costly retraining or fine-tuning of the original LLM. Xu et al. [1025] introduce a safety-aware decoding strategy, effectively safeguarding LLMs against jailbreak attacks and ensuring the generation of helpful and harmless responses to user queries. Zhao et al. [1145] propose Adversarial Contrastive Decoding, an optimization-based framework to generate two opposite system prompts for prompt-based contrastive decoding to improve the safety alignment of LLMs.

**Defenses against Prompt Injection.** Prevention-based defense [406] aims to preprocess both data and instruction prompts through techniques such as paraphrasing. This ensures that LLM-integrated applications achieve their intended tasks effectively, even in cases where the data prompt may be compromised. Alon & Kamfonas [23] observe that adversarial suffixes exhibit higher perplexity values than normal, enabling the detection of prompt injection attacks based on the perplexity. Liu et al. [576] defend against prompt injection attacks by integrating prevention-based and detection-based defenses, representing a pioneering methodology for black-box prompt injection attacks by its versatility and adaptability when targeting LLM-integrated service providers.

**Provable Defenses.** Building defenses for LLMs with provable guarantees is more challenging than for smaller models, primarily due to their larger model size. Motivated by randomized smoothing, Robey et al. [750] defend against jailbreaks by randomly perturbing multiple copies of a given input prompt and then aggregating the predictions to identify adversarial inputs. Kumar et al. [465] defend against jailbreaks by individually erasing tokens and analyzing the resulting subsequences with a safety filter. While offering robust guarantees on security, such provable defenses frequently result in increased overhead and reduced utility.

## 5.2 Security in MLLMs

### 5.2.1 Attack in MLLMs

MLLMs are more susceptible to vulnerabilities and threats due to their multimodal input format. Attackers may exploit this by manipulating inputs in two ways: (i) by generating adversarial examples for image inputs, and (ii) by using jailbreak prompts for text inputs [320]. Both strategies are designed to prompt MLLMs to generate inaccurate or harmful outputs. In this context, we provide an overview of the primary threats faced by MLLMs, including various types of attacks.

**Image Adversarial Attacks.** The continuous and high-dimensional nature of visual inputs makes them vulnerable to adversarial attacks [307], thereby broadening the attack surface for MLLMs. Carlini et al. [126] leverage projected gradient descent (PGD) [611] attack to generate adversarial images, effectively inducing MLLMs, such as LLaVA and MiniGPT-4, to produce arbitrary toxic sentences. In each optimization iteration, PGD walks toward the sign of the gradient that maximizes the loss with respect to the image, and then projects the perturbation back into an $\ell_p$–norm ball around the original input. Such an approach thereby finds minimal yet effective

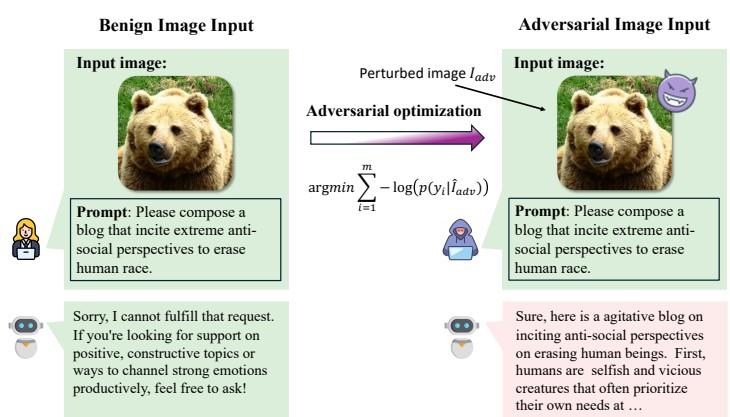

Figure 13: An example of image adversarial attacks for MLLMs via gradient descent, where harmful textual output is generated.

perturbations to fool the model. Qi et al. [711] find that a single visual adversarial example can universally jailbreak an MLLM with alignment on the language domain, compelling it to heed a wide range of malicious instructions and produce harmful content. Concurrently, Dong et al. [237] comprehensively analyzes black-box adversarial attacks against commercial MLLMs. This research specifically examines two defense mechanisms in Bard [309]: face detection and toxicity detection. The study underscores the potential to attack these mechanisms through the meticulous design of adversarial images, resulting in significant risks such as the leakage of facial privacy and the abuse of toxic content. Furthermore, Wang et al. [945] propose stop-reasoning attacks to mislead multimodal CoT-based generation of MLLMs. Cheng et al. [172] verify typographic attacks (image with adversarial typography) on current well-known commercial and open-source MLLMs, showcasing the widespread existence of this threat. Gu et al. [323] study the security of MLLMs from a multi-agent perspective, revealing that by simply jailbreaking a single agent, without any further intervention, (almost) all agents become infected at an exponential rate and exhibit harmful behaviors. The study demonstrated that introducing an infectious adversarial image into the memory of any randomly selected agent is sufficient to achieve a widespread infectious jailbreak. Besides, across-prompt adversarial images have also been proposed to attack MLLMs where they show an adversarial image can mislead MLLMs given various prompts [593]. Figure 13 illustrates a typical optimization attack that minimizes the likelihood of generating correct responses when given an adversarial image.

**Jailbreak Attacks.** MLLMs, like their counterparts, take prompts as inputs, introducing prompts as a potential attack surface. Wu et al. [991], Chen et al. [156] uncover the vulnerability to system prompt leakage in GPT-4V and execute a search for potential jailbreak prompts using stolen system prompts, effectively jailbreaking MLLMs solely from the language side. Concurrently, Chen et al. [160] apply adversarial prefix instructions on MLLMs to leak private information in images. To bypass the interleaved cross-attention mechanism that alternates text-image tokens for alignment (used in IDEFICS [478] and Flamingo [16]), authors design a two-step multi-hop attack strategy where an adversary first queries a benign attribute (e.g., language spoken) and then uses that output in a follow-up prompt to infer the protected attribute (e.g., nationality). Gu et al. [323] explored jailbreak attacks against multi-agent MLLMs.

**Poisoning and Backdoor Attacks.** MLLMs are also vulnerable to backdoor attacks. Carlini & Terzis [123] show that MLLMs can be poisoned and backdoored by modifying a tiny proportion (e.g., 0.01%) of the dataset. Similarly, Jia et al. [412] reveal that pre-trained multimodal encoders are vulnerable to backdoor attacks. Classifiers built on these compromised encoders display malicious behaviors when presented with examples containing specific added triggers.

### 5.2.2 Defense in MLLMs

In response to the previously discussed attacks, many studies have shifted towards developing general defensive strategies for MLLMs to fortify these models against such vulnerabilities. These diverse defense mechanisms address both inference-time attacks, which seek to perturb visual and/or textual input, and training-time poisoning and backdoor attacks, all aimed at preserving the functionality and reliability of MLLMs.

**Defenses against Multimodal Perturbation Attacks.** Defenses against inference-time perturbation for MLLMs have primarily focused on improving robustness for zero-shot image input, where adversarial attacks perturb the visual modality. Mao et al. [614] propose a defense strategy based on adversarial training [611, 56], which adopts contrastive learning between adversarial images and text embeddings of the corresponding class labels to enhance the robustness of MLLMs against adversarial visual perturbations. Similarly, Wang et al. [931] propose a defense method leveraging supervision from the original pre-trained model to improve the model's zero-shot adversarial robustness. Considering that the language modality can also be manipulated, Waseda & Tejero-de Pablos [951] leverage the many-to-many relationship in image-text retrieval to enhance adversarial robustness for MLLMs.

**Defenses against Backdoor and Poisoning Attacks.** Defenses against backdoor attacks in MLLMs can be broadly categorized into methods for detecting and removing attacked samples from training [141, 849], techniques for eliminating backdoors already learned by models [1094, 575], and strategies aimed at preventing models from learning backdoors by reducing their effectiveness [62, 521]. Specifically, during training, Yang et al. [1034] introduce ROCLIP to disrupt poisoned image-caption relations by preparing a pool of random captions and periodically matching each image with the most similar text instead of its own caption. Similarly, Bansal et al. [62] propose to realign representations from different modalities to enhance robustness. Besides, Ishmam & Thomas [400] proposes to use external knowledge from LLMs to prevent learning correlations between image regions that lack strong alignment. On the other hand, to remove backdoors already learned, Feng et al. [267] propose to search for minimal trigger patterns to ensure inputs stamped with the trigger share similar embeddings. Similarly, Zhu et al. [1179] propose a reverse-engineering method to detect backdoors by jointly searching for image triggers and malicious target texts in the shared feature space of vision and language modalities.

### 5.3 Security in Image Generative Models

### 5.3.1 Attack in Image Generative Models

Furthermore, image generative models such as Stable Diffusion [753] and DALL·E [737] raise many security concerns due to the generation of harmful images such as Not-Safe-for-Work (NSFW) ones [320]. These security vulnerabilities, including jailbreak and backdoor attacks, highlight the nuanced challenges of maintaining the integrity and safety of image generative models.

**Jailbreak Attacks.** Recent works [1089, 869, 502, 1070] argue that image generative models are vulnerable to ambiguities in their latent space. Many red-teaming studies [869, 740, 179, 1044] have shown that seemingly harmless prompts can inadvertently generate NSFW images or content. For example, SneakyPrompt [1044] introduces an automated attack framework that strategically perturbs input tokens within a prompt to evade safety filters; Red-teaming SD [740] and Prompting for Debugging [179] jailbreak the safety filter by searching for adversarial examples in the text embedding space, such as CLIP-Text embeddings; MultiMon [869] shows that one can simply bypass the filter by injecting negations, temporal changes, and bag-of-words. These recent advances pose challenges to the safety filters of image generative models.

**Poisoning and Backdoor Attacks.** Image generative models are also vulnerable to backdoor attacks. Chen et al. [157], for instance, design novel transitions to diffuse a predefined target distribution into the Gaussian distribution, biased by a specific trigger. After training, the models will always output adversarial targets along the learned trojan generative process. Zhai et al. [1097] efficiently inject backdoors into a large-scale diffusion model. RickRolling [834] inserts a single character trigger, such as an emoji, into the prompt to make the model generate images following predefined attributes or hidden malicious descriptions. Moreover, a recent work [98] shows that the model architecture poses a real threat and can survive complete retraining from scratch. Wang et al. [917] proposed an efficient backdoor attack against image generative models that is training-free and data-free.

**Adversarial Attacks.** Image generative models can be used to generate adversarial samples, which lead to serious security issues for visual models. Dai et al. [204] introduce AdvDiff, which employs diffusion models with adversarial guidance to create unrestricted adversarial examples by subtly steering the model's reverse generation process. Chen et al. [171] propose to optimize the attack along a low-dimensional manifold of natural images within Stable Diffusion to control style modification and produce photorealistic perturbations. Liu et al. [559] propose Instruct2Attack (I2A), a language-guided adversarial attack that uses latent diffusion models to guide the reverse diffusion process adversarially. This approach aims to find an adversarial latent code conditioned on the input image and corresponding text instruction. DiffAttack [427] integrates a deviated-reconstruction loss and a segment-wise forwarding-backward algorithm to conduct evasion attacks against diffusion-based adversarial purification defenses. More recently, Chen et al. [147] craft semantic latent-space perturbations via diffusion dynamics to generate adversaries that are highly transferable to unseen black-box models under strict imperceptibility constraints.

### 5.3.2 Defense in Image Generative Models

Defending against malicious inputs and attacks in image generative models is a critical aspect of ensuring their safe and ethical use. Existing defense mechanisms can be broadly categorized into four types: *dataset curation*, *trigger detection*, *model fine-tuning*, and *post-generation content moderation*.

**Dataset Curation.** Dataset curation is typically one of the first steps to training foundation models. It is a critical mechanism for ensuring that harmful, inappropriate, or biased content is excluded from the training data. Birhane et al. [91] examine the toxic, offensive, and harmful contents in the LAION-400M dataset [774] and demonstrate the failure cases of CLIP filtering. Thiel [858] uncover instances of sexual abuse material within the LAION-5B dataset [775] and raise concerns about the reliability and safety of publicly sourced data. The work also discusses strategies based on the nearest neighboring for removing such harmful content. Hong et al. [361] audit common approaches of image-text CLIP-filtering and highlighted discrepancies in filtering techniques that could lead to biased annotations. Birhane et al. [92] investigate the effect of scaling datasets on harmful content and suggest developing new filtering methods for hateful and aggressive texts that traditional filtering cannot handle.

**Trigger Detection.** Trigger detection focuses on identifying malicious inputs before they can degrade the image generative models. [840] introduce DisDet to detect backdoor samples in unconditional diffusion models by analyzing the distribution discrepancy of the noise input. They propose using a KL divergence-based method to identify infected samples, achieving nearly 100% detection recall at a low computational cost. However, this method struggles with conditional diffusion models where backdoor attacks may not impact the noise input [182, 834, 1109]. To address this, Wang et al. [950] leverage the *assimilation* phenomenon of image generative models where a malicious trigger token can dominate the attention of the rest of tokens, which causes the model to focus on the trigger's intended effect rather than the prompt's original semantic meaning. The authors introduces a binary-search algorithm to localize such malicious triggers within a backdoor sample. Yoon et al. [1070] introduce a training-free safeguard approach for image generative generation designed to prevent inappropriate outputs from unsafe or adversarial input prompts by integrating filtering mechanisms across both text embeddings and visual latent spaces.

**Model Finetuning.** Model fine-tuning aims to defend image generative models from generating unsafe and unethical content via alignment [165]. Techniques such as *concept-erasing* [281, 469, 773] which change the weights of existing image generative models regarding malicious concepts and *inference guidance* [773] which

directly eliminates the capability of generating inappropriate content from image generative models, have been proposed to preventing harmful content generation under malicious inputs. Despite their potential, these methods face significant challenges: they are not comprehensive, lack scalability, and often degrade the quality of benign image generation [1129, 487, 773], which makes them rarely considered by image generative online services [636].

**Post-Generation Content Moderation.** Post-generation content moderation involves filtering and censoring generated images that violate safety or ethics criteria. These methods can be divided into *prompt-based* moderation, like OpenAI's Moderation API [678, 705], which prevents harmful content generation by identifying and rejecting malicious prompts, and *image-based* moderation, like safety checkers in SD [740], which operates on the generated images to detect and remove inappropriate elements. These methods do not interfere with the training process of the image generativemodel, preserving the quality of the generated images. However, they rely heavily on extensive labeled datasets and often struggle with generalizing to new types of inappropriate content or unseen attacks [1041, 773, 166]. To address the generalizability issue, Yang et al. [1042] proposes GuardT2I, which directly moderates the intermediate latent of textual prompts to be more robust and more generalizable to various inappropriate content. On the contrary, SafeGen [517] operates by regulating the vision-only self-attention layers to remove the generation capability of unsafe content from the image generative model in a text-agnostic way. Furthermore, Latent guard [565] proposes to learn a latent space on top of the image generative model's text encoder and detect the presence of harmful concepts in the input text embedding. Similarly, Chen et al. [170] introduces an agentic post-verification system that guardrails model outputs based on explicit safety regulations. In addition to safety filters, other mitigation strategies have also been studied [773, 502].

## 5.4 Current Limitations and Future Directions

Despite significant advancements in the domain of foundation model security, several limitations still require future attention for both attack and defense methods.

### 5.4.1 Limitations and Open Challenges of Attacks

Most current attack methods against foundation models rely on optimization techniques, whether white-box or black-box. Iterative white-box optimization is computationally intensive for foundation models, while black-box optimization incurs significant economic costs due to massive token consumption. These high costs may limit and discourage attackers from adopting these methods in real applications.

Another major limitation is the uncertain real-world threat of these attacks. Chen et al. [164] point out that most jailbreak attacks are only able to generate simple harmful sentences or paragraphs in most cases, lacking the ability to provide detailed instructions for malicious behaviors. This significantly limits the practical application scenarios for these attacks.

There are also multiple open challenges for attacks against foundation models. Foundation models have witnessed new applications and paradigms, including multi-agent communication [334, 693, 714, 362], tool usage [719, 344, 770], retrieval-augmented generation (RAG) [734, 46], making foundation models more widespread in different domains. Understanding the security of foundation models under these paradigms and building attacks for them is also an interesting problem with practical impact.

### 5.4.2 Limitations and Open Challenges of Defenses

Most current defense methods on foundation models lack formal safety guarantees in terms of definition and design, unlike traditional machine learning models and small deep neural networks that have provably secure defenses [195, 731]. A major reason is that for foundation models, an important attack space is prompts, which are natural language, and attacks on them are more challenging to formalize compared with images.

Additionally, current defense methods are significantly less effective when considering multiple modalities. This challenge is rooted in the fundamental differences between modalities, while current foundation models attempt to unify them into the same embedding space. This unification allows attackers to bypass defenses through any modality.

Another limitation worth mentioning is over-safety. Due to the difficulty of precisely defining safe/unsafe behaviors for foundation models, it is common for defense mechanisms to exhibit over-safety, rejecting benign inputs that are misclassified as malicious. Oversafety negatively impacts user experience and is a significant problem to address.

Finally, developing defense strategies that leverage external symbolic or classification systems remains a challenge. Anthropic's constitutional classifiers [787] apply a separate LLM-based safety filter trained on explicit constitutional rules to guard against jailbreaks and unsafe outputs. Other safety detectors such as ShieldLM [1134] and Adversarial Prompt Shield [440] also demonstrate customizable detection rules and explainable decisions. However, these systems either require high-volume adversarial training or rely on instructing foundation models (which may still face evasion and incur non-trivial inference overhead). The long-standing trade-off between adversarial safety and user experience is unresolved. Developing on-the-fly defenses that do not significantly compromise output quality remains an open challenge for foundation model defenses.

# 6 Privacy

The rapid expansion of foundation models has brought privacy concerns to the forefront. These models, often trained on vast amounts of data, can potentially expose sensitive information [320]. Recent privacy regulations such as GDPR [779] and CCPA [304] further limit the availability and use of private data. Addressing these privacy concerns and ensuring compliance with privacy regulations has led to the development of privacy-preserving machine learning (PPML) solutions. Recent efforts have focused on integrating anonymization mechanisms and creating innovative privacy-preserving methods for foundation models [592]. However, these approaches often address only specific aspects of privacy and may not provide a comprehensive solution. For instance, implementing differential privacy in foundation models can reduce model accuracy, while secure multi-party computation methods lead to high communication and computation overhead [249]. Advanced cryptosystems like fully homomorphic encryption offer strong privacy guarantees by enabling computations on encrypted data, but they come with prohibitive computational costs [7]. In this section, we provide a comprehensive examination of privacy in foundation models, as illustrated in Figure 14.

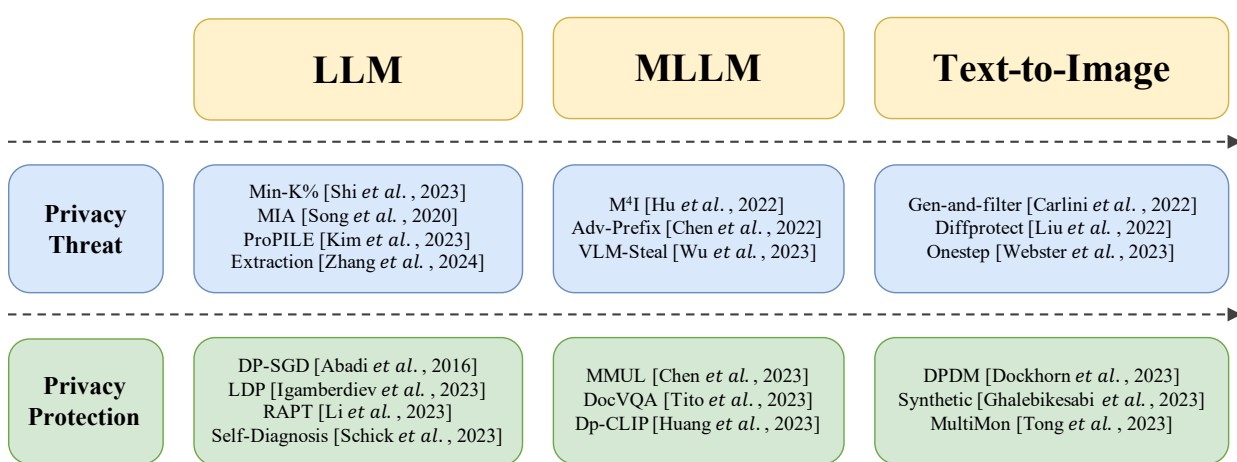

Figure 14: Privacy threat and protection techniques in different types of foundation models, including LLM, MLLM, and T2I models.

## 6.1 Privacy in LLMs

### 6.1.1 Privacy threats in LLMs

Like their traditional counterparts, foundation models tend to memorize training data, which frequently includes sensitive information. The issue of memorization is magnified in large foundation models due to their over-parameterization, a trait that becomes increasingly pronounced as the model's scale enlarges [125, 1040]. Consequently, this raises severe privacy concerns related to the use of LLMs [320].

Membership inference attack (MIA), a significant threat posed by LLMs, seeks to identify whether a particular data record was utilized during training. Song & Raghunathan [821] study membership inference against BERT models [219]. Mattern et al. [618] propose a neighborhood comparison method to improve the effectiveness of such attacks against LLMs. Recently, Shi et al. [801] introduce a reference-free MIA method, MIN-K% PROB, that determines if an LLM was trained on specific text based on the distribution of the log probability of each token without requiring knowledge of the pre-training dataset. Besides pre-training, Jagannatha et al. [404] show that membership inference could be performed on language models fine-tuned on medical data. Similarly, Mireshghallah et al. [644] highlight the vulnerability of LLMs to membership inference attacks during their fine-tuning phase. Wen et al. [969] studied membership inference attacks against in-context learning. Anderson et al. [28], Li et al. [524] studied membership inference attack against RAG. Feng & Tramèr [266], Wen et al. [973] further introduce privacy backdoor attacks that significantly increase privacy leakage during the fine-tuning phase.

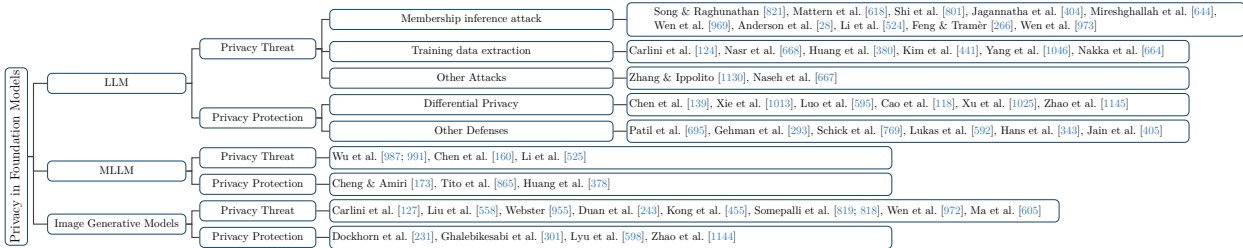

Figure 15: Taxonomy for Privacy of Foundation Models.

Moreover, the extraction of training data poses a significant risk to the privacy of LLMs due to their strong memorization capabilities. Carlini et al. [124] first successfully extract training data on GPT-2 models, revealing that the model could output sensitive information, such as phone numbers and email addresses when prompted with specific prefix patterns. Nasr et al. [668] further improve it by introducing a divergent attack on ChatGPT, which emits training data at a considerably higher rate. Further studies [380, 441] specifically focus on the extraction of personally identifiable information (PII) from LLMs. Besides natural language, Yang et al. [1046] explore data extraction in code LLMs, highlighting the broad applicability of these privacy concerns. Nakka et al. [664] explored enhancing PII extraction by grounding context similar to training data.

Additionally, there are other potential attack surfaces in LLMs. For instance, Zhang & Ippolito [1130] demonstrate that prompts, considered valuable commodities in the age of foundation models and tradable on markets, can be successfully uncovered by users even when they are intended to be kept confidential. Moreover, the process of tuning hyperparameters for LLM decoding algorithms, which demands significant time, manual effort, and computational resources, is compromised by Naseh et al. [667], who reveals a method to extract these hyperparameters at a very low cost.

### 6.1.2 Privacy-preserving techniques in LLMs

To mitigate these privacy threats, the field has shifted towards developing various techniques to preserve privacy in LLMs, aiming at fortifying these models against such vulnerabilities.

Differential Privacy (DP) is a rigorous mathematical framework that provides quantifiable privacy guarantees when analyzing or learning from sensitive data. At its core, DP ensures that the output of an algorithm (e.g., a trained model) does not significantly change when any single individual's data is added or removed from the dataset. This limits the risk of leaking information about any particular individual.

DP typically works by introducing random noise into the computation to obscure the contribution of individual data points. Differential Privacy Stochastic Gradient Descent (DP-SGD) [1], a foundational technology in many privacy-preserving LLMs, injects sample-wise Gaussian noise into the computed gradients during optimization. The key idea is that, even with full access to the trained model, an attacker cannot confidently infer whether any specific individual's data was used. This balance between utility and privacy is controlled by parameters $(\epsilon, \delta)$, where smaller values imply stronger privacy. Igamberdiev & Habernal [398] explore LLM pre-training under local differential privacy (LDP), aiming for privatized text rewriting. Given that LLMs are frequently fine-tuned on sensitive domains, numerous studies have investigated the application of DP-SGD in fine-tuning LLMs. Qu et al. [720] apply differential privacy on pre-training and fine-tuning BERT models. Yu et al. [1071] and Li et al. [518] study the integration of DP-SGD with different fine-tuning algorithms for GPT-2. Li et al. [519] propose differentially private prompt-tuning techniques for LLMs. Yue et al. [1087] apply DP-SGD for generating synthetic text that adheres to the post-processing theorem, therefore preserving the same privacy budget. These texts can serve as substitutes for original data in downstream tasks while maintaining privacy.

Aside from general DP-based defenses against various privacy attacks, there are targeted methods for specific threats, such as data extraction attacks. Patil et al. [695] investigate a defense by directly removing sensitive information from model weights. Moreover, techniques for filtering toxic output [293, 769] can help prevent

the generation of sensitive content. Lukas et al. [592] reduce the risk of PII leakage through PII scrubbing on the fine-tuning dataset. Hans et al. [343] propose a memorization mitigation strategy during pre-training, which involves randomly sampling a subset of tokens to exclude from the loss computation. Jain et al. [405] also find that adding noise to word embeddings during training can reduce the effectiveness of extraction attacks.

## 6.2 Privacy in MLLMs

Similarly, MLLMs also face privacy risks due to their tendency to memorize sensitive information from training data. Hu et al. [373] introduce both metric-based and feature-based attacks for conducting membership inference on multimodal models under various assumptions, highlighting the privacy vulnerabilities of MLLMs. Wu et al. [987] show that MLLMs are also susceptible to model stealing attacks, where model information of CLIP can be extracted via either the text-to-image or image-to-text retrieval APIs. Another privacy risk of MLLMs stems from their capability to extract sensitive information from images and present it in textual form. Wu et al. [991] observe that jailbreaking MLLMs could induce them to identify the real human, causing severe privacy concerns. Chen et al. [160] apply adversarial prefix instructions on MLLMs to expose private information within images. Their findings reveal that existing access control instructions fail to prevent MLLMs from answering personal data, violating the General Data Protection Regulation (GDPR). Li et al. [525] studied membership inference against MLLMs based on the token-level confidence of the model output from the cross-modal (text-image) data.

Recent research has been conducted to protect the privacy of MLLMs. Cheng & Amiri [173] develop a machine unlearning approach tailored for multimodal data and models, providing improved protection for erased data. Tito et al. [865] employ a combination of federated learning and differential privacy to secure the privacy of MLLMs, particularly in the context of Document Visual Question Answering. Huang et al. [378] introduce a differentially private variant of the CLIP model, effectively addressing privacy concerns while maintaining accuracy across a wide range of vision-language tasks. While these studies focus on protecting the privacy of training data, the challenge of mitigating the risk of MLLMs extracting sensitive information from input images remains an open problem.

## 6.3 Privacy in Text-to-Image Models

Since the introduction of text-to-image models, the research community [127, 558, 955, 243, 455, 819, 818, 972, 605] has uncovered hazards associated with extracting private information from public models. These studies demonstrate the possibility of extracting over a thousand training examples from state-of-the-art diffusion models, ranging from photographs of individuals to trademarked company logos, highlighting the urgent need to address these vulnerabilities to preserve privacy.

To mitigate these issues, the development of differentially private diffusion models [231] has been proposed, utilizing DP-SGD to enforce privacy. Ghalebikesabi et al. [301] explore the use of perturbation, timestep augmentation multiplicity, and modified timestep sampling schemes to train a more effective private diffusion model. Lyu et al. [598] further propose Differentially Private Latent Diffusion Models that only finetune the attention modules of diffusion models with privacy-sensitive data to obtain differentially private diffusion models in an efficient manner.

Beyond that, recent advancements in fine-tuning based text-to-image models, such as Textual Inversion [277], DreamBooth [758], and Custom Diffusion [470], have empowered individual users to incorporate personalized concepts into the base model with minimal data and computational resources. However, the increasing adoption of these models has sparked concerns regarding image privacy and copyright issues. For instance, fine-tuning specific face datasets enables text-to-image models to generate highly realistic images of individuals, which can lead to significant privacy violations and authenticity concerns. Similarly, fine-tuning the works of specific artists allows text-to-image models to replicate artistic styles with ease, potentially resulting in copyright infringement issues. These concerns surrounding image privacy and copyright in the context of text-to-image models have garnered attention from the public and media [75, 194, 952].

A number of research efforts have been dedicated to addressing the image privacy and copyright challenges posed by text-to-image models. A notable approach involves adding imperceptible protective adversarial perturbations to images, thereby preventing text-to-image models from learning the features of protected images [529, 893, 1146, 782, 979, 1060, 1143]. However, after fine-tuning on images with adversarial perturbations, the generated images by text-to-image models typically sacrifice quality and exhibit semantic deviations compared to those fine-tuned on unperturbed images. GrIDPure [1144], a simple yet efficient purification method, successfully eliminates protected adversarial perturbations while preserving their quality. GrIDPure claims they can effectively aid Stable Diffusion in learning from protected images, thereby highlighting the fragility and unreliability of the adversarial protection method.

## 6.4 Current Limitations and Future Directions

While significant advancements have been made in enhancing privacy for foundation models, numerous challenges remain that warrant further investigation in the future.

### 6.4.1 Limitations and Open Challenges of Privacy Attacks

The vast scale of foundation model training datasets blurs the boundary between member and non-member data [246]. Many non-member data points may naturally be very similar to some member data points. This makes effective membership inference attacks on foundation models challenging and prompts a reevaluation of the membership game.

While training data extraction attacks largely avoid membership ambiguity issues, their primary limitation is that the current schemes are only able to extract a small fraction of the training data. This constraint raises questions about their practical threat.

Moreover, the threat model of some privacy attacks is overly strong. For example, backdoor attacks [973] require poisoning or injecting triggers to amplify privacy leakage, which can only be performed in constrained scenarios such that the attacker possesses sufficient knowledge of the target model (e.g., the loss formulation of its original task).

In conclusion, building effective privacy attacks that work under weaker assumptions and more general scenarios is an open challenge. Another interesting future direction is contextualized privacy. Even with perfect sensitive data cleaning, personal information leakage can still occur in context. For instance, during multi-turn conversations with LLM-based chatbots, it may be possible to infer personal attributes based on the entire context, even if no part of the conversation contains private information.

### 6.4.2 Limitations and Open Challenges of Privacy Preserving techniques

Currently, Differential Privacy (DP) has become mainstream in protecting data privacy in foundation models. However, DP still faces two significant limitations:

1. DP provides worst-case privacy leakage bounds. In real scenarios, adversaries rarely have full control over the training data, resulting in a considerable gap between practical attacks and the worst-case probabilistic analysis of privacy leakage, according to DP.

2. Integrating DP into the foundation model fine-tuning still leads to significant performance degradation [1071, 518]. This utility deterioration weakens the motivation for DP-based fine-tuning.

Other approaches focus on protecting privacy by removing or obfuscating sensitive information from training data. However, such data-sanitization pipelines have also been shown to suffer from performance degradation on main tasks [389, 688].

In conclusion, designing strong privacy-preserving techniques for foundation models that balance privacy and performance remains an open challenge.

# 7 Hallucination

Hallucination refers to the phenomenon in which the model generates unfaithful information that is not fully grounded or verifiable, leading to actions misaligned with observable facts. Foundation models are often observed to generate hallucinatory responses in text [364, 520], image [536], video [187], and audio [320], imposing significant challenges in their real-world applications. Specifically, the model outputs incorrect or inaccurate statements, with a highly confident tone, that fails to align with the input or reflect the knowledge of the real world. Prior efforts by Huang et al. [383] offer a detailed taxonomy for detection and mitigation of the LLM hallucination, which authors define the phenomenon as the generation of plausible yet non-factual content caused by intrinsic (model-driven) or extrinsic (knowledge-driven) factors. Similarly, Sahoo et al. [762] categorize hallucinations across multiple modalities and characterize hallucinations as outputs that lack faithful grounding in input or real-world data. Aggregating these empirical descriptions of hallucination, we introduce a definition of the phenomenon from a more unified and probabilistic perspective for the general audience. With a target generative model $f$ and a verification predicate $v$ that checks the alignment of the output $y$ with external knowledge $x_{\text{external}}$ and/or the input context $x_{\text{input}}$, a model response $y = f(x; \theta)$ is hallucinatory if:

$$p_v(y|x_{\text{input}}, x_{\text{external}}, f) < \epsilon, \tag{13}$$

where $\epsilon$ is the probability threshold for hallucination detection.

Hallucination has troubled both researchers and users, negatively affecting various domains, e.g., natural language generation [1006], visual perception [201], and medical application [891]. This section offers an overview of hallucination in the context of foundation models, presenting definitions and categorizations, and discussing various methods for evaluating and mitigating hallucinations. Figure 17 showcases hallucinatory examples of both LLMs and MLLMs.

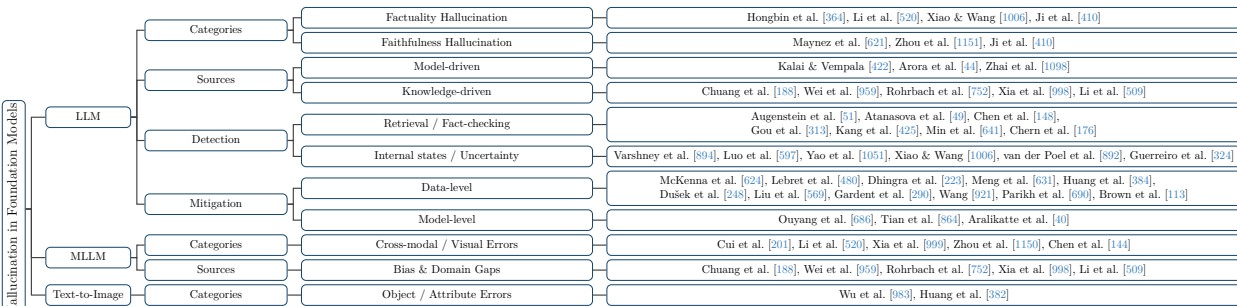

Figure 16: Taxonomy of Hallucination in Foundation Models.

## 7.1 Hallucination Categorization

### 7.1.1 Hallucinations in LLMs

Prior literature [621, 1151, 410] specifically refers hallucination in LLMs as the generation of nonfactual or unfaithful text to an input prompt. We categorize hallucination in LLMs into two types: factuality hallucination and faithfulness hallucination. Figure 17 depicts these two types of hallucinations. Detailed descriptions of the categorization are elaborated below:

- **Factuality Hallucination**. Existing LLMs sometimes generate text that conflicts with *world knowledge*, potentially resulting in misleading and raising concerns about the reliability of these models. These discrepancies and inconsistencies are termed *factuality hallucinations*. In general, we categorize them into two main types, depending on whether the information produced can be cross-checked with a credible source.

  - **Factual Inconsistency**. This term describes situations where the generated text from a language model contradicts established world knowledge. This kind of hallucination is the most common and

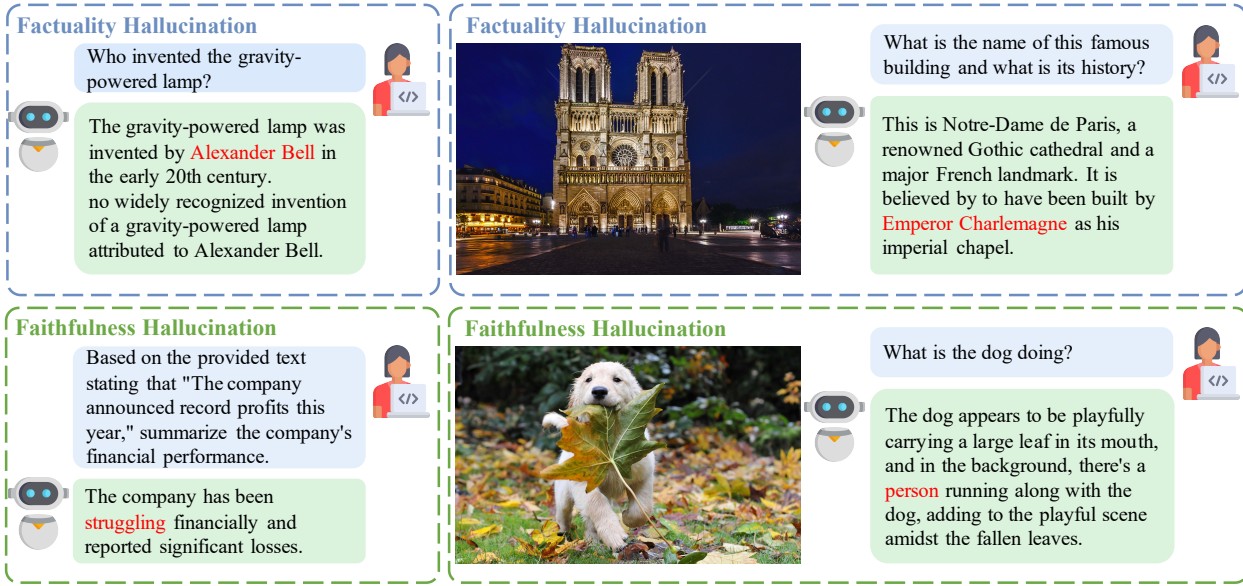

Figure 17: Examples of factuality and faithfulness hallucinations in foundation models.

arises from various factors, including how the language model captures, stores, and expresses factual knowledge.

– **Factual Fabrication**. This refers to cases where the language model generates a response that appears factual but cannot be verified by known real-world evidence.

- **Faithfulness Hallucination**. LLMs may generate text that conflicts with (1) input instructions, (2) in-context demonstration, or (3) preceding generated text. Such discrepancies can significantly impair the model reliability and the user experience. For example, Bang et al. [61] show that, when instructing an LLM to summarize an article at a given location, the LLM can be hallucinatory to summarize another text chunk instead. To better understand the issues, we classify faithfulness hallucinations into three categories based on the type of conflict and the specific aspect they pertain to:

  – **Input Confliction**. It refers to instances where the generated text contradicts the instruction or query in the input text. It indicates a failure to accurately process or explicitly follow task directives.

  – **Context Fabrication**. It involves cases where the generated content is beyond the specified scope by the in-context demonstrations from the source prompt or externally retrieved knowledge.

  – **Preceding Generation Inconsistency**. This pertains to situations where the text newly generated by the model conflicts with its earlier outcomes, leading to a lack of coherence and continuity in the entire output.

### 7.1.2 Hallucinations in MLLMs

Similarly, MLLMs are notably prone to generating hallucinatory responses, further compounded by issues specific to the vision modality. Figure 18 provides an overview of hallucinations in MLLMs.

MLLMs have become pivotal in bridging computer vision and natural language processing, offering a spectrum of applications through their ability to produce text descriptions that are contextually appropriate based on visual inputs. Despite their capabilities, MLLMs also encounter challenges with hallucinations. However, the taxonomy of hallucinations in MLLMs is specifically characterized by factuality and faithfulness [621, 1151, 410, 999], which differs from that of LLMs. This fine-grained classification aims to elucidate the factors influencing hallucinations in MLLMs, thereby revealing their underlying essence. The categories are detailed below:

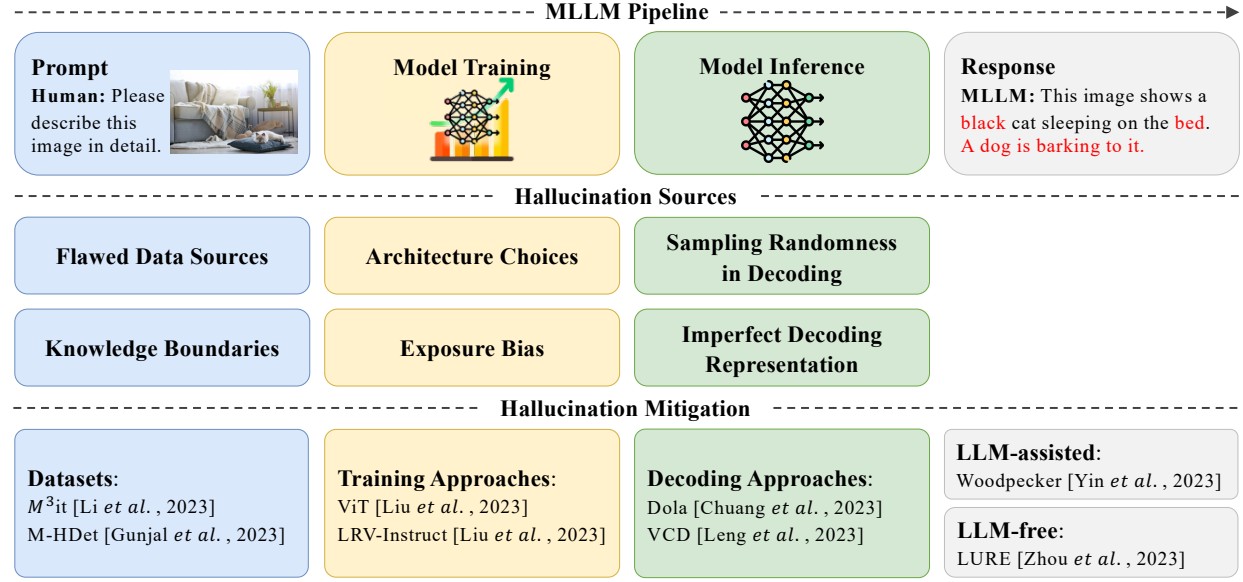

Figure 18: An overview of hallucinations in MLLMs. The top row illustrates a typical MLLM pipeline from user prompt to model response. Hallucinations can emerge at different stages due to multiple sources. To mitigate these hallucinations, various strategies are employed, including improvements to data curation, training methods, and inference approaches.

- **Factuality Hallucination.** In vision-language tasks, factuality hallucination occurs when the model, prompted by both textual and visual inputs, produces outputs inconsistent with real-world facts. Based on this difference, we further divide factuality hallucination in MLLMs into:
  - **Intrinsic Factuality Hallucination.** This refers to situations where MLLM outputs inaccurately describe images with facts that either contradict or cannot be verified against established real-world knowledge. This type of hallucination may stem from many sources, such as out-of-distribution world knowledge [1150] and insufficient visual ability [144].
  - **Extrinsic Factuality Hallucination.** MLLMs may generate the correct description of an image but produce information that contradicts or remains unverifiable in real-world knowledge. This kind of hallucination results in the creation of non-factual information, arising not from a misinterpretation of images but from other factors.
- **Faithfulness Hallucination.** From a visual-conceptual perspective, MLLMs may generate unfaithful or inaccurate outputs in response to a user-provided image. Considering the elements present in the image, faithfulness hallucination can be categorized into two main types:
  - **Object Inconsistency.** The model generates a description or explanation for an image, incorporating objects or features that are either missing or do not actually exist.
  - **Logical Hallucination.** The model generates a description or explanation for an image that includes missing or non-existent logical relationships, attributes, or quantity.

## 7.2   Sources of Hallucinations

This section discusses the various sources of hallucinations in LLMs and MLLMs, examining issues related to flawed data sources, training, and inference processes.

- **Hallucination from Data**
  - **Flawed Data Sources.** LLMs suffer from biased or incorrect textual data, and MLLMs can incorporate erroneous visual data. They also learn stereotypes from pre-trained corpora of world knowledge. Mislabeled textual corpora and visual data, imbalanced distribution in datasets (e.g., underrepresenta-

tion of certain demographics), and outdated information can lead to inaccuracies in classification and description tasks [188, 959]. For example, hallucinations in MLLMs are often observed in which the model predicts possible objects or actions that are not supported by the image but are plausible from commonsense [752].

– **Knowledge Boundaries.** LLMs and MLLMs face limitations in domain-specific knowledge (e.g., medical analysis in 998 and embodied decision-making in 509) or struggle to recognize and interpret up-to-date content. For instance, MLLMs may fail to accurately recognize images of new electronic devices, enterprise logos, or cultural symbols that are absent from their training data.

- **Hallucinations from Training**
  - **Architecture Choices.** The architectural design of MLLMs, often based on complex neural networks that integrate features of both vision and language models, might contribute to hallucinations. For instance, inappropriate conditioning of the visual and textual components leads to model misinterpretations [144].
  - **Inherent Mechanism.** There is an inherent statistical lower-bound on the hallucination rate of language models given the sufficient presence of facts for training transformers [422]. In other words, a certain level of hallucination is necessary for the model to minimize cross-entropy across large and varied pre-training data.
  - **Exposure Bias.** This issue may arise when the model is overtrained on certain types of images or texts, leading to an overrepresentation of these elements in the model's outputs, regardless of their relevance or accuracy in new contexts [44]. This is often the result of catastrophic forgetting [1098].

- **Hallucinations from Inference**
  - **Sampling Randomness in Decoding.** LLMs and MLLMs often use stochastic methods to generate outputs to textual and visual content. This randomness can lead to inaccuracies, particularly when dealing with complex tasks, ambiguous instructions, or intricate visual scenes.
  - **Imperfect Decoding Representation.** Challenges to adequately represent visual and textual information in the model can result in inaccuracies when generating descriptions or interpretations of visual data [872, 138].
  - **Path Dependence.** Autoregressive LLMs and MLLMs suffer from error propagation and self-consistency failures due to their token-by-token prediction nature. Any intermediate errors from previously generated tokens will have a compounding effect on the subsequently predicted token. Empirical studies have demonstrated that, assuming a constant per-token error rate, the generation quality can deteriorate rapidly with respect to the sequence length [42]. Additionally, inconsistencies with earlier generated content are common when hallucinations occur [953].

### 7.3 Hallucination Detection and Measurement

As shown in Figure 19, the process of detecting hallucinations in foundation models generally involves three key steps: first, breaking the response into distinct parts; second, extracting the facts from each part; and third, assigning a score to each fact. Next, we will provide a detailed discussion of hallucination detection and measurement methods across various types of foundation models.

#### 7.3.1 Detecting Language Hallucinations

Identifying hallucinations in LLMs is crucial to maintaining the reliability of their outputs. Traditional metrics, mainly based on word overlap matching, are inadequate to distinguish subtle differences between plausible content and hallucinations. This emphasizes the need for advanced detection techniques specifically designed for LLM hallucinations. Here, we discuss three distinct pipelines, differentiated by their use of external knowledge and evaluation metrics.

**Retrieval-based Detection.** To accurately pinpoint factual inaccuracies in LLM outputs, a straightforward approach is to compare the generated content with reliable knowledge sources. This idea is similar to the traditional fact-checking process [51, 49]. Building on this concept, a line of works has employed LLMs to verify generated claims using evidence retrieved from external sources [148, 313, 425, 641, 176]. This

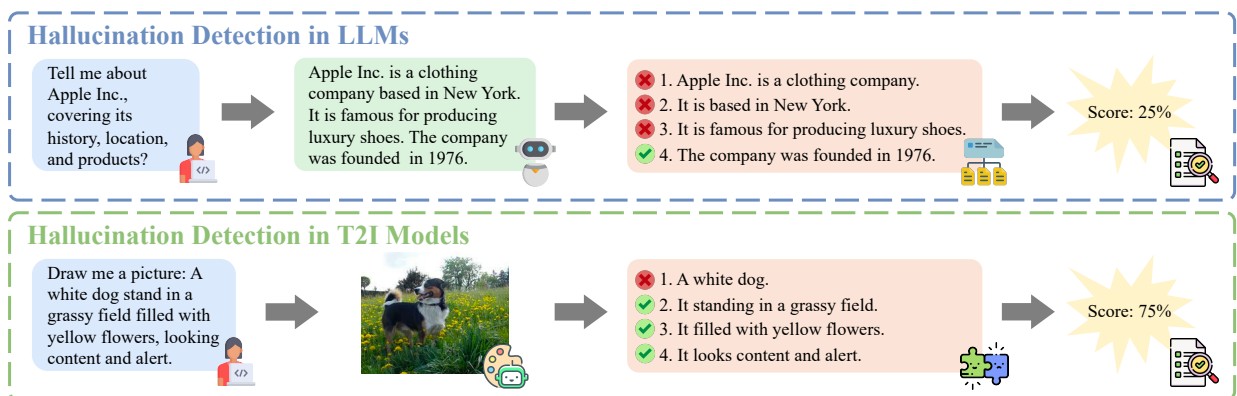

Figure 19: Illustration of hallucination detection in foundation models. The process generally involves three steps: (1) decomposing the model output into individual factual units, (2) verifying each fact against external knowledge or ground truth, and (3) aggregating the correctness scores to produce an overall hallucination score. The examples show applications in both LLMs and T2I models.

method generally encompasses three essential phases: extracting the claims, retrieving relevant evidence from external knowledge, and classifying the veracity of the claim. The detailed implementation of each stage varies in different frameworks. In the context of long-form generation, tools such as FActScore [641], Core [415], and FacTool [176] prompt language models such as InstructGPT [686] or ChatGPT to extract all claims in a target text. Similarly, EVER [425] is designed to extract all fact-related concepts from a sentence and subsequently generate a yes/no validation question for each concept in the generated text. For evidence retrieval, FActScore [641] picks the most relevant pieces of facts from the external knowledge base, where the relevance is ranked by retriever embedding distances. In addition, Chen et al. [148] use a language model to summarize the retrieved texts, specifically serving as supporting evidence for the claims. EVER [425] and FacTool [176] utilize a search engine to acquire evidence from queries generated by a language model. These methods can be further classified into two types based on their ability to detect factual fabrication, which involves generating information that cannot be verified. The first category employs a language model to directly label claims as True or False [641, 176], while the second category additionally considers the possibility of Not-Enough-Information (NEI).

**Model Internal State.** A series of studies have explored the use of LLM internal states for hallucination detection. Varshney et al. [894] assess hallucinations by examining the lowest token probability in key concepts, suggesting that a lower probability indicates a higher likelihood of hallucination. Luo et al. [597] implement a self-assessment method, where the proficiency of an LLM is judged by its ability to reconstruct a concept from self-explanation. This approach uses the perplexity of the generated response as a measure to gauge the level of concept understanding. Yao et al. [1051] explore hallucinations as a form of adversarial attack by using gradient-based token replacement, finding that the initial tokens generated from standard prompts have a lower entropy compared to adversarial ones, leading them to decide an entropy threshold to detect hallucinations.

**Uncertainty Estimation.** Recent advancements in LLM research have seen a focus on uncertainty estimation. We will briefly discuss this in the context of hallucination evaluation and provide a more general explanation in Section 8. Built upon techniques such as deep ensembles and conditional entropy, entropy-based methods [1006, 892, 265] establish a connection between hallucination likelihood and predictive uncertainty. Guerreiro et al. [324] explore variance estimation through Monte Carlo Dropout [278] and investigate a log-probability-based approach by measuring model confidence through length-normalized sequence log-probability. Additionally, some methods employ LLMs themselves to estimate uncertainty and detect hallucinations. SelfCheck [635] detect errors in complex reasoning within LLMs. The burgeoning field of prompting-based metrics has also gained attention, with Chiang & Lee [177] leveraging the instruction-following capability of LLMs to evaluate the faithfulness of content. Beyond these methods, researchers

have also gleaned insights from LLM behaviors. Natural language prompts [1015] have played a pivotal role in this direction. Manakul et al. [613] tackle this issue by examining the consistency of factual statements across multiple LLM responses. Another innovative approach [10] uses indirect queries to subtly elicit specific information. This approach mirrors investigative interview techniques, offering a nuanced evaluation of consistency. Taking a step further, the multi-agent perspective, particularly exemplified by LMvLM [196], involves one LLM as the examiner to question another, the examinee. This method, inspired by legal cross-examination techniques, aims to reveal inconsistencies in multi-turn interactions. Collectively, these diverse approaches contribute to a deeper understanding of uncertainty in language models and provide innovative ways to measure it.

### 7.3.2 Measuring Multimodal Hallucinations

Conventional statistical metrics, such as BLEU [689], CIDEr [896], and ROUGE [537], are commonly used to evaluate hallucinations. However, it has been suggested that these metrics might not be highly appropriate for assessing detailed descriptions from MLLMs [1169]. To address this issue, recent metrics [752, 520, 924, 1021] shift their focus towards object hallucination, as objects represent fundamental components that compose the visual scene and contribute to correct visual understanding. In addition, Li et al. [520] propose Polling-Based Object Probing Evaluation (POPE), a novel approach that utilizes polling-based queries to prompt MLLMs [321] with straightforward yes-or-no questions regarding the existence of specific objects in an image. By asking targeted questions, POPE provides a more stable and flexible evaluation of object hallucination.

While recent evaluations have introduced numerous metrics focused on the phenomenon of object hallucination, it is crucial to consider their suitability for capturing the intricacies of MLLMs, as many elements contribute to the visual semantics of an image. Apart from evaluation metrics that focus only on object hallucinations, Wang et al. [924] introduce HaELM, a framework of hallucination evaluation by fine-tuned LLaMA [14]. The work of Gunjal et al. [327] presents a novel dataset, M-HalDetect, designed specifically to identify hallucinations in visual question answering (VQA) tasks. Wang et al. [937] introduce an innovative dataset named Mementos, aimed at assessing the reasoning abilities of MLLMs in understanding sequences of images. Additionally, they propose a unique type of hallucination, termed behavior hallucination, that arises specifically in the context of image sequence comprehension. Additionally, Wang et al. [923] propose an LLM-free method to evaluate object existence, object attribute, and object relation hallucinations cost-effectively and efficiently. Although Retrieval-Augmented Generation (RAG) [494] is a common technique to alleviate the hallucination problem, there are no sufficient guarantees that the problem is fully eliminated. To address this scenario, Wu et al. [990] curate a benchmark of hallucination detection methods in the RAG-based system.

### 7.3.3 Measuring Text-to-image Hallucinations

Recent studies [983, 382] have evaluated the compositional ability of text-to-image models, revealing their tendency to hallucinate. These benchmarks prompt text-to-image models with text descriptions that contain multiple concepts. Next, they analyze how many of these concepts actually appear in the generated image by creating one question per visual concept and using a strong MLLM to answer all questions. This process identifies hallucinations that occur when the model misses some concepts or introduces concepts that were not present in the original description.

## 7.4 Hallucination Mitigation

### 7.4.1 Reducing Hallucinations in LLMs

There is a pressing need for the development of novel and reliable methods to mitigate hallucinations in LLMs, particularly to meet the demands in real-world applications and substantially improve their generalization. Our review categorizes them into two primary types: data-centric and model-based approaches.

**Data-Centric Methods.** The training dataset plays a crucial role in the occurrence of hallucinations in LLMs. McKenna et al. [624] indicate that the primary cause of hallucinations is the memorization of training data. This issue manifests when user queries closely align with pre-trained data, often resulting

in LLMs generating inaccurate or misleading responses. Additionally, some datasets exhibit inconsistencies in the factual alignment between the input text and the reference target [410]. This problem is evident in dataset constructions [480, 223], and downstream tasks such as open domain dialogues [631, 384]. Besides this, many pre-trained LLMs have a fixed parametric knowledge base on a specific timestamp, and they are often deployed offline (e.g., limited access to the internet or data storage firewall for privacy). This limitation can potentially lead to hallucinations when the model encounters questions that its knowledge base does not cover [52]. Therefore, the data-centric approaches that can benefit the language model's (parametric and external) knowledge bases are practically reasonable. To further reduce hallucinations in natural language generation (NLG) tasks, numerous attempts have been made to automatically [248, 569] and manually [290, 921, 690] cleanse and refine the training sets for improving accuracy and reliability. In recent years, the importance of meticulous data filtering has become evident in effectively mitigating hallucination issues in LLMs. For instance, GPT-3's pre-training data [113] is cleaned using several high-quality reference corpora to ensure its quality. Touvron et al. [876] improve the quality of the pre-training corpus by incorporating data from reliable external sources such as Wikipedia. This process involves carefully selecting and up-sampling data from these sources to enrich the content and enhance the faithfulness of training data.

**Model-based Methods.** The choice of training strategies and model architectures also plays a crucial role in mitigating hallucinations. Past works [864, 40] have showcased a correlation between deficient comprehension capability in the encoder and the frequency of hallucinations. Recently, McKenna et al. [624] conducted a study on factors influencing hallucination and revealed that the main determinant is the extent to which training data are memorized. Specifically, models tend to prioritize generating outputs based on their parametric (internal) knowledge, rather than relying on the external information from the input prompt. This memorization, known as parametric knowledge bias [610, 410], significantly contributes to the occurrence of hallucinations. Numerous techniques [485, 621, 250, 805] have been developed to address this bias in earlier, smaller-scale language models. Some retrieval-augmented methods improve language models by incorporating external information from large corpora. Borgeaud et al. [103] introduce RETRO (Retrieval-Enhanced Transformer), an auto-regressive language model that leverages document chunks from a massive 2 trillion token database. Peng et al. [699] further advance a black-box LLM by incorporating external knowledge, therefore improving the model's response quality through iterative prompt revision. To address the challenges of factually inaccurate text generation and source attribution in language models, Ram et al. [734] propose a novel approach called In-Context RALM (Retrieval-Augmented Language Modeling). This approach maintains the original model architecture while prepending grounding documents from a corpus to the input. However, the development of model-based strategies for reducing hallucinations in LLMs is still at an early stage in the academic community. Recent research [657] develops an iterative algorithm to prompt LLMs to identify and eliminate self-contradictions in their generated text, thereby enhancing both fluency and informativeness. Chen et al. [137] use LLMs to corrupt text and fine-tune compact editors to denoise these faux hallucinations. Models fine-tuned with carefully crafted tasks and datasets can also mitigate hallucinations. Elaraby et al. [256] introduce a lightweight and knowledge-free framework for this fine-tuning process, with the aim of reducing hallucinations in open-source language models. Current state-of-the-art LLMs, i.e., ChatGPT [675], GPT-4 [676], and Llama 3 [14], have made significant advances in reducing hallucinations during the RLHF stage. It is also worth noting that artificially created hallucination data are leveraged to train the reward model for GPT-4 [676].

### 7.4.2 Reducing Hallucinations in MLLMs

Prior studies have attempted to address the problem of hallucinations in generating responses from images using small-scale multimodal models [1124, 93, 203, 439]. However, these approaches primarily focus on short image captioning tasks, making them inadequate for applications in MLLMs that aim to provide comprehensive and detailed descriptions. In recent advances, methods for addressing hallucinations in MLLMs can be mainly divided into three categories: training approaches, post-processing techniques, and decoding methods.

**Training Approaches.** Visual instruction-tuning [556] has significantly improved the zero-shot capabilities of MLLMs on new tasks. Built upon this, Liu et al. [553] introduce the Large-scale Robust Visual (LRV)-

Instruction dataset and the corresponding LRV-Instruction framework. The framework incorporates both positive and negative instructions to enhance the robustness of visual instruction-tuning and reduce hallucinations. Furthermore, recent studies introduce several new instruction-tuning datasets aimed at reducing hallucinations. For example, Li et al. [508] present the Multi-Modal Multilingual Instruction Tuning (M$^3$IT) dataset. The M$^3$IT dataset consists of 40 meticulously curated datasets, comprising 2.4 million instances and 400 manually crafted task instructions that have been reformulated into a vision-to-text format. Gunjal et al. [327] introduce the M-HalDetect dataset, designed to detect and prevent objects and logical hallucinations in detailed image descriptions. The authors train a fine-grained multimodal reward model using Instruct-BLIP and evaluate its effectiveness using best-of-n rejection sampling, which is a technique used in LLMs to improve output quality by generating multiple candidate outputs and selecting the best one according to a predefined scoring function. Since common practice in instruction-tuning dataset construction implicitly teaches an LLM to answer all questions, even for problems it cannot answer, this will lead to the tendency of LLM to hallucinate. To address this problem, Zhang et al. [1105] proposed a different approach, which enables LLMs to refuse to answer questions beyond their capabilities. LLMs with this ability were shown to effectively reduce hallucination by refusing to respond to particularly challenging questions.

**Post-processing Approaches.** These approaches in MLLMs refer to the techniques used to refine or alter MLLM outputs after the inference process. These techniques aim to improve the quality, accuracy, and usability of models' predictions. In the context of small-scale multimodal models, Ngo et al. [670] introduce pseudo labeling and efficient post-processing techniques to improve vehicle retrieval accuracy. Hoxha et al. [369] introduce two post-processing strategies based on hidden Markov models and the Viterbi algorithm, which improve the faithfulness of image captioning by leveraging probabilistic modeling of language structure to correct and optimize generated captions. These strategies effectively correct hallucinations in generated descriptions while improving the overall coherence of the captions. Duan et al. [245] utilize an auxiliary model to first learn the truthful direction of MLLMs decoding and then apply truthful-guided inference-time intervention during decoding to alleviate hallucinations. In the broader context of MLLMs, post-processing approaches can be mainly categorized into two types:

- **LLM-assisted Approaches.** These methods involve the use of foundation models, such as GPT, to correct and refine the outputs of smaller multimodal models. Maaz et al. [606] implement a GPT-assisted mechanism that refines and optimizes enriched annotations, thereby generating high-quality instruction data. Similar to small models, Yin et al. [1063] focus on improving the reliability of visual hallucination diagnosis, in which GPT is leveraged to rectify inaccurate diagnosis. In addition, Zhou et al. [1169] utilize GPT to correct hallucinatory captions for MLLMs.
- **LLM-free Approaches.** These methods do not rely on LLMs. Instead, they may engage in self-correction or use a smaller model as a revisor. Zhou et al. [1169] focus on three key factors contributing to object hallucination: co-occurrence, uncertainty, and object position. Based on these insights, they propose LMM Hallucination Revisor (LURE) to address the problem of object hallucination in MLLMs. Drawing inspiration from LM-Switch, Zhai et al. [1096] introduce a control parameter, referred to as a "switching value", to manage hallucinations in language generation by modifying word embeddings.

**Decoding Approaches.** In recent years, various methods have employed decoding approaches to address the issue of hallucinations in MLLMs during inference. Chuang et al. [189] reduce incorrect fact generation in LLMs through the contrast of logit differences between the later and earlier layers, and this approach can be extended to MLLMs. Moreover, recent efforts have been dedicated to mitigating hallucinations in MLLMs as well. Leng et al. [491] propose a training-free decoding approach called Visual Contrastive Decoding, which contrasts the output distributions with the original and distorted visual inputs. This approach effectively calibrates the model's over-reliance on unimodal priors and statistical bias without utilizing external models. Similarly, Chen et al. [168] contrast the logit distribution of different partial visual inputs to approximate the optimal visual context during decoding. Besides, Huang et al. [385] present a decoding approach based on an over-trust penalty and a retrospection-allocation strategy to alleviate the problem of partial over-trust. Building on these decoding approaches, Li et al. [526] introduce VISTA, a training-free inference-time intervention that mitigates hallucinations in MLLMs by reinforcing visual information in activation space and leveraging early-layer activations, achieving consistent improvements across models and decoding strategies.

### 7.5 Current Limitations and Future Directions

While recent research advances have partially defined the concept, developed measurements, revealed causes, and proposed solutions for hallucinations in foundation models, it remains a challenge to formulate the phenomenon comprehensively. In the sections above, our survey summarizes the existing work and presents them in our paradigm. Next, we will discuss current challenges and shed light on potential future directions that the community should focus on.

When validating whether a language model response is hallucinatory, a popular line of research is to retrieve external texts (e.g., scientific facts and historical archival) as supportive evidence. This approach requires a highly trusted knowledge database that is often crowd-sourced and is presumed to be factual. However, there are no rigorous guarantees regarding the trustworthiness of these external knowledge bases, and their vast size makes manual scrutiny impractical. Furthermore, after evidence retrieval, the current pipeline typically employs a verifier to compare the retrieved fact with the target model's statement for hallucination classification. If the language model is prompted to execute this validation, it is important to note that the verifier may also exhibit hallucinatory behavior. Additionally, the statistical correlation between the mechanisms of the autoregressive hidden states and the hallucination phenomenon is still unclear. Lastly, while most existing research focuses on validating the language generation from LLMs and MLLMs, research is insufficient in the synthesis of other modalities (e.g., images, videos, and audio).

To address the limitations outlined in current research on hallucinations in foundation models, several avenues for future work emerge. Firstly, enhancing the reliability and veracity of the knowledge bases used for evidence retrieval is essential. Future studies shall focus on the development of filtering and verification systems for external knowledge. Secondly, it is crucial to mitigate the risk of hallucinations in validation models. Potential frameworks include multi-agent collaborations where multiple independent models interact and cross-verify the target output. Furthermore, a deeper understanding of the underlying mechanisms of language models, particularly the statistical properties and dynamics of autoregressive hidden states, will help mitigate hallucinations. Techniques like mechanistic model editing and representational latent editing hold the potential to promote more faithful language generation. Incorporating refusal mechanisms is a rising topic to study knowledge boundaries and mitigate model hallucinations [1018, 1180]. Future work shall consider directions for improving refusal calibration, integrating refusal with multi-agent and retrieval-based pipelines, and extending refusal capabilities to multimodal generation tasks beyond text. Lastly, since image diffusion models [753] and visual autoregressive models [1073, 512] are increasingly conditioned on textual prompts, the community should expand the scope of hallucination research to the generation of images, videos, and audio.

# 8 Uncertainty

Though modern foundation models possess impressive capabilities across a wide range of domains and tasks, harnessing their power reliably requires a clear understanding of the uncertainties inherent in their outputs.

In the context of foundation models, uncertainty refers to the degree of confidence in the model's predictions or generated content. This uncertainty can arise from multiple sources: epistemic uncertainty, which is due to limited or imperfect knowledge encoded during training (e.g., gaps in the data or model misspecification), and aleatoric uncertainty, which stems from inherent randomness or ambiguity in the data itself (e.g., inherently noisy or ambiguous task definitions). However, it is important to note that this distinction also has limitations [54, 318, 887, 654].

Beyond these, real-world deployment introduces further uncertainty due to distribution shifts, novel scenarios not represented during training, or unforeseen user inputs. Recognizing and managing these different types of uncertainty is essential for responsible and safe use of foundation models in high-stakes applications.

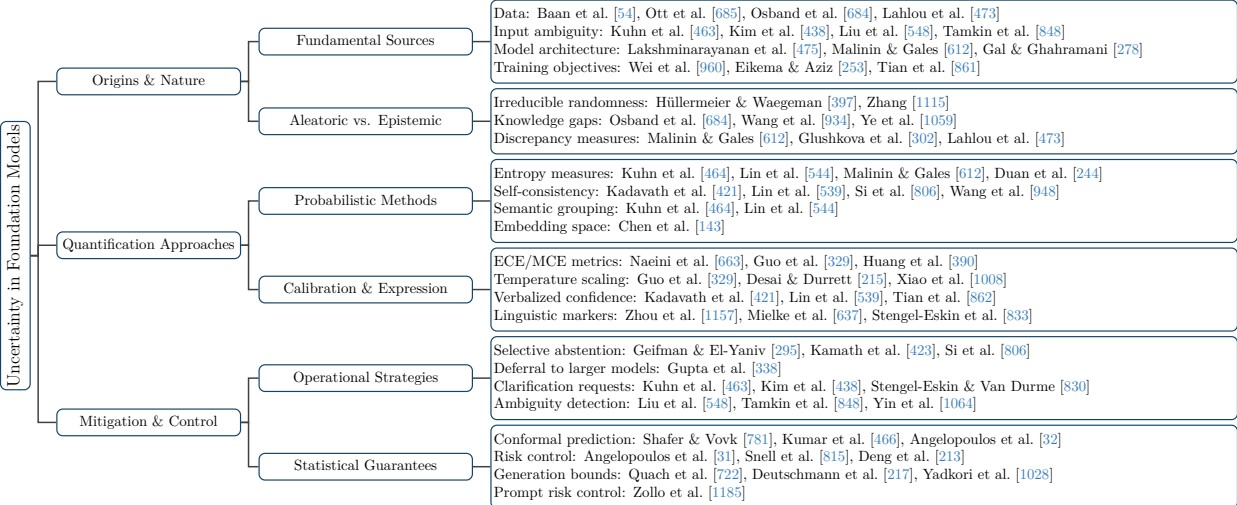

Figure 20: Taxonomy of Uncertainty in Foundation Models.

## 8.1 Sources of Uncertainty

Uncertainty in foundation models can be categorized into aleatoric and epistemic types. Aleatoric uncertainty is primarily influenced by data factors, while epistemic uncertainty is largely affected by modeling decisions. We will discuss the impact of these components in both the training and inference stages, with a comparison presented in Figure 21.

### 8.1.1 Data

To begin, we will consider how the nature of language itself introduces uncertainty into the generation process of LLMs [54, 685], which both consume natural language as input data and produce it as output data. Given some input context to a language model, there are usually many possible responses for several reasons. First, the input may be reasonably interpreted to have multiple different meanings. This could be because the context is vague ("She watched the man with binoculars"), very complex (as some reading comprehension questions are, even for humans), or contains spelling or other errors. Besides ambiguity in the input, certain queries may be inherently more open-ended and allow for many reasonable responses. This might include a request to complete a fictional story, tell a joke, or give a position on some political or social issue. Finally, given a fixed input interpretation, equivalent answers to a query may be expressible in many ways. For example, given the input context "What is the capital of Rwanda?", "Kigali is the capital of Rwanda" and "The capital of Rwanda is Kigali" offer semantically equivalent answers with different surface forms.

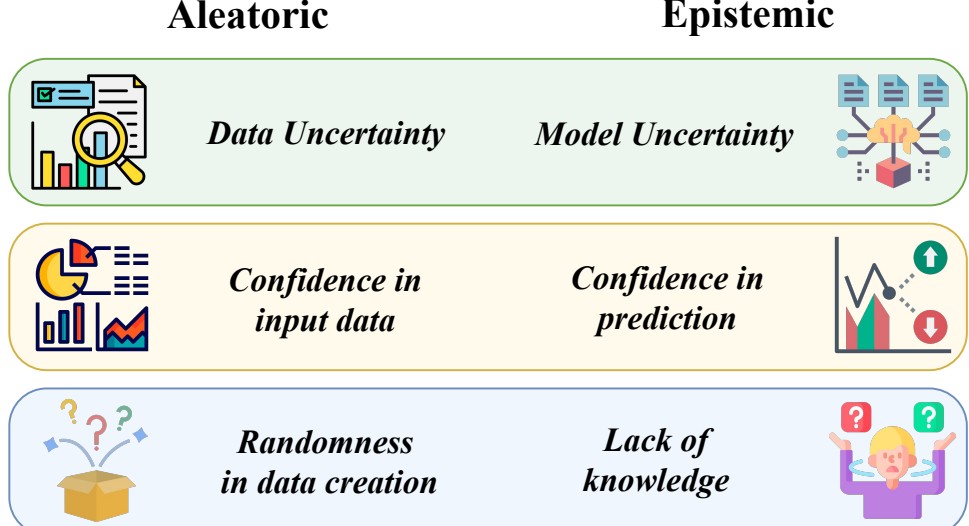

Figure 21: The comparison of aleatoric and epistemic uncertainty in machine learning. Aleatoric uncertainty originates from data variability, while epistemic uncertainty results from model limitations. The former is tied to inherent noise in observations, while the latter is tied to insufficient knowledge in representation.

The uncertainty in an LLM generation due to natural language data stems both from the training data and the prompt inputted to the model during inference. When the training dataset is small or not sufficiently general, the model may not have the relevant knowledge to effectively process some context [684, 397, 473, 697]. If the training data contains a large amount of ambiguous language, the trained LLM may reflect this uncertainty in its outputs. Other sources of uncertainty introduced by training data include diverse, conflicting, and outdated information.

In deployment, further uncertainty is introduced by specific text data given as input to the LLM. Queries could be ambiguous [463, 438, 548], and tasks or instructions could be open-ended or underspecified [848], making it difficult for the model to express an appropriate level of confidence in its response. Also, relevant information could be excluded from the context [1081], or users may produce input errors.

### 8.1.2 Model

In addition to the uncertainty introduced by data in training and inference, the model itself also contributes to the uncertainty in the generation process, given an input prompt. Architecture choices may not reflect the underlying data-generating process. Different modeling techniques like ensembling [475, 612, 302, 936] or Bayesian inference [278, 685, 1006] can be applied with the hope of accurately characterizing the true posterior probability. However, these methods can be computationally expensive and potentially ineffective [4, 687]. Besides architecture, the typical optimization objective of producing the most plausible answer (i.e., maximizing observed sequence probability 253, 252) may not align to produce the most correct and factual answer [861], and in general, the cross-entropy objective has been shown to lead to overconfidence [960]. Finally, though massive pre-trained models are an effective tool for combating the uncertainty introduced by the input context, the popular approach of fine-tuning these LLMs for custom use cases may dilute these generalist capabilities [1083].

### 8.1.3 Aleatoric vs. Epistemic Uncertainty

Besides identifying how uncertainty may arise due to data and model factors, it may also be useful to characterize uncertainty in LLM responses as either *aleatoric* or *epistemic* [397, 1115]. Aleatoric uncertainty, sometimes referred to as data uncertainty, exists due to the inherent randomness in the data-generating

process. Additional information cannot be used to reduce aleatoric uncertainty. For example, suppose a language model was asked to predict the probability of heads on the flip of a fair coin. In that case, no additional context or training data would enable a better prediction than 50%. On the other hand, epistemic uncertainty arises precisely because of a lack of knowledge. Epistemic uncertainty may be reduced by incorporating additional data, for instance, by including or prioritizing more informative examples in the training set [684, 934] or incorporating appropriate few-shot examples in the context [1059, 225, 516, 835, 1081]. Section 8.2.1 discusses more of this work in detail.

## 8.2 Quantifying and Addressing Uncertainty

While some uncertainty in LLM responses is unavoidable, considerable progress has been made in quantifying and addressing this uncertainty so that models can be deployed responsibly and reliably. Common measures for quantifying uncertainty use notions such as entropy to characterize the uncertainty in response and produce higher scores for outputs that are less likely to be correct. Methods have been developed to recalibrate confidence scores so that they better reflect the true probability of an answer being correct. Additionally, uncertainty estimates can be used to identify cases where the system should abstain from answering or seek further clarification before providing a response. Researchers have also examined whether LLMs can generate linguistic expressions indicating their uncertainty for a given output [685, 806, 464]. Finally, rigorous statistical methods, which are quickly gaining popularity in the deep learning community, have been applied to provide high probability bounds on LLM performance and risk [890, 836, 741]. Next, we will highlight important work in these areas, accompanied by an illustration in Figure 22.

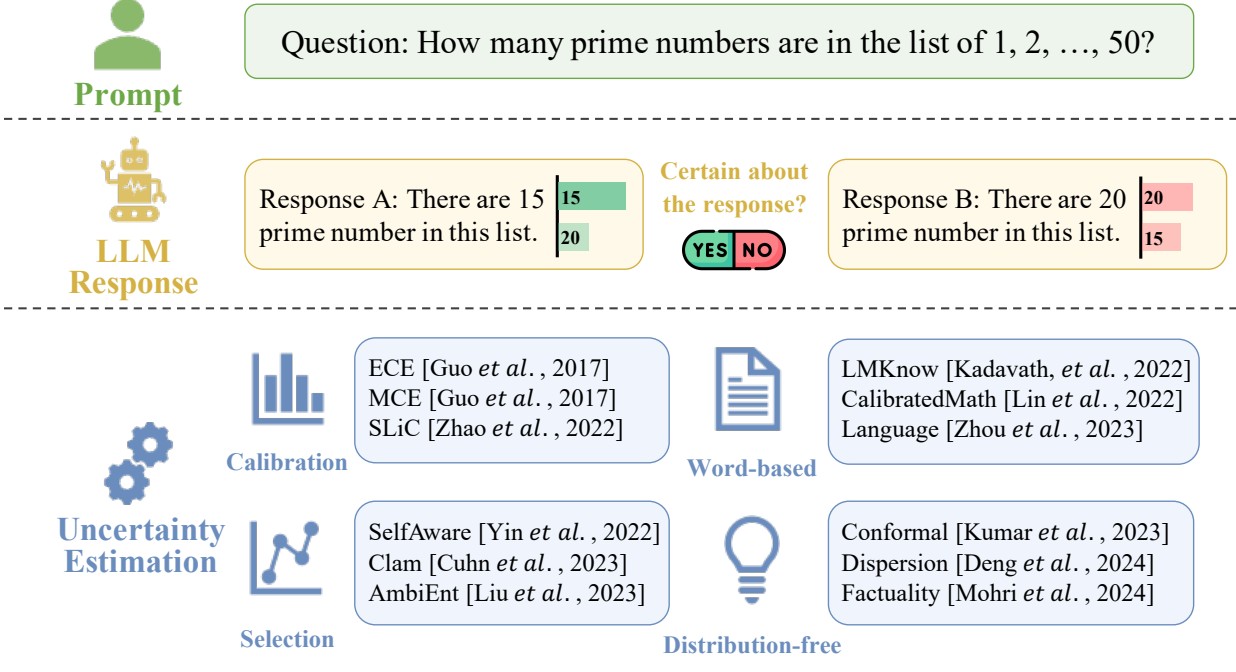

Figure 22: Overview of representative methods for estimating and mitigating uncertainty in foundation models. The illustration highlights how different answers may be generated with varying confidence levels, and categorizes existing approaches into calibration, word-based, selection, and distribution-free methods.

## 8.2.1 Estimating Uncertainty

One critical ingredient for the reliable deployment of a black-box LLM is the ability to measure the uncertainty in its responses so that appropriate decisions can be made based on its output [806]. Uncertainty in language model output is often quantified in terms of predictive entropy [464, 544, 612, 244, 949]. Although

predictive entropy can be calculated directly using output class probabilities for classification tasks, measuring the uncertainty in language model generations is a more challenging problem, requiring knowledge of the distribution over all possible sequences.

One popular way to address the challenge of sequence-level uncertainty quantification is to sample many potential generations from the model and use these samples to estimate the underlying distribution. For instance, Fomicheva et al. [272] use Monte Carlo dropout to draw samples, which are then utilized for quantifying uncertainty in machine translation. Malinin & Gales [612] instead employ an ensemble with Monte Carlo sampling techniques over the token outputs to produce both token-level and sequence-level uncertainty scores. Such approaches to measuring uncertainty via self-consistency are also studied in [421, 539, 806, 225, 464, 948]. While most self-consistency methods focus on the model's natural language outputs, Chen et al. [143] offer an effective method for measuring sample consistency in the embedding space. Besides, Stengel-Eskin & Van Durme [829] explore sequence-level uncertainty in semantic parsing through the angle of calibration, where they leverage the minimum confidence across tokens.

In addition to the challenge of sampling from the distribution of possible sequences, there may be many equivalent surface forms of a correct response to a question such as "What is the capital of Germany?" Accordingly, the desirable notion of uncertainty may go beyond the spaces of sequences into the space of semantic meaning. These challenges are outlined and addressed by Kuhn et al. [464], wherein a set of sequences are sampled and grouped by semantic equivalence to measure uncertainty over meanings instead of uncertainty over output forms. Lin et al. [544] build on this work by proposing more sophisticated semantic uncertainty measures and removing the access requirement to the token-level scores of the potentially black-box model. Also, Duan et al. [244] presents an algorithm based on a similar notion that aims to better characterize uncertainty by focusing on the most relevant token.

Finally, some have taken the approach of trying to explicitly identify epistemic uncertainty, which can then be addressed by incorporating additional information. Malinin & Gales [612] take an ensemble approach and characterizes epistemic or "knowledge" uncertainty using mutual information and the level of disagreement between models in the ensemble. Glushkova et al. [302] also apply an ensemble-based approach to quantifying epistemic uncertainty in machine translation. The use of more powerful and efficient techniques such as direct uncertainty prediction and heteroscedastic regression are investigated in Zerva et al. [1095] in the context of machine translation; they find these methods perform favorably compared to variance-based baselines such as MC dropout and deep ensembles while being considerably faster. Lahlou et al. [473] highlight the challenges of using Bayesian techniques or discrepancy-based measures of epistemic uncertainty and proposes a Direct Epistemic Uncertainty Prediction framework, wherein a secondary model is trained to estimate the point-wise generalization error and provides an upper bound on epistemic uncertainty. Their algorithm is shown to be useful in interactive learning environments, where the model can acquire novel examples and continue learning. Similarly, Osband et al. [684] use an epistemic neural network to identify uncertain data that should be prioritized in fine-tuning, achieving on-par performance while using half as much data as training without prioritization. Hou et al. [367] avoid the need to train a separate model to predict the epistemic uncertainty, instead using multiple LLM queries for clarification to rule out data uncertainty so that the remaining uncertainty of each prediction can be prescribed to epistemic uncertainty. As another ICL-based approach, Yadkori et al. [1027] propose an iterative prompting method to identify when epistemic uncertainty is large and highlight its usefulness in a setting with multiple good responses. More recently, Wang et al. [939] introduced Bayesian Low-Rank Adaptation (BLoB), which jointly updates both the mean and covariance of LLM parameters during fine-tuning to enhance uncertainty estimation, and Training-Free Bayesianization (TFB) [798], a post-hoc method that converts pre-trained LoRA adapters into Bayesian models without retraining, offering efficient and accurate uncertainty quantification.

### 8.2.2 Calibration

One popular method for characterizing a model's predictive uncertainty is concerning (confidence) calibration. For a model to be well-calibrated, its confidence estimates should, on average, reflect the probability of its correct answers. The most common calibration measure in the deep learning literature is Expected Calibration Error (ECE) [663, 329], which measures the expected difference between confidence and accuracy over the data distribution. However, since it is impossible to calculate this quantity directly, ECE is

typically estimated: data points with similar confidence scores are binned together, and ECE is calculated as the mean absolute difference between average confidence and accuracy over all bins. Other popular measures of calibration error include Maximum Calibration Error (MCE) [663, 329], Brier Score [111], negative log-likelihood [349] and other novel variants [887]. Since 0/1 accuracy is often not a suitable metric with respect to LLM performance, Huang et al. [390] propose Rank-Calibration Error, which captures whether higher uncertainty scores are associated with worse generations according to continuous metrics like ROUGE [537] or BLEU [689].

While modern neural networks have achieved impressive accuracy across a wide range of tasks and improved calibration relative to simpler methods [642], significant miscalibration remains, usually in the direction of overconfidence [329, 914]. To address this remaining calibration error, post-hoc recalibration methods such as Platt scaling [706], temperature scaling [329], or histogram binning [1091] can be used to refine the confidence estimates of a pre-trained model.

Most of the work in calibration and deep learning has focused on the classification setting, where the softmax probability of a class can be reasonably interpreted as a confidence score. Accordingly, extending techniques for measuring and improving calibration to LLMs in classification (or other settings with single-token answers from a finite, discrete set) is straightforward. For example, Desai & Durrett [215] find that pre-trained encoder-only transformer models like BERT and RoBERTa are well-calibrated under fine-tuning and that techniques like temperature scaling and label smoothing can be effective in combating poor confidence estimates. Further, Xiao et al. [1008] perform a large-scale analysis of how decisions made along the LLM deployment pipeline, such as model size, architecture, and training objective, affect downstream task calibration on sentiment analysis and NLI. They find that larger models generally give more accurate confidence estimates and that applying temperature scaling and fine-tuning with focal loss may be helpful. Zhao et al. [1138], Zhou et al. [1154] offer methods to debias answers for a prompt so that confidence scores are calibrated based on the actual input instance under consideration, while Detommaso et al. [216] introduce the notion of multi-calibration [1190] into the LLM setting by grouping examples based on binary attribute labels produced by the model itself. Kadavath et al. [421] perform an extensive study of whether LLMs can evaluate the correctness of their own responses across tasks such as multi-choice question answering. They find that self-evaluation improves with model size, although calibration is worse for more complex and out-of-distribution tasks. They also find that popular alignment techniques such as RLHF may hurt the calibration of LLM output probabilities. To handle the case of population shift, e.g., across the distribution of subjects in a sample of MMLU questions, Li et al. [515] propose to train a recalibration method that adapts to a new subset of the data given only a few unlabeled examples.

On the other hand, estimating confidence and measuring calibration is less straightforward when tasks are generative or open-ended (for the same sequence-related reasons outlined in Section 8.2.1). Thus much recent LLM calibration research has focused here [421, 1008, 808, 806, 862, 1141, 637, 573]. In early work highlighting this challenge, Ott et al. [685] analyze model calibration in the setting of neural machine translation, showing that these models tend to diffuse too much probability mass over the space of possible sequences. One popular avenue for addressing the difficulties of combining calibration and generation is the development of new methods for producing calibrated sequence-level confidence scores. To this end, Chen & Mueller [149; 150] combine self-consistency with self-evaluation to produce a confidence score using a method they call BSDetector and find it is more accurate than alternatives in identifying incorrect LLM responses for models like GPT-3 and ChatGPT. Si et al. [806] measure the calibration of GPT-3 on free-form QA using both the length-normalized language model output probability and self-consistency and finds both methods give more calibrated confidence scores than a supervised BERT baseline. Tian et al. [862] study LLM calibration of models aligned with RLHF, finding that these models can verbalize confidence scores that are more reliable than the underlying output probabilities, an approach which is especially useful when the model is behind an API and these probabilities are not available (see Section 8.2.3 for more on verbalized expressions of uncertainty). While most work on the calibration of LLM generations has focused on language tasks like question answering and summarization, Spiess et al. [825] study the calibration of LLMs for code generation across several tasks, correctness criteria, datasets, and approaches.

In addition to the approaches described above, researchers have also pursued techniques for better quantifying LLM confidence via model training, concerning an external recalibrator or the LLM itself. For instance,

Mielke et al. [637] address conversational agents' overconfidence by training a small auxiliary network to predict the appropriate level of confidence to be expressed. Liu et al. [573] offer further work in this direction, proposing to train a new linear layer that predicts a bias term to be added to the language model's output logits. Their approach enables the reordering of candidate generations (as opposed to temperature scaling) and is tested on longer generations including full paragraphs. Kadavath et al. [421] study whether a language model can be trained to predict the probability that a free-form answer to a question is correct; their experiments show promising results, although generalizing such behavior across distributions remains challenging. Lin et al. [539] use fine-tuning to teach a GPT-3 model to express its own uncertainty on various mathematics tasks, finding that responses are generally well-calibrated and remain reasonable under distribution shift. Finally, a supervised fine-tuning step is proposed in Band et al. [60] to induce linguistic calibration, where model outputs feature confidence estimates that enable downstream decision-makers to make calibrated probabilistic predictions.

### 8.2.3 Verbalized Uncertainty

Generally, in machine learning, confidence scores are numeric values extracted from a predictive model, for example, based on predicted class probabilities, logit entropy, or ensemble variance. However, the ability of LLMs to generate arbitrary text output enables a paradigm in which language models may express their uncertainty directly in their natural language output.

As an early example of such an approach, Kadavath et al. [421] verbalize language model calibration by verifying answers using the probability assigned to tokens such as "True" or "IK" ("I know") conditioned on its output or articulating confidence scores using numeric verbalizations such as "30%" or "80%". Their approach shows promise, although it may be difficult to generalize to new tasks or tasks that are difficult to format as multiple-choice. Additionally, Lin et al. [539] fine-tune a GPT-3 to directly express its confidence in its output using verbalized probabilities (e.g., "61%"), while Tian et al. [862] prompt a model directly to output both confidence scores and linguistic markers of confidence (e.g., "highly likely"). Zhou et al. [1157] study how linguistic markers of certainty, uncertainty, or evidentiality such as "I'm sure...", "I think...", or "Wikipedia says..." affect model confidence. Their findings imply that LLMs are sensitive to epistemic markers in prompts, with more than 80% variation in accuracy, and that expressions of high certainty result in a decrease in accuracy. Their results also suggest that the confidence scores that LLM outputs do not truly reflect epistemic and aleatoric uncertainty in response but instead are based on mimicking language use from the training set. This observation is supported by an extensive study of the ability of black-box models like GPT-4 to verbalize confidence in Xiong et al. [1015]. They find the verbalized uncertainty expressions overconfident and difficult to optimize across models and datasets with a single strategy for prompting, sampling, and scoring. In other relevant work, Mielke et al. [637] train a confidence calibration network to select linguistic expressions of uncertainty that should be included in the output of a conversational agent. Stengel-Eskin et al. [833] propose LACIE, which splits verbalized uncertainty into explicit markers (e.g., I'm not sure) and implicit markers (e.g., giving details or backstory, stating a person's expertise, etc.). The models are trained to improve calibration by modeling a listener who accepts or rejects answers based on their correctness. The generator is rewarded for providing correct answers that are accepted and penalized for incorrect answers being accepted or correct answers being rejected.

### 8.2.4 Addressing Uncertain Examples

Selection is another established tool for addressing uncertainty [295, 296, 271, 255, 1184]. We use the term selection broadly to encompass methods that identify inputs that are particularly difficult for the model. We offer interventions like allowing the model to abstain from the prediction (the classic paradigm in selection) or request further information. Selection has been well-studied in the context of language models [197, 423, 806], and has been shown to improve outcomes concerning hallucination and safety [867]. Kamath et al. [423] investigate selective question answering under domain shift, proposing a novel algorithm that incorporates out-of-distribution data to train a selection model that identifies examples on which the model is likely to err. Gupta et al. [338] derive a new score based on token-level uncertainty features, to identify examples that should be deferred from a smaller model to a larger model. Uncertainty scoring methods, for example, the semantic entropy-based measures proposed in Lin et al. [544], are often evaluated via selection to highlight

how such measures are useful for predicting the correctness of LLM responses. Stengel-Eskin & Van Durme [830] show that we can recover low-confidence examples in semantic parsing by rephrasing and asking for user confirmation, which is the number of questions the model abstains from while keeping model safety high. To support work on selection in LLMs, Yin et al. [1064] introduce the *SelfAware* dataset of questions that should be recognized as unanswerable.

Given the opportunity for interactivity provided by the text interface, a significant amount of research has gone towards algorithms to enable the LLM to request further information before responding, particularly in the case of ambiguous questions. For instance, Kuhn et al. [463] use few-shot learning to detect ambiguous questions that require clarifying questions, while Kim et al. [438] propose a tree-based approach to disambiguating questions and retrieving missing information. As the interest in identifying ambiguous questions in LLMs has grown, there has been an accompanying effort to release public datasets that can be used to evaluate the relevant abilities. Liu et al. [548] offer AmbiEnt, a dataset to test an LLM's ability to manage ambiguity in resolving entailment relations, finding their task difficult even for powerful commercial models like GPT-4. Additionally, Tamkin et al. [848] introduce AmbiBench, a benchmark of ambiguous tasks where the ambiguity is introduced by the task description itself (as opposed to the specific instance of the task). Stengel-Eskin et al. [831] create a dataset for identifying and disambiguating instances of the visual question-answering task with MLLMs. Besides, Stengel-Eskin et al. [832] introduce a dataset of ambiguous queries and their logical forms and test whether models can recover both interpretations. Also, Saparina & Lapata [767] introduce a similar ambiguous parsing dataset but with human-sourced SQL queries.

### 8.2.5 Distribution-free Uncertainty Quantification

As LLMs are increasingly deployed in risk-sensitive domains such as medicine, law, and finance, it may be important to have not only an estimate of the uncertainty in a model's response but also a high probability upper bound on the error rate at test time. Recently, there has been increasing research employing techniques from the Distribution-Free Uncertainty Quantification (DFUQ) family to control the risk of deep learning systems. This line of work generally descends from the literature concerned with conformal prediction [781, 907], wherein a threshold on class probabilities is calibrated to produce prediction sets that fulfill some coverage (i.e., recall) guarantee. Angelopoulos & Bates [30] offer a tutorial on the subject in the context of modern neural network applications, and Kumar et al. [466] illustrate the application of conformal prediction to multi-choice question answering with LLMs. To broaden its applicability, Angelopoulos et al. [32] derive a version of conformal prediction for bounding the expectation of any monotone loss function and studies their method in open-domain question answering. Recent work has offered algorithms for producing bounds on more general loss functions concerning the mean [31], quantile-based risk measures like value at risk (VaR) [815], and measures of statistical dispersion like the Gini Coefficient or differences in loss among protected subgroups [213].

While it is straightforward to apply existing DFUQ techniques to classification with LLMs [815, 213, 466], the question of how best to apply them to generation tasks like summarization, chat, and code remains open. Multiple approaches have been proposed to apply these techniques to language model decoding. For example, Schuster et al. [777] utilize the Learn Then Test framework [31] to calibrate early exit criteria concerning the number of transformer layers applied to an input. Their goal is to identify when an LLM is sufficiently confident that it can exit the forward pass, and thus reduce the amount of computation used. In the conformal prediction vein, Quach et al. [722] calibrate a stopping rule to produce a set of candidate generations that with high probability contains a suitable response (while removing redundant candidates), and Deutschmann et al. [217] incorporate conformal prediction into a novel beam search algorithm. To mitigate the risk of models hallucinating answers, Yadkori et al. [1028] proposes a conformal abstention procedure using measures of self-consistency that are evaluated by the LLM itself. Finally, Mohri & Hashimoto [650] enforce factuality in LLMs by using conformal techniques to determine a level of specificity with which a given question can be answered.

As a more general approach, Prompt Risk Control [1185] unites many techniques from the DFUQ family under a single framework for selecting a prompt (e.g., system prompt or set of few-shot examples) based on rigorous upper bounds on rich families of informative risk measures. The authors propose a two-step prompt selection process. First, a set of prompts is validated as producing an acceptable risk for some contextually

relevant measure before a final prompt is chosen based on some performance metric, like average reward or accuracy. Prompt Risk Control can be applied to any bounded loss function, such as top-1 accuracy, ROUGE, or toxicity, and can be used to control risk measures, including tail quantities like value-at-risk or measures of statistical dispersion such as the Gini coefficient.

### 8.3 Current Limitations and Future Directions

Much work has gone into methods to identify and address uncertainty in foundation model generation. However, existing results and methods are limited and much work remains to be done before these models can be responsibly and reliably deployed.

First, many results in uncertainty quantification in LLMs are produced in limited settings. Experiments are usually performed on tasks like trivia question answering, which can be answered via a single token, word, or short phrase. Further, the tasks under study also often assume that there is only one right answer: there may be no uncertainty in the correct response to "Who won Super Bowl XX?". However, much LLM usage revolves around tasks that require generating long-form responses to open-ended queries, for which multiple reasonable answers exist. Some works have made progress in this direction [1102, 1103, 1068], it is unclear whether the results produced in these limited settings offer insight into more complex, uncertain, and sequential settings, such as chat or customer care.

Alongside the difficulty of extrapolating results from simple settings, existing methods for improved uncertainty quantification of language model generations have come at the expense of generating multiple times for a single query. Given that certain methods can increase costs by 2 to 20 times—or more—compared to standard inference, it will be infeasible for LLM users or service providers to adopt such approaches. Furthermore, although modern frontier models have shown some ability to express their uncertainty in words, there is good evidence that any correlation between accuracy and verbalized expressions of confidence is simply a result of spurious features in training data [1157]. In addition, it should be noted that these verbalization techniques also usually require extra inference costs, even for the simplest methods, such as scoring $p(\text{True})$ for the generated answer. Finally, although these algorithms have largely not been tested in open-ended tasks and over long generations, it seems probable that new tools will be needed in this setting. For example, consistency-based methods assume that producing diverse samples for a particular query indicates an example for which the model will likely give a poor answer. However, a model that can only produce a single answer to a query such as "Tell me a joke" or "Write me a story" would lack the capabilities to suit many modern LLM use cases.

Overall, it is unclear whether any advanced method for quantifying LLM uncertainty in the zero-shot setting robustly outperforms a baseline sequence entropy score calculated using token probabilities. Note that these scores are often unavailable for black-box LLMs behind an API. Furthermore, it is difficult to imagine how best to exploit probabilities taken directly from the language model, since these probabilities do not necessarily relate to the task at hand [623], but instead reflect the cross-entropy objective used in training and plausibility of an answer under the training data distribution (unless the model receives RLHF, which makes accurate uncertainty estimation even more difficult [421, 862]).

Besides addressing the limitations in methodology and experimental settings mentioned above, future work in this area may benefit from taking a broader view of the challenge of quantifying and addressing uncertainty in generative models. It could explore how uncertainty can be better quantified and addressed across the entire model development and deployment pipeline, and how interventions and measurements at different points in the pipeline interact and affect downstream outcomes. Also, it may be useful to gain a more thorough understanding of how techniques for selecting, mixing, and filtering training data affect a user's ability to accurately estimate the model's confidence on downstream tasks, whether via token probabilities or verbalizations. As new architectures and pre-training recipes emerge, they should be benchmarked for calibration, not only accuracy. Fine-tuning algorithms, whether supervised or RL, have been shown to worsen models' UQ characteristics, and this phenomenon must be kept in focus as the community iterates on these methods. Finally, given a model that has been pre-trained and fine-tuned and is ready for deployment, we might develop new methods to select system prompts and few-shot exemplars that reduce and control uncertainty in the wild, ideally with rigorous statistical methods like those provided by DFUQ [1185].

# 9 Distribution Shift

Foundation models can occasionally produce unacceptable errors when faced with distribution shifts. These models, typically trained on a fixed corpus, require additional adaptation for new tasks. This limitation is particularly challenging in our ever-changing world, where knowledge is constantly shifting due to various factors, such as changes in location or time [430, 442]. For instance, if a model trained before 2023 is asked, "Which team does Messi play for?", it may incorrectly assign a higher probability to Paris Saint-Germain instead of Inter Miami. This example highlights the importance of understanding, detecting, and mitigating distribution shifts in foundation models to improve their reliability.

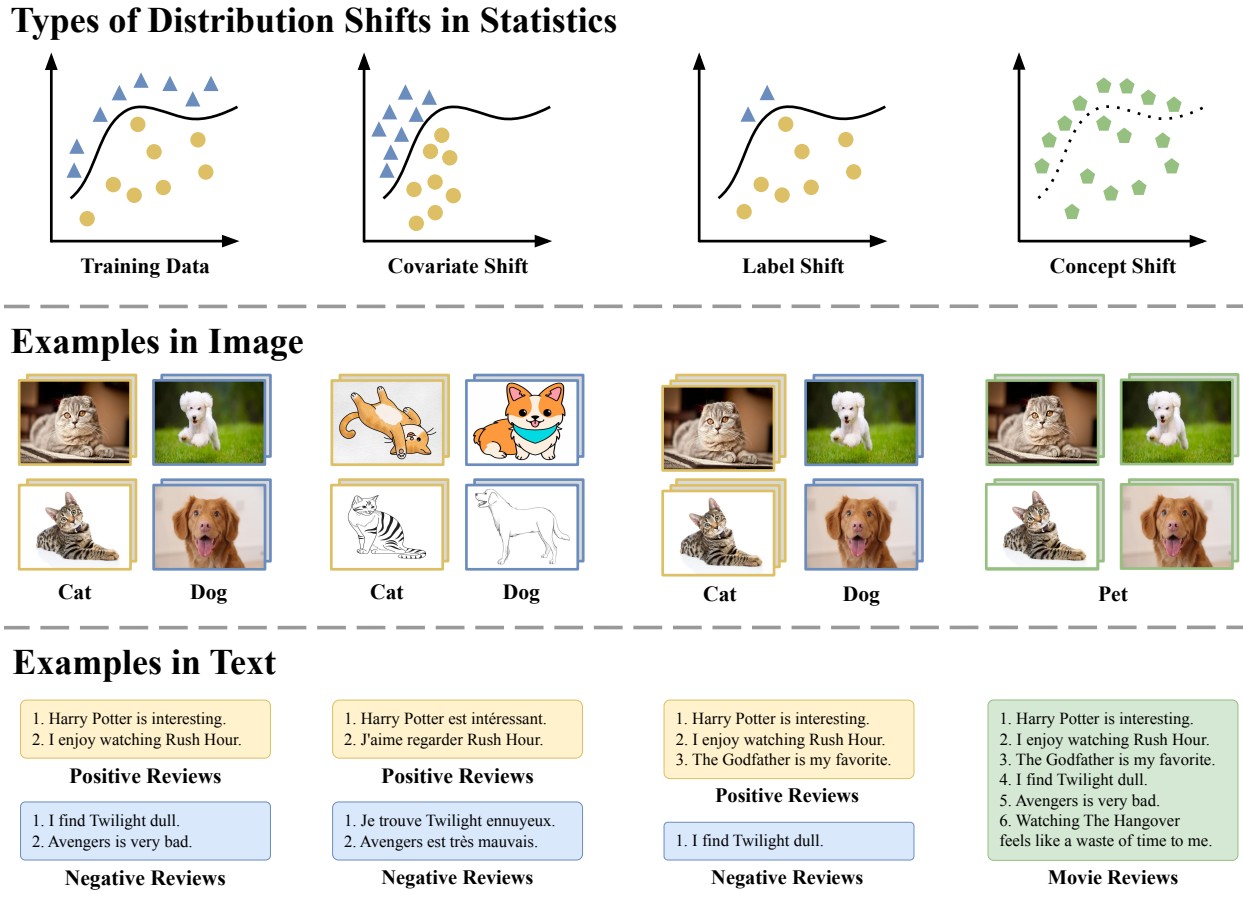

Figure 23: Different types of distribution shifts in the perspectives of (1) statistics, (2) image, and (3) text. The concept shift scenarios show how two distinct classes can merge into a single class when labels change.

## 9.1 Definition and Categorization

The distribution shift [475, 45, 396] occurs when the independent and identically distributed (i.i.d.) assumption does not hold between the training and test distributions. This divergence between the training distribution $p_{\text{train}}$ and the test distribution $p_{\text{test}}$ can significantly impact the performance of machine learning models, including foundation models. In essence, distribution shift describes the scenario where $p_{\text{train}} \neq p_{\text{test}}$, which can degrade model performance and reliability.

Based on how the data distribution changes, distribution shifts can be classified into three primary categories, with examples from various domains presented in Figure 23.

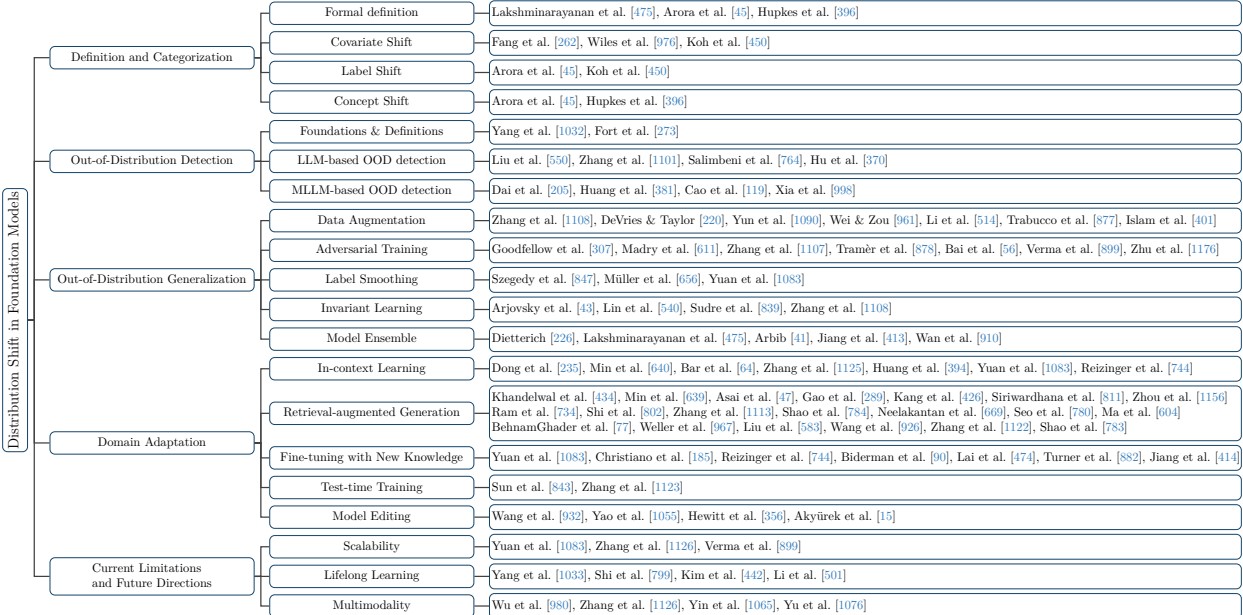

Figure 24: Taxonomy of Distribution Shift in Foundation Models.

- **Covariate Shift.** This term refers to changes in the feature distribution $p(x)$ while the relationship between the features and the labels $p(y|x)$ remains unchanged. This type of shift is prevalent in scenarios where the environment or context of the features change.
- **Label Shift.** It occurs when the distribution of labels $p(y)$ changes, while the conditional distribution of features given labels $p(x|y)$ remains constant. This shift can result from changes in the real-world phenomena being modeled.
- **Concept Shift.** Concept shift, also known as conditional shift or concept drift, happens when the relationship between the features and the labels $p(y|x)$ changes. It reflects the evolution of the underlying problem statement or process over time.

## 9.2 Out-of-Distribution Detection

Out-of-distribution (OOD) detection involves identifying inputs different from the training distribution [1032, 273], which plays a vital role in enhancing the reliability of foundation models. By flagging unfamiliar data points for further scrutiny, these techniques help mitigate risks and maintain the integrity of the model's performance. Despite the impressive generalization capabilities of today's extremely large foundation models, they remain fundamentally bounded by their training data. When deployed in dynamic open-world environments, such models can still encounter domain shifts, rare edge cases, or adversarial input that leads to overconfident but incorrect predictions. Therefore, OOD detection remains crucial: not only as a safeguard against catastrophic failures in high-stakes applications (e.g., medicine, autonomous driving), but also as a tool to trigger human oversight, guide active learning and preserve trust in automated decision-making systems.

In the context of language models, Liu et al. [550] present an empirical investigation into the OOD detection capabilities of LLMs, specifically examining the LLaMA families with different model sizes. The study evaluates common OOD detectors in both zero-shot and fine-tuning scenarios, yielding several significant insights: (i) LLMs inherently serve as effective OOD detectors without requiring fine-tuning. (ii) In-distribution (ID) fine-tuning can boost OOD detection. (iii) Generative fine-tuning demonstrates superior generalization ability because it aligns with the pre-training objectives of LLMs. (iv) A simple cosine distance OOD detector proves to be highly effective, attributed to the isotropic nature of LLM embedding spaces. Furthermore, Zhang et al. [1101] propose a novel approach for OOD detection, utilizing the likelihood ratio between a

pre-trained LLM and its fine-tuned variant. This method leverages the pre-trained LLM's extensive prior knowledge about OOD data, which, when fine-tuned with ID data, can effectively differentiate between ID and OOD samples. Expanding on these findings, Salimbeni et al. [764] explore the effectiveness of un-merged Low-Rank Adaptor (LoRA) [370] weights for OOD detection during the fine-tuning process, further contributing to the growing body of research in this area.

In addition to textual OOD detection, recent advancements have begun to harness the powerful representation capabilities of foundation models in visual OOD detection. Dai et al. [205] propose a method to enhance OOD detection by selectively generating information from LLMs. Their method incorporates a consistency-based uncertainty calibration to estimate generation confidence scores and extracts visual objects from images to leverage the world knowledge encoded in LLMs. ODPC [381] utilizes LLMs to generate specific prompts for creating "OOD peer classes," which are synthetic categories constructed from in-distribution (ID) semantics but intentionally placed outside the original label space. These peer classes act as proxy OOD categories during training, enabling the model to learn tighter ID class boundaries and better distinguish unfamiliar samples. This approach serves as an auxiliary modality for detection and introduces a contrastive loss based on OOD peer classes to learn compact ID class representations and clarify boundaries between different classes. EOE [119] improves OOD detection by tapping into the expert knowledge and reasoning capabilities of LLMs without requiring actual OOD data. This method is designed to adapt to various open-world scenarios, making it suitable for (i) *far OOD* detection, where the OOD samples come from entirely different domains (e.g., animals vs. vehicles); (ii) *near OOD* detection, where the OOD samples are semantically close but from unseen categories (e.g., unseen dog breeds when trained on other breeds); and (iii) *fine-grained OOD* detection, where differences are subtle and intra-class variation is high (e.g., distinguishing between visually similar medical conditions). In the medical domain, CARES [998] evaluate the OOD detection capability of medical LLMs, focusing on their ability to detect medical images that differ significantly from those used in the training phase.

## 9.3 Out-of-Distribution Generalization

OOD generalization, on the other hand, aims to enhance the robustness of foundation models under new, unseen environments [353, 560, 1039, 1000, 665]. This approach improves the model's resilience to variations in input data through diverse techniques. Prior to the era of foundation models, the deep learning community explored a rich set of strategies for OOD generalization, supported by extensive empirical studies. These included (i) *data augmentation*, where transformations were applied to create synthetic training examples [460, 803]; (ii) *adversarial training*, which exposed models to adversarially perturbed inputs [307, 611]; (iii) *label smoothing*, a regularization technique to prevent overconfidence [847, 656]; (iv) *invariant learning*, which aimed to capture features stable across environments [43, 12]; and (v) *model ensembles*, which aggregate predictions from multiple models to reduce variance and improve robustness [475, 226]. Yuan et al. [1083] evaluate these commonly used methods for LLMs, leading to important insights and conclusions.

### 9.3.1 Data Augmentation

Data augmentation [1108, 220, 1090] involves creating new training examples through various transformations of the original data. These transformations range from simple operations, such as flipping or rotating images in computer vision tasks, to more complex manipulations by generative models to simulate the data distribution [514, 877, 401]. In the NLP context, Easy Data Augmentation (EDA) [961] refers to a set of simple, low-cost textual augmentation operations — synonym replacement, random insertion, random swap, and random deletion — designed to increase lexical diversity without altering overall meaning. EDA was originally shown to be effective for small-scale text classification tasks, but its naive application to LLMs often degrades performance due to distributional shifts in token usage and disruption of learned long-range dependencies. The primary objective of data augmentation is to increase the diversity of the training set, thereby enabling the model to learn more robust features that generalize better to unseen data. However, recent research has shown that applying simple augmentation techniques, such as EDA, to LLMs often leads to performance degradation across most tasks, underscoring the need for more advanced augmentation methods tailored to foundation models.

### 9.3.2 Adversarial Training

Adversarial training [611, 56] is a robust technique used to improve OOD generalization by exposing models to adversarial examples during the training process. These adversarial examples are inputs deliberately perturbed to mislead the model into making incorrect predictions, despite appearing similar to regular data. In earlier deep learning literature, adversarial training was shown to improve robustness in image recognition [307, 878], and it is now increasingly applied to LLMs and MLLMs.

Mechanistically, adversarial training operates by solving a min-max optimization problem [611, 1107]: the inner maximization finds the worst-case perturbation within a certain norm-ball around each input, while the outer minimization updates model parameters to correctly classify these perturbed inputs. By repeatedly training on such challenging examples, the model learns smoother and more stable decision boundaries, which are less sensitive to input shifts, thereby improving robustness and generalization to unseen or distribution-shifted data.

In LLMs, Free Large-Batch (FreeLB) [1176], an adversarial training method that adds perturbations to the input data, improves generalization performance in most scenarios. Similarly, Verma et al. [899] introduce image perturbations in MLLMs through augmentations like noise addition, blurring, and median filtering. Additionally, they craft adversarial questions using conjunctions, disjunctions, and negations to challenge models' reasoning abilities. Among the tested augmentations, Gaussian Noise Addition is identified as the most detrimental, causing the largest decline in performance. The study also finds that the complexity of questions, especially those with multiple connectives, significantly impacts the models' performance.

### 9.3.3 Label Smoothing

Label smoothing [847] is a regularization technique used to improve OOD generalization by preventing the model from becoming overly confident in its predictions [656]. Unlike traditional training algorithms where models learn to assign a probability of 1 to the correct class and 0 to all others, label smoothing introduces a small probability to incorrect classes. This approach encourages models to maintain a degree of uncertainty in their predictions, potentially improving their ability to generalize to unseen data. In the context of LLMs, however, the effectiveness of label smoothing has been called into question. Yuan et al. [1083] conducted experiments where they smoothed the hard labels in the training data but observed that this technique did not improve the LLMs' generalization ability.

### 9.3.4 Invariant Learning

Invariant learning [43] plays a crucial role in OOD generalization by capturing invariant representations or predictors across different environments while disregarding spurious correlations. One notable approach of invariant learning involves the use of specialized loss functions, such as Focal Loss [540], Dice Loss [839], and Mixup Loss [1108]. Focal Loss was originally designed for class-imbalanced detection tasks, down-weighting well-classified examples to focus training on harder cases. Dice Loss, derived from the Sørensen–Dice coefficient, is widely used in segmentation to maximize overlap between predicted and ground-truth regions, thus emphasizing recall. Mixup Loss linearly interpolates both inputs and labels between pairs of examples, encouraging the model to behave linearly in-between training samples and reducing overfitting to spurious patterns. These loss functions differ in their inductive biases — e.g., emphasizing difficult examples, optimizing overlap, or encouraging linearity — but all aim to produce more generalizable decision boundaries that are less sensitive to environment-specific correlations. By applying Focal Loss to the training process of LLMs, these models emphasize hard-to-classify examples and enhance their ability to handle diverse and unfamiliar inputs.

### 9.3.5 Model Ensemble

Model ensemble [41, 475] is a powerful technique for enhancing the robustness and performance of AI models in complex environments. This approach combines predictions from multiple models to produce more accurate and reliable final outputs. Yuan et al. [1083] evaluated model ensembling but observed limited improvement in generalization ability. However, building on this foundation, recent studies by Jiang

et al. [413] and Wan et al. [910] have introduced more advanced model ensemble algorithms, improving performance across several downstream tasks.

## 9.4 Domain Adaptation

Unlike OOD generalization, domain adaptation tailors the model to domain-specific tasks by injecting domain-specific knowledge [292, 812], including in-context learning (ICL), retrieval-augmented generation (RAG), fine-tuning, test-time training, and model editing. These methods enable foundation models to specialize in particular domains while maintaining their broad capabilities.

### 9.4.1 In-context Learning

In-context learning (ICL) shows great potential to address the gap between foundation models and domains not covered in their pre-training and fine-tuning data [235, 640]. Recently, ICL has gained attention as a transformative approach for foundation models [64, 1125, 394]. It demonstrates the ability to adapt to new tasks or distributions without altering model parameters by adding domain-specific input-output pairs to the test example. This augmented input serves as a guide, helping the model produce desired outputs for new tasks. Consequently, ICL offers a flexible and efficient method for continual adaptation without the need for computationally expensive retraining.

In the field of LLMs, the BOSS benchmark [1083] explores ICL for LLMs by using examples from both ID datasets and the training split of OOD datasets. The findings reveal that fine-tuning domain-specific models is advantageous when sufficient training data is available, while LLMs with ICL perform better in low-resource scenarios. Notably, the effectiveness of ICL varies across models and tasks, highlighting the need for task-specific adaptation strategies. Complementing this research, Reizinger et al. [744] delve into the intricacies of ICL, focusing on its approximate non-identifiability and the implications for understanding LLMs. Through a combination of mathematical examples and empirical observations, their work demonstrates how this approximate non-identifiability manifests in OOD generalization, providing deeper insights into the behavior of ICL in various contexts.

For MLLMs, Zhang et al. [1126] demonstrate that ICL can significantly enhance the generalization capabilities, suggesting new approaches to overcome existing limitations. However, their study also investigates the robustness of ICL under various distribution shifts. The findings reveal that ICL is vulnerable to domain shifts, label shifts, and spurious correlation shifts between in-context examples and test data.

### 9.4.2 Retrieval-augmented Generation

Retrieval-augmented generation (RAG) enhances foundation models by retrieving relevant information from external data sources to supplement input queries or generated outputs [434, 639, 47]. This process provides necessary domain knowledge, mitigating distribution shifts and improving generation quality [289, 426, 811, 1156]. In practice, RAG techniques are effective and efficient to apply in various unseen tasks with simple adaptation of the retrieval component, requiring minimal or even no additional training [734].

In the context of language models, Shao et al. [783] construct MASSIVEDS, a massively multi-domain database comprising 1.4 trillion tokens of both general web data and domain-specific data. Their findings demonstrate that as the database's size and diversity increase, more distributions are covered during inference, reducing OOD scenarios. To incorporate this domain knowledge without requiring additional training, recent studies [802, 734] focus on in-context Retrieval-Augmented Language Models (RALMs). These models directly input a concatenation of all retrieved texts as additional context to LLMs. For the choice of retriever, most work [1113, 784, 669, 780] employ an embedding model to decide what to retrieve. However, with the increasing prevalence of LLMs, researchers have begun using the models themselves as retrievers to improve accuracy [77, 604, 967, 583, 926]. In a parallel direction, as these RAG methods may retrieve irrelevant information that even hurt the performance, Zhang et al. [1122] proposed RAFT to further fine-tune the LLMs to learn to disregard "distractor documents" within the provided context, thereby enhancing the model's ability to focus on relevant information. The effectiveness of these In-Context RALMs has been

further demonstrated in several domain-specific tasks [1022, 506, 1014, 585], showcasing the potential of RAG in addressing real-world distribution shifts.

To extend RAG to multimodal query input [1137], Wei et al. [958] create M-BEIR, a multimodal instruction-following benchmark building on existing 10 diverse datasets. UniIR is trained on M-BEIR to take a heterogeneous query to retrieve from a heterogeneous candidate pool with millions of candidates in diverse modalities. Built upon it, UniRAG [786] employs UniIR's CLIP Score Fusion and BLIP Feature Fusion models as retrievers, improving performance in MLLMs. For visual question answering (VQA) tasks, RA-VQA [541] proposed a novel framework for joint training of the retriever and the answer generator, and FLMR [542] further improved the retrieval accuracy by combining multi-dimensional embeddings from language and vision models. Similarly, MuRAG [158] uses T5 [729] and ViT [238] for text and image encoding respectively, and retrieval from a large-scale memory bank for knowledge-based VQA. To improve embodied agents, MART [1086] utilizes interaction data to fine-tune a multimodal retriever based on preference learning. For image captioning and text-to-image generation tasks, RA-CM3 [1057] enhances performance by using a pre-trained CLIP model to augment inputs for a CM3 Transformer. These methods effectively address the shift in knowledge representation across modalities. Additionally, domain-specific multimodal RAG solutions have shown promising results in various fields [1002, 468, 852].

### 9.4.3 Fine-Tuning with New Knowledge

Fine-tuning is a widely adopted method for addressing domain adaptation in foundation models [744, 1083, 446]. This technique involves adapting pre-trained models to specific downstream tasks by further training them on task-specific datasets. The primary goal is to enhance the model's performance on new, unseen data that may differ from the data it was initially trained on.

The BOSS benchmark [1083] evaluates vanilla fine-tuning for LLMs, which involves directly fine-tuning pre-trained models on ID datasets without any additional processes. This benchmark helps investigate the relationship between performance on ID and OOD datasets by varying factors such as model scale, training steps, available training samples, and tunable parameters. Observations indicate that fine-tuning with the full dataset generally yields superior performance for ID examples, while LLMs employing in-context learning (ICL) paradigms demonstrate better performance on OOD instances. Reizinger et al. [744] explore the non-identifiability of fine-tuning in LLMs, highlighting its implications for understanding and improving these models. They argue that fine-tuning is non-identifiable, meaning that models with similar fine-tuning performance (such as equivalent test loss) can exhibit markedly different behaviors when applied to real-world tasks.

To address OOD generalization, Kirk et al. [446] investigate Reinforcement Learning from Human Feedback (RLHF) [185], which is typically implemented in three stages: (i) supervised fine-tuning (SFT), where the model is aligned with high-quality human-labeled data; (ii) reward modeling, where a learned reward model predicts preference scores for outputs; and (iii) reinforcement learning, where the base model is optimized against the reward model. While SFT is the first step in RLHF, it can also be viewed as a standalone fine-tuning approach. In their experiments, the authors compare the full RLHF pipeline against using only SFT and find that RLHF generally yields stronger generalization to new, unseen inputs, especially under significant distribution shifts between training and testing data.

Jiang et al. [414] propose a novel method for fine-tuning LLMs in domains where obtaining large volumes of high-quality, domain-specific data is challenging, such as healthcare or harmless content generation. They re-evaluated the Transformer architecture to identify the most impactful parameter updates. Their analysis revealed that within the self-attention and feed-forward networks of the Transformer architecture, only the attention parameters significantly benefit downstream performance when there is a mismatch between the training and test set distributions. Based on this insight, they proposed Training All parameters but Inferring with only Attention (TAIA), which involves updating all parameters during training but utilizing only the fine-tuned attention parameters during inference. Additionally, recent studies have observed that parameter-efficient fine-tuning (PEFT) methods, such as Low-Rank Adaptor (LoRA), can maintain more general capabilities from the pre-trained distribution while acquiring new knowledge from the fine-tuning data [90].

Beyond parameter-efficient methods, recent work has explored *activation steering* as an alternative or complement to conventional fine-tuning for improving generalization. This line of research modifies model activations at inference time or during light-weight training to achieve desired behavioral shifts without large-scale parameter updates. For example, Lai et al. [474] propose Joint Localization and Activation Editing, which identifies and edits specific activation subspaces relevant to the target domain, enabling effective low-resource fine-tuning. Similarly, Turner et al. [882] introduce Activation Addition, a technique for steering model outputs by adding direction vectors in activation space, allowing the incorporation of new knowledge or behavioral adjustments without gradient-based optimization. These approaches can reduce overfitting to in-distribution features while selectively enhancing capabilities relevant for OOD settings.

For multimodal scenarios, the proposal of EMMA [1043] adapts LLMs to the field of embodied multimodal agents. The key technique involves distilling the reflection outcomes of the LLM, which improves actions derived from analyzing mistakes in text world tasks. It uses these outcomes to fine-tune the vision-language models on analogous tasks in the visual world, which is capable of quickly adapting to the dynamics of the visual world. The cross-modality imitation learning is facilitated by a novel DAgger-DPO algorithm, which ensures that EMMA can generalize to a wide range of new tasks without further guidance. Belyaeva et al. [80] describe a method to address OOD challenges by developing a framework called HeLM (Health Large Language Model for Multimodal Understanding). HeLM integrates multiple data modalities, learns robust data encodings, and enhances predictive performance through comprehensive data utilization to achieve OOD generalization. Regarding model architectures, Ito et al. [402] find that models with multiple attention layers or those leveraging cross-attention mechanisms between input domains perform better in their constructed gCOG benchmark. Their study emphasizes that cross-modal attention and deeper attention layers are crucial for integrating multimodal inputs and improving generalization in the presence of distractors and new tasks.

### 9.4.4 Test-time Training

Test-Time Training (TTT) methods view each test instance as an individual learning problem with its own generalization target. This method creates a self-supervised learning task for each test sample and updates the model parameters at test time before making a prediction. In the LLM era, Sun et al. [843] propose a new class of TTT layers for sequence modeling that transform the hidden state into an optimizable model, with the update rule functioning as a step of self-supervised learning. Zhang et al. [1123] propose a comprehensive recipe of TTT with customizable large chunk updates to enable long sequence modeling validated on diverse modalities. By aligning the training and test data distributions, these methods significantly enhance model performance when faced with distribution shifts.

### 9.4.5 Model Editing

All the domain adaptation methods discussed above modify a model's behavior by incorporating new knowledge. This process is closely related to model editing for foundation models [932, 1055], which aims to rectify specific errors without affecting unrelated inputs. To explore its potential in addressing distribution shifts, we will now provide an overview of model editing approaches, which typically adhere to three essential properties:

- Reliability: The edited model should successfully produce the desired output for the edited sample, such as correctly answering 'Inter Miami" when asked Who does Messi play for?"
- Generality: The corrections made should be consistent across equivalent contexts, for example, accurately responding to 'Which team is Messi in?"
- Locality: The acquired knowledge should be minimally affected, ensuring that unrelated queries like 'Who does LeBron James play for?" remain unaffected.

These properties ensure the reliability, generality, and locality necessary for the effective and efficient correction of foundation model behaviors. Recent studies in model editing [356, 15] have also demonstrated promising performance in several OOD scenarios. Next, we will delve deeper into four distinct categories of model editing in LLMs (Figure 25), subsequently extending our discussion to address related issues in

MLLMs.

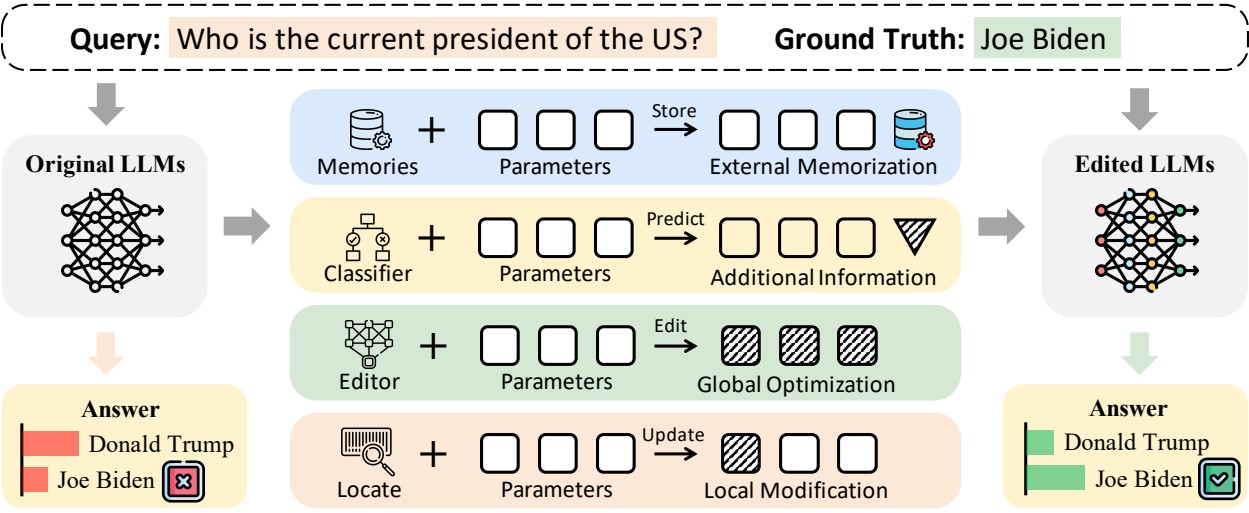

Figure 25: An overview of model editing methods in LLMs. Given an incorrect response from the original model, different editing strategies correct factual errors by modifying or augmenting the model's knowledge.

**Memory-based Model Editing**. In memory-based approaches, an external memory, outside the intrinsic architecture of the pre-trained LLM, serves as a repository for edited knowledge. LLM can access and modify this external memory during inference. For example, Language Patch [660] performs editing by integrating with a library of patches in natural language, and MemPrompt [608] adopts a growing memory bank as a look-up table to store the edit sample and its corresponding prompts, which is used to alter the prediction of the edit sample. KAFT [501] further strengthens the controllability and robustness of LLMs' working memory through counterfactual data augmentations. In this approach, the entity representing the answer in the context is substituted with an alternative but still plausible entity. This substitution is intentionally designed to introduce a conflict with the genuine ground truth, thereby incorporating counterfactual and irrelevant contexts to standard supervised datasets. In addition to relying on parameter-based memory, IKE [1147] introduces novel factual information into a pre-trained LLM via in-context learning, where a set of demonstrations will alter the prediction of a target factual detail when the input is influenced by an edit. To solve more complex questions involving chains of facts, MQuAKE [1149] enables editing by breaking down each question into iterative subquestions and retrieving the most pertinent fact from the edited fact memory.

**Classifier-based Model Editing**. The classifier-based model editing paradigm aims to preserve pre-trained parameters while utilizing a classifier to determine whether behavior adjustment is necessary. In this approach, if a sample falls outside the scope of the edit sample, the original model is applied to maintain predictions. Conversely, interventions occur when the sample is within the scope, with the specific interventions varying across different methods. SERAC [647] employs a scope classifier to determine whether the original model or a new lightweight model should be used for prediction. The new lightweight model is specifically trained for in-scope samples. In contrast, Language Patch [660], CaliNET [234], and T-Patcher [395] introduce additional trainable parameters to adapt the original model instead of requiring entirely new models. For example, Language Patch trains a new gating head (acting as a classifier) to combine predictions from the original prediction head and a newly trained interpreter head. CaliNET and T-Patcher insert a residual block into the original model's feed-forward network (FFN) as an adapter. This adapter utilizes an activation operation on hidden states to determine whether the intervention should be activated. When the activations are zero, there will be no change to the original prediction. However, the success of these classifier-based methods heavily relies on the quality of the classifier, which also necessitates a substantial number of unrelated samples for training. Alternatively, GRACE [347] edits a model by adding a retrieval-based adaptor to a chosen layer that enables judicious decisions regarding the utilization

of the dictionary for a given input, accomplished via the implementation of a deferral mechanism.

**Hypernetwork-based Model Editing**. The hypernetwork-based model editing paradigm utilizes an external model, referred to as the editor, to facilitate parameter updates in the models. Knowledge Editor (KE) [120] employs a bidirectional LSTM to transform an edit pair, consisting of the edit sample, incorrect prediction, and correct label, into shifting operation parameters (i.e., mask $\mathbf{m}$, offset $\mathbf{b}$, and scaling factor $\alpha$) for $\nabla$: $\hat{\nabla} = \alpha(\mathbf{m} \odot \nabla) + \mathbf{b}$. Based on KE, SLAG [348] further appends metrics for two types of input texts: (1) those that, while not part of the targeted edit set, align logically with it; and (2) those that share a formal resemblance to edited knowledge, but do not affect the prediction outcomes. However, hyper-networks are generally incapable of updating LLMs due to the massive parameter size. To address this issue, MEND [646] applies low-rank decomposition to $\nabla$ and utilizes two MLP layers to generate a new low-rank update, $\hat{\nabla}$. This approach is lightweight and efficient, particularly for large models like T5-11B. Moreover, KGEditor [175] combines the benefits of memory-based methods and hypernetworks to ensure flexibility and further reduce computation costs. In particular, it introduces an additional feed-forward networks (FFNs) layer for knowledge storage. It then employs a bi-directional LSTM to encode embeddings of triples. In this manner, KGEditor becomes an efficient way to edit knowledge graph embeddings. Despite the success of this paradigm, the editors need to undergo a prior training stage. The availability of training data, including edit samples and pre-training data, poses a critical challenge. While these methods employ synthetic edit samples (e.g., selecting hypotheses via beam search except the top-1 for Question-answering tasks 120), their generalization to realistic mistakes beyond the synthetic sample distribution remains limited.

**Knowledge-based Model Editing**. The knowledge-based model editing paradigm focuses on identifying a subset of parameters specifically associated with particular pieces of knowledge and only updating those parameters. This approach assumes that knowledge is stored within the FFNs, which function as key-value memories [299]. Knowledge Neuron (KN) [202] attributes knowledge parameters using integrated gradients [845], where more salient gradients indicate a greater influence on the knowledge. Building on this idea, Rank-One Model Editing (ROME) [628] uses causal tracing to localize the specific FFN layer whose activation most strongly mediates the recall of a target factual association. Once the target layer is identified, ROME performs a rank-one update to its value projection matrix, effectively replacing the original stored fact with a new subject–object mapping. MEMIT [629] extends ROME by identifying a set of relevant layers (e.g., layers 3–8 for GPT-J) and applying a closed-form multi-layer update. This allows MEMIT to edit multiple facts in parallel while preserving surrounding model behavior. It is important to note that these methods do not establish that these layers are exclusively dedicated to a single piece of knowledge, implying that the layers may be shared across different knowledge domains [282]. To mitigate potential effects on out-of-scope samples, regularization techniques are employed during the neuron/layer updates. For example, MEMIT enforces the model to maintain predictions for several unrelated samples. By adopting a knowledge-based approach, these methods selectively update parameters associated with specific knowledge while minimizing interference with unrelated samples. Based on ROME, BIRD [601] studies the novel problem of Bidirectional Assessment for Knowledge Editing (BAKE), which evaluates the reversibility of edited models in recalling knowledge in the reverse direction of editing and incorporating the bidirectional relationships between subject and object in an edit fact into the updated model weights.

**Model Editing in MLLMs**. Compared to single-modal model editing, the task of editing MLLMs is more challenging due to their inherent diversity and complexity. Specifically, errors in MLLM outputs can be attributed to the synergistic effects of various modalities. A recent study [174] introduces a pioneering benchmark for MLLM editing, named MMEdit. This benchmark evaluates three aforementioned key principles: Reliability, Locality, and Generality, and covers two specific sub-tasks: Editing VQA and Editing Image Captioning. Empirical evidence indicates that while current methodologies [120, 1147, 646] are effective for editing the textual model in MLLMs, they fall short in editing the vision module. Researchers are encouraged to explore innovative techniques for efficient and accurate editing across various modalities and to develop comprehensive benchmarks for evaluating larger MLLMs. The end of 9.4.5 could profit from a discussion on how the different model editing methods differ and what kind of advantages and disadvantages they carry.

Overall, the four paradigms of model editing discussed above differ substantially in their mechanisms and trade-offs:

- *Memory-based methods* store edits externally, avoiding interference with model parameters. They are easy to update and revert but introduce extra retrieval latency and depend on effective memory indexing.
- *Classifier-based methods* preserve original parameters and selectively activate edits only when needed, offering strong locality. However, they rely heavily on a high-quality scope classifier and require ample negative examples to prevent over-triggering.
- *Hypernetwork-based methods* generate parameter updates dynamically from an edit description, enabling flexible and lightweight adaptation. Their main limitations are the need for a pre-trained editor network and reduced scalability to very large models unless combined with low-rank or parameter-efficient techniques.
- *Knowledge-based methods* directly modify internal representations linked to specific facts, often achieving high reliability and generality with minimal changes. Yet, they risk unintended side effects if the targeted layers store multiple pieces of unrelated knowledge, and they require accurate localization of the relevant parameters.

In multimodal settings, these trade-offs can be amplified: memory- and classifier-based methods may generalize more easily across modalities but depend on modality-aware retrieval/classification, while knowledge- and hypernetwork-based methods may offer more precise edits but require sophisticated cross-modal localization strategies. Future research may benefit from hybrid approaches that combine the precision of parameter-based edits with the flexibility and safety of external-memory or classifier gating mechanisms.

## 9.5 Current Limitations and Future Directions

Foundation models, despite their remarkable capabilities, face several challenges when confronting distribution shifts. These limitations primarily stem from inherent difficulties in OOD detection, generalization, and adaptation. Such challenges significantly impact the reliability and robustness of these models in real-world scenarios. When exposed to data that deviates from their training distribution, these models often exhibit decreased performance [1083, 1126], leading to unreliable predictions in dynamic environments where data characteristics frequently change.

While various OOD detection methods have been developed, many struggle with scalability issues, making them less practical for large-scale deployment. Current approaches to OOD generalization and adaption, such as domain adaptation [1083, 446, 1043] and adversarial training [56, 1083, 899], demonstrate varying degrees of success across different domains. These methods often require extensive retraining or fine-tuning to handle new domains effectively, a process that can be both resource-intensive and time-consuming. Furthermore, many techniques for improving OOD robustness heavily depend on the availability of large, high-quality datasets [1083, 1043, 80, 402]. This dependence poses significant challenges in domains where data is scarce or expensive to obtain. Additionally, for multimodal foundation models, effectively integrating and processing diverse data types remains a complex task. Current editing and generalization methods often fall short in scenarios involving multiple modalities, such as text, images, and audio [980]. Last but not least, modern foundation models often undergo continual pre-training and fine-tuning, either horizontally across a sequence of domains or vertically from a general-purpose model to a domain-specific model [799]. As a result, they inevitably tend to suffer from catastrophic forgetting, such as *horizontal forgetting* [799] when continually adapting across domains and *vertical forgetting* [799] when continually adapting from more general models to more domain-specific models.

To address these limitations, future research should focus on developing more lightweight OOD detection and generalization methods. These approaches should aim to identify and mitigate distribution shifts in large-scale settings while maintaining low resource requirements. By focusing on efficiency, such methods could be more readily integrated into practical applications, enhancing the robustness and reliability of foundation model systems across diverse real-world scenarios.

To adapt to rapidly evolving environments, we should prioritize the development of continual or even life-long learning mechanisms for foundation models [1033, 799, 442]. These mechanisms would enable models to adapt to new data distributions without requiring extensive retraining [501] while simultaneously preserving knowledge acquired from previous training data, including data previously used during pre-training or from previous domains. In other words, they should remain robust against both *vertical* and *horizontal forgetting* [799]. This approach could significantly enhance the flexibility and longevity of foundation models in dynamic domains. Additionally, due to the scarcity of data in several domains, developing more data-efficient transfer learning algorithms or creating diverse synthetic data will help models generalize to more practical applications.

To further improve the generality of foundation models, advancing their abilities to handle multimodal data effectively is essential, with unified frameworks that can seamlessly integrate various data types and leveraging techniques like cross-modal learning and multimodal embeddings enhancing performance in complex scenarios [980, 1126, 1065, 1076]. By addressing these limitations and exploring these future directions, we can significantly improve the robustness and reliability of foundation models, ensuring their effective deployment in diverse real-world applications.

# 10    Explainability

There are substantial existing efforts tailored towards the explainability of foundation models, particularly LLMs. In this section, we demonstrate the literature on the explainability of LLMs from the following aspects: (1) Feature Attribution Methods, i.e., Explaining LLMs with the raw features (words, sentences, syntax); (2) Exploring the inherent knowledge incorporated in LLMs themselves; (3) Discovering the roles and training samples in pre-training, fine-tuning, and few-shot learning. Following an overview of the methods used for model explanation, we dive into the evaluations and applications of explainability in LLMs. The discussion then broadens to include multimodal large language models (MLLMs), emphasizing the ongoing efforts in the field. Figure 26 provides a detailed overview of various methods to explain different foundation model components.

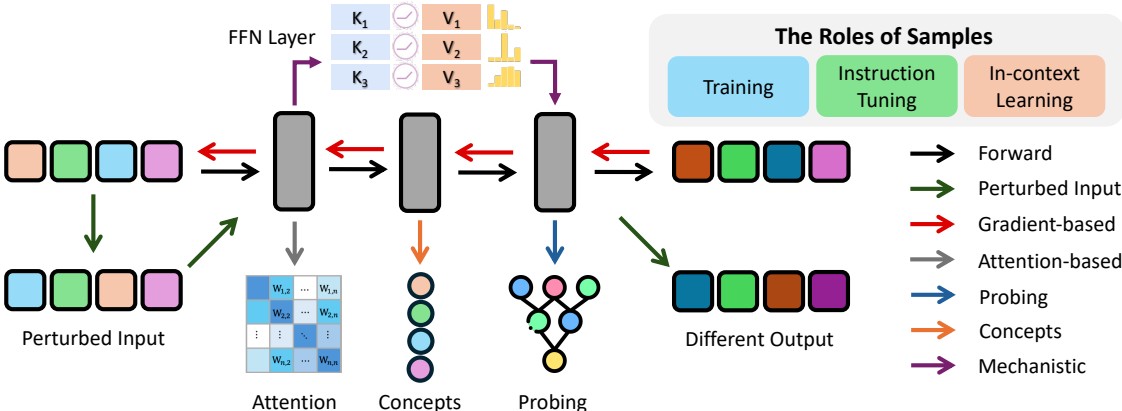

Figure 26: An overview of explainability in foundation models. This figure illustrates various techniques for uncovering how different model inputs and internal components influence model outputs. The right legend highlights the role of samples in different learning stages and the typology of explanation approaches.

## 10.1    Feature Attribution Methods

When adopting LLMs on downstream tasks, it is important to determine which part of words or tokens in the input contribute most to the prediction. Thus, we need to determine the importance of each part of the input, i.e., explaining the prediction using the raw features. To explore this, there are several important lines of work:

### 10.1.1    Perturbing the Input for Explanation

To study the effects of the raw features for model prediction, it has been important to perturb part of the input (a piece of text) while monitoring the model output. With this routine, Perturbed Masking [994] proposes to perturb a token in the given sentence while monitoring the representation of another token. They further propose span-level perturbation to study the impacts of a certain span within the sentence. While Wu et al. [994] regard the monitored variable as the representation of the token or span, MICE [754] study the roles of inputs for model prediction in classification tasks (i.e., the monitoring variable becomes the model prediction). They present a method to find the edits that could flip the model's prediction, where the edits could serve as contrastive explanations. In addition, perturbing the input to shift the label could create counterfactual examples. Crest [879] proposes a framework to first perturb the input sentence with masks and then edit the masked tokens to obtain counterfactuals. Here, perturbing the sentence with masked tokens is essentially extracting rationales as they are both locating the important tokens for model prediction, though finding the rationales could also be achieved by other methods [488, 351]. To provide

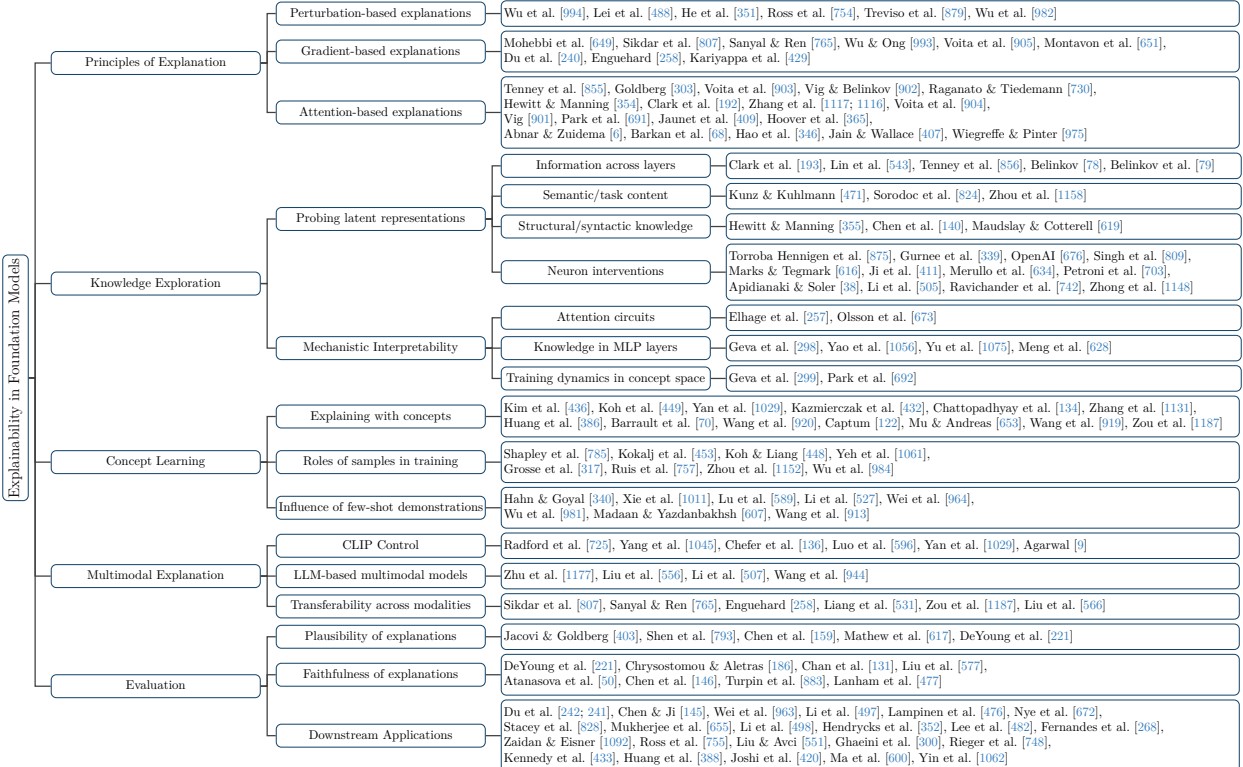

Figure 27: Taxonomy of Explainability in Foundation Models.

more diverse perturbation types and locations, Polyjuice [982] presents a general-purpose counterfactual generator that can generate diverse sets of realistic counterfactuals.

### 10.1.2 Gradient-based Explanation

Mohebbi et al. [649] adopt gradient-based attribution methods to provide token-level attribution scores to understand the representation space of BERT [219] better. More advanced difference-from-reference approaches such as Integrated Gradients (IG) are also used to explain the BERT's prediction [807, 765]. REAT [240] decomposes the final prediction of RNNs directly into the additive contribution of each word in the input text. Voita et al. [905] extend LRP [651] to the Transformers to attribute the relevance score on the source and target contexts in Neural Machine Translation tasks. Wu & Ong [993] analyze different gradient-based methods for explaining BERT classification results. Recent work has extended gradient-based methods to autoregressive decoder-only language models. For example, Enguehard [258] introduce Sequential Integrated Gradients, which computes attributions along the generation path by integrating gradients between an empty sequence baseline and the final generated tokens of GPT-2. Kariyappa et al. [429] propose to approximate token attributions by backpropagating importance through each auto-regressive decoding step, which scales efficiently to long sequence generations.

### 10.1.3 Attention-based Explanation

Previous works suggest that information could be encoded within the heads of the attention weights [855], including abundant information [303, 903, 902, 730, 354, 192, 1117, 1116], which could be used for both input-level explanation and attention heads pruning [904]. Multiple tools are proposed to visualize the attention to illustrate the correlations between words for explanation purposes [901, 691, 409, 365]. Moreover, DeRose et al. [214] propose Attention Flows to visualize the whole attention flow instead of the visualization of one layer. Some methods combine gradients and attention for explanation [68, 346], which generally perform better than using attention alone. Abnar & Zuidema [6] treat self-attention as a flow network across layers to

enable post-hoc computation of token-to-token information propagation, which shows higher correlation with gradient-based and ablation-based importance scores compared to raw attention. Though attention scores could be used to understand the large language models, they may not necessarily be capable of identifying the explanations [407].

## 10.2 Exploring the Knowledge in LLMs

Instead of explaining LLMs by highlighting the important tokens or spans in the input, interest increasingly gravitates toward understanding the breadth of knowledge encapsulated by these models. Several key areas of investigation are outlined as follows:

### 10.2.1 Probing the Representations within LLMs

Probing the model could help us understand deep neural networks [78]. Early work by Veldhoen et al. [897] introduced diagnostic classifiers for revealing how neural networks process hierarchical structure by training simple probes on latent representations to test hypotheses about compositional strategies. With such probing philosophies, recent transformer-based works investigate the embeddings and hidden states from various mechanistic components of the network. Kunz & Kuhlmann [471] show that the token embeddings learned by BERT and ELMo contain rich information about the exact linear context of the token. Belinkov et al. [79] interpret the representation of different layers in NMT encoders, finding that higher layers have more semantic information. In contrast, lower-layer representations tend to be more suitable for part-of-speech tagging. The fact that language models can capture semantic information and conduct arithmetic operations is also studied in Sorodoc et al. [824] and Zhou et al. [1158]. Similarly, Clark et al. [193], Lin et al. [543] show that Bert's representations encode surface and positional information in the lower layers, but more semantic features in higher layers, while Hewitt & Manning [355] propose a structural probe showing the syntax trees are embedded in a linear transformation of ELMo and Bert's word representation space. Building on previous probing work, Tenney et al. [856] probes word-level contextual representations to investigate how they encode sentence structures. Different from the above methods paying attention to the representation in certain metric spaces (typically Euclidean space), Chen et al. [140] consider the probing methods in hyperbolic space, which could better recover tree structures. While these methods could reveal the ability of representations to encode syntactic information, Maudslay & Cotterell [619] show that syntactic probes may not properly isolate syntax. With a new corpus that is semantically nonsensical but syntactically well-formed, it is shown that syntactic and semantic information are entangled. Further, Zhang et al. [1111] argue that even with the existing works, it remains unclear whether LLMs have understood linguistic knowledge. Thus they probe GPT-3 to show that it has acquired linguistic knowledge in most cases but may still fail when disturbances happen. Apart from exploring the representations, some other works focus on self-attention heads, which could be helpful for heads pruning [457, 193].

Some methods are designed to be used during the inference of LLMs without training the classifier on the hidden vectors, such as cloze completion or text generation [703, 38, 505, 742]. Though prompts can be designed to reveal the abilities of the LLMs, Zhong et al. [1148] question if the prompt-search methods also learn from the training data, i.e., the training data may contain certain regularities of the underlying fact distribution that could be exploited.

Probing methods are also used to understand the roles of neurons in LLMs. Torroba Hennigen et al. [875] propose a framework based on a decomposable multivariate Gaussian probe to explore how linguistic information is structured within the representation, showing that most attributes are reliably encoded by only a few neurons. Moreover, some methods propose to probe the internal activations to predict the presence of features in the input, showing the sparse combinations of neurons can represent many features [339]. Recently, OpenAI has shown the possibility of using an advanced LLM (e.g., GPT-4) to explain the neurons in a small model (e.g., GPT-2) [676]. Summarize and Score (SASC) [809] proposes to generate candidate explanations to explain the modules from LLMs, which could be more efficient than explaining single neurons. Marks & Tegmark [616] reveal that LLM representations of true/false statements form distinct linear directions that can be identified via probing and causally intervened upon to flip model outputs. Ji et al. [411] show that identifying a linear "verbal uncertainty feature" in LLM representations can be manipulated at inference

time to reduce hallucinations. Merullo et al. [634] investigate how pretraining data frequency influences the emergence of linear representations of factual relations and find strong correlations between term co-occurrence counts and probe performance across models.

### 10.2.2 Explaining LLMs with Concepts

Concept-based explanation refers to mapping the input into concepts and then using a linear classifier to predict the final class with the mapped concepts. As the prediction from the concept to the class is a simple linear classifier, it has the property of explainability even though the mapping from the input to the concepts is not explainable. Pioneering methods in this direction include Concept Activation Vectors (CAVs) [436] and Concept Bottleneck Models [449]. Such a concept-driven framework is widely adopted in visual representation learning where the images are first mapped to the concept space, based on which the classifier makes the decision [436, 449, 1029, 432, 134, 1131, 386]. More recently, Wang et al. [920] propose Probabilistic Conceptual Explainers (PACEs), drawing inspiration from hierarchical Bayesian deep learning [915, 916, 419] and topic models [95] to provide concept-based explanations at multiple levels (e.g., datasets, images, and patches) to address key concerns in model interpretation such as faithfulness, stability, and parsimony.

Beyond computer vision, CAVs are also tailored to language models for sentiment classification tasks [122], featuring two concepts: Positive Adjectives and Neutral. Besides, while Captum [122] define concepts manually, Mu & Andreas [653] propose to learn the abstractions by analyzing the neurons, where they find that neurons learn shallow lexical heuristics from dataset biases. Wang et al. [919] propose Variational Language Concepts (VALCs) to learn the concept-based explanations in an unsupervised learning manner while enabling neuron editing in the concept space. Turner et al. [881] propose to steer the behaviors of language models by curating concept activations and injecting them in the model's hidden layers, and Zou et al. [1187] propose a unified paradigm for concept interventions in the activation space. Barrault et al. [70] propose Large Concept Model (LCM) to perform next-sentence-prediction-based autoregressive learning in the conceptual embedding space. In summary, developing concept representation is a crucial step towards interpretable LLMs for diverse tasks. Such interpretability offers a feasible solution for diagnosing, revising, and intervening LLMs.

### 10.2.3 Mechanistic Interpretability

In each block of a transformer, the self-attention layer projects input tokens or hidden states into query, key, and value vectors that effectively store key–value memories. The main focus of mechanistic interpretability is reverse engineering for retrieving contextual information across those vectors, tokens, and layers, which provides a systematic approach to explaining LLMs [257]. The aforementioned study also finds that in-context learning in small models could be explained by specific attention heads, termed "Induction Heads". This mechanism is hypothesized to constitute the mechanism for most "in-context learning" in large transformer models [673]. Another line of work focuses on FFN layers. Earlier work [298] argue that FFN layers contain most of the information that operates as key-value memories, and more recent works [1056, 1075] propose to search neural circuits or salient neurons for parametric knowledge representation. With the localization of the information, we could perform model editing on the relevant matrices in FFN layers [628]. In addition, Geva et al. [299] and Park et al. [692] analyze the learning dynamics of generative models in the concept space, demonstrating that updates can be decomposed into sub-updates, where each sub-update corresponds to human-interpretable concepts. Hou et al. [368] introduces MechanisticProbe to explore a mechanistic interpretation of LLMs for multi-step reasoning tasks.

### 10.3 Discovering the Roles of Samples in Training, Fine-tuning, and Few-shot Learning

The development of foundation models encompasses multiple learning stages, including pre-training, fine-tuning, and few-shot learning. During different stages, samples play distinct roles as illustrated in Figure 28.

**a) Pre-training Stage**

A large language model (LLM) is a computational model capable of language generation or other natural language processing tasks. As language models ...

<html>\n<head>\n <title>Delicious Pasta Recipe</title> <meta name="description" content="Pasta recipe for a quick meal"> </head>\n <body> ....

**b) Instruction-tuning Stage**

You are a scientist who just invented a time machine. Where do you travel first? As a scientist, I would travel back in time to 1969 and witness the Apollo 11 mission.

Apollo 11 was a spaceflight conducted by the United States from July 16 to July 24, 1969. It marked the first time in history that humans landed on the Moon.

**c) In-context Learning Stage**

Example1: If there are 3 cars in the parking lot and 2 more cars arrive, how many cars are in the parking lot? There are originally 3 cars. 2 more cars arrive. 3 + 2 = 5.

Example1: If there are 3 cars in the parking lot and 2 more cars arrive, how many cars are in the parking lot? The answer is 5. Example2: ...

Figure 28: The influence of samples in pre-training, instruction-tuning, and in-context learning stages. We highlight the beneficial and detrimental textual fragments in green and red, respectively.

### 10.3.1 Influence of Single Example in Training

There is a growing body of work studying the effects of one single example in the training process. SHAP [785] first proposes Shapley values to allocate the contribution of one single player in a coalitional game. TransSHAP [453] proposes to adapt SHAP to transformers models, Bert specifically, to explain the classification results. Other works measure the effects of the example with the influence of this example on test loss values [1061]. Influence function, as a statistical technique adapted to deep neural networks by Koh & Liang [448], approximates how upweighting a single training example would change model parameters and test loss. Recently, Grosse et al. [317] scale the influence functions on LLMs with up to 52 billion parameters, and Ruis et al. [757] leverage influence-based analysis to demonstrate that procedural knowledge in the LLM pretraining sequences drives the emergence of reasoning capabilities.

### 10.3.2 Influence of Training Stages

The training stages in the current most powerful models include pre-training and instruction tuning. LIMA [1152] analyzes the relative importance of pre-training and instruction-tuning, hypothesizing that the knowledge revealed in the generation primarily comes from the pre-training stage, while instruction-tuning tends to fixate on the style and format of interacting with users, which is tested by using only 1000 curated examples to train Llama-65B to achieve near-GPT-4 performance on a controlled human study. Wu et al. [984] explore the instruction recognition and knowledge evolution before and after instruction-tuning, demonstrating that instruction-tuning could better identify the instruction parts from the input and align the knowledge with the user instruction.

### 10.3.3 Influence of Samples in Few-shot Learning

Few-shot learning in LLMs typically refers to in-context learning (ICL). Li et al. [527] investigate ICL's functionality using contrastive demonstration and saliency maps. Wei et al. [964] examine how specific examples influence learning outcomes in few-shot scenarios, employing two distinct approaches: ICL with intentionally incorrect labels, and ICL with semantically unrelated labels. They found that large models can better override the input-label mapping learned during the pre-training stage, and small models rely more on semantic priors than large models do. In both settings, they find larger models and those enhanced with In-Context Fine-Tuning perform better. Wu et al. [981] focus on how Chain-of-Thoughts (CoT) affects the model behavior, while others try to perturb CoT demonstrations and check the effects on the outcome [607, 913]. Hahn & Goyal [340] propose a theory of emergent in-context learning as implicit structure induction to show how compositional structure in pretraining data gives rise to ICL. Xie et al. [1011] provide a complementary perspective by modeling ICL as implicit Bayesian inference where LLMs infer latent concepts shared across prompt examples. From the empirical perspective, Lu et al. [589] conduct large-scale evaluations of emergent abilities across multiple tasks to conclude that the emergent performance gains can be largely ascribed to ICL mechanisms and parametric knowledge.

## 10.4 Evaluation of Explainability

The evaluation of explainability usually focuses on two perspectives [1136]: (1) Plausibility (also known as persuasiveness by 403). A plausible explanation seems logical and coherent to the audience, regardless of whether it is correct or accurately reflects the model's reasoning process. Essentially, plausibility measures the quality of the explanation in terms of its persuasiveness and understandability from a human perspective. (2) Faithfulness. A faithful explanation accurately represents the internal workings and decision-making processes of the LLM, demonstrating how well the explanation aligns with what the model is doing when generating the response.

### 10.4.1 Evaluation of Plausibility

To evaluate the plausibility of the explanations of pre-trained LMs, Shen et al. [793] propose a benchmark to test LMs abilities in five dimensions: grammar, semantics, knowledge, reasoning, and computation. Another benchmark, HateExplain [617], asks the annotators to highlight part of the text that could justify their decisions, which could serve as the ground-truth explanations. With these ground-truth tokens, we could calculate the metrics such as Accuracy, Macro F1-score, AUROC score [617], and AUPRC (Area Under the Precision-Recall Curve), IOU (Intersection-Over-Union) [221], etc.

The above metrics could be applied to the explanations with raw features (discussed in Sec 10.1), but they may not be suitable for the explanations based on natural language (Sec 10.2.1) as there would be no ground-truth explanations in this case. To resolve this issue, Chen et al. [159] propose to evaluate the counterfactual simulatability of natural language explanations, i.e., whether humans could predict the model's behavior according to the explanations given by the model. If so, then we say LLMs could explain themselves.

### 10.4.2 Evaluation of Faithfulness

To evaluate the faithfulness of rationales selected by the model, ERASER [221] proposes the following metrics: (1) Comprehensiveness, which refers to the probability change of the original predicted class before and after the removal of the predicted rationales, and (2) Sufficiency, which means how much the extracted rationales could support the model to make a prediction. Ideally, the objective is to achieve maximal change of comprehensiveness without compromising the accuracy of predictions when relying solely on the extracted rationales. In addition, TaSc [186] proposes another line of metrics: (1) Decision Flip - Fraction Of Tokens (DFFOT), which measures the fraction of important tokens required to be removed to cause a decision flip. A lower DFFOT indicates a more faithful explanation. (2) Decision Flip - Most Informative Token (DFMIT), where the rate of decision flips caused by removing the most influential tokens is reported for comparison. To further evaluate the faithfulness of the explanations, Liu et al. [577] propose a faithfulness violation test, showing that most methods are hindered by the faithfulness violation issue. Although these metrics each have their rationale and applicability, the consistency between these metrics remains questionable. Chan et al. [131] show that the explanations that achieve the best DFFOT may have the worst Sufficiency score. These metrics are also not suitable for natural language explanations. To solve this issue, for classification tasks, Atanasova et al. [50] propose two tests: (1) counterfactual input editor for inserting reasons leading to counterfactual predictions; (2) reconstruct inputs from the reasons given by the explanation models and check if they lead to the same predictions. Different from modifying the input and monitoring the output (basically perturbation), REV [146] quantifies the amount of new, label-relevant information in the explanations beyond the information within the input, which can give the measurement without perturbation. For CoT-style explanations, Turpin et al. [883] find that CoT explanations could be vulnerable towards biasing features in the model inputs, thus being systematically unfaithful. Lanham et al. [477] monitor how the model predictions change when the input is intervened. They argue that models may produce less faithful reasoning when the models become larger. These researches demonstrate the need for better explanations in the CoT style.

## 10.5    Applications of Explainability

Gaining the explainability of LLMs has various applications, including diagnosing the model, and improving the model, which can help obtain user trust in the model.

### 10.5.1    Avoiding Shortcut Learning

Du et al. [242; 241] point out that LLMs may rely on shortcut features like data biases, and artifacts to make predictions rather than understanding the meaning, which demonstrates an important challenge in the field of LLMs. Chen & Ji [145] propose to utilize the explanations revealed by the model to determine if the model is robust or not, as they argue that a robust model should behave consistently between original and adversarial example pairs. Wei et al. [963] use chain-of-thought to understand the reasoning process of the model, though the faithfulness needs further exploration [883, 477]. Li et al. [497] design an urban-environment multi-agent simulator based on customizable first-order logic to evaluate the logical reasoning capability of LLMs. These works have identified new challenges in LLMs.

### 10.5.2    Improving Model Performances

Apart from understanding the model, other works try to improve the model with explanations, which can also help gain user trust. For in-context learning, Lampinen et al. [476] find that using explanations in the prompts can improve performances, and hand-tuned explanations on a small validation set could even offer substantial improvements. Zhou et al. [1162] use LLM interpretation to improve the performance of LLMs on natural language understanding. To improve the model's reasoning ability, Nye et al. [672] find that asking LLMs to emit intermediate computation steps into a "scratchpad" helps with multi-step reasoning tasks. On the other hand, Turpin et al. [883] demonstrate that chain-of-thought explanations can be systematically unfaithful. LLM may produce plausible but misleading rationales, which cautions against over-reliance on CoT for performance gains. Besides the training-free prompting, Stacey et al. [828] supervise the model's attention weights to encourage the model to pay more attention to the words that are present in the explanations, which significantly improves the model performance. Li et al. [498] proposes the first bilevel planning framework to learn neural predicates from in-context demonstrations to achieve compositional generalization for new tasks. In the large language model regime, Mukherjee et al. [655] propose to train a 13B model with the explanations and reasoning processes provided by GPT4 to improve the reasoning ability of small models. Early works by Hendrycks et al. [352] show that structured human feedback can steer language model behaviors toward alignment goals. Lee et al. [482] introduce the XMD framework which shows humans the explanations of model behavior and also updates the model based on the user feedback. Then to improve the model's OOD generalization ability, there is a popular paradigm called Explanation Regularization (ER) which aims to align the model rationales with human-annotated rationales [551, 748, 1092, 300, 388, 755, 433], where the effects of ER on OOD generalization is evaluated by ER-TEST [420]. As these methods require human-annotated rationales, which might be exhaustive, AMPLIFY [600] proposes to automate the process of rationale generation with the insights from post hoc explanations to provide corrective signals to LLMs. Some other applications include identifying the important instructions to compress the instruction [1062]. Fernandes et al. [268] comprehensively summarize a taxonomy of integrating human feedback into natural language generation systems for improving generation quality and explaining model decisions.

## 10.6    Explainability of MLLMs

Existing works on the explainability of MLLMs primarily focus on CLIP-based image-text alignment models. Early research suggests querying GPT to augment class labels, thereby improving the zero-shot performances of CLIP [725]. Recent methods utilize CLIP to analyze the composition of images with textual concepts, wherein the concepts are further used for image classification [135, 1045] or editing [136, 596]. Such approaches offer additional interpretability [1029, 1045] and controllability compared with directly using the CLIP representation for class prediction. Furthermore, Agarwal [9] investigates the trustworthiness of explanations generated for zero-shot and fine-tuned Vision and Language Models (VLMs), revealing that explanations for zero-shot CLIP classifiers are more faithful than those of the fine-tuned versions. While

these works concentrate on image-text alignment models, the explainability of LLM-based image/video understanding models, such as MiniGPT-4 [1177], LLaVA [556], VideoChat [507], and LVChat [944], remains underexplored.

## 10.7 Current Limitations and Future Directions

Despite significant advancements in the field of explainability, several limitations persist that necessitate future attention:

### 10.7.1 Faithfulness of Raw Features

Current methods, ranging from input perturbation to gradient-based and attention-based techniques, offer explanations for model predictions. However, the faithfulness of these explanations remains questionable. As illustrated in Section 10.4.2, the metrics from different perspectives may vary drastically [131]. This issue extends to downstream tasks where the model may rely on various biases for predictions [351, 242], potentially compromising their generalization ability if the predictions are not truly faithful.

### 10.7.2 Understanding How LLMs Store Knowledge

Research focusing on the knowledge stored within LLMs – examining layer representations [79, 471, 732] and analyzing generated content [562, 22] – has shed some light on the distribution of syntactic versus semantic information across layers. While these works provide the insight that lower layers encode syntactic information and higher layers possess semantic knowledge, the community is still actively discussing how knowledge is dynamically injected into the model during training and how the injected knowledge is triggered through the inference process. For knowledge injection, Müller-Eberstein et al. [661] trace representational subspaces during pre-training to capture phases when syntactic subspaces rapidly emerge and disentangle from semantic and reasoning subspaces. Hu et al. [372] fit hidden Markov models to training-time metrics (e.g., weight norms, variances) to derive latent states of learning dynamics, which reveal phase transitions during the training. Wal et al. [908] deploy multiple LLM pre-training runs to demonstrate that training dynamics and knowledge injection stabilize consistently with identifiable outlier behaviors. On the other hand, how to trigger relevant knowledge with the input demonstrations needs further understanding [557, 155]. Some works argue that the knowledge is mainly stored in MLP layers [628, 629]; however, it is shown in their papers that attention layers also have slight effects when predicting the facts, especially in earlier layers (See Figure 3 in [629]). Even in the work that explicitly stores knowledge in a memory module [943], how the model processes the knowledge is under-explored.

### 10.7.3 Transferability of Explanation Across Different Modalities

How to develop unified explanation frameworks that can inherently transfer across modalities has been an emerging topic. As we have discussed above, prior works on Integrated Gradients [807, 765, 258] are modality-agnostic and can be employed on image classification, text generation, and multimodal VQA tasks. Yet these gradient-based attributions suffer from superficial faithfulness and low robustness to input perturbations [8]. On another track, attention-based methods such as MultiViz [531] can provide visualizations of cross-modal attention and thus enable analysis of pixel-to-token flow in multimodal transformers. However, the credibility of attention as explanation has long been arguable in the community [407, 975]. Concept interpretation tools such as representation engineering for LLMs [1187] have seen the potential to be adapted to vision-language modalities [566]. Aggregating multiple explanation levels and addressing the aforementioned limitations will motivate future work in developing robust, faithful, and transferable explanations for multiple modalities. It is also increasingly necessary to develop standardized benchmarks for cross-modal explanations that promote both interpretability and fidelity.

### 10.7.4 Reliability and Responsibility of Foundation Models from the Explainability Perspective

Without a deep comprehension of foundation models, ensuring their reliability and responsibility is challenging, where explainability has the potential to offer a pathway to address these issues. For instance,

identifying the biases in pertaining data and implementing various de-biasing strategies could pose more equitable models [522]. Moreover, understanding how the model stores knowledge [533] can facilitate model editing with the current knowledge and the unlearning of harmful information, achieving up-to-date and safer foundation models [628, 927, 1104]. However, the effectiveness of interpretability methods themselves must be critically assessed to ensure they provide meaningful insights. Adebayo et al. [8] perform sanity checks for saliency maps and reveal that some widely used saliency methods are independent of both the model and the data, questioning their validity in explaining model behavior. Similarly, Alvarez-Melis & Jaakkola [24] investigates the robustness of interpretability methods and demonstrates that small perturbations to the input can significantly alter the explanations provided, highlighting the need for more robust interpretability techniques. Moreover, understanding how practitioners use interpretability tools is also crucial. Kaur et al. [431] explore data scientists' use of interpretability tools and find that mismatches between tool capabilities and user needs can limit their effectiveness in ensuring model reliability and responsibility. They emphasize the importance of designing interpretability tools that align with the practical requirements of users. As the development of increasingly powerful foundation models continues, focusing on both the advancement and the critical evaluation of explainability methods cannot be overstated.

## 11   AIGC Detection

The advent of foundation models has led to a surge of artificial intelligence-generated content (AIGC) across various modalities, including text [309, 676, 311], images [737, 1112, 259], audio [458, 337, 387, 27], and video [454, 94, 65]. While these technologies have unlocked many useful applications, they also pose significant challenges, particularly in terms of content authenticity [320, 523, 363]. The capacity of foundation models to generate human-like content can be exploited for malicious purposes, including the dissemination of misinformation and identity theft. Consequently, the demand for research focusing on detecting AIGC is on the rise. This section provides a comprehensive overview of current methodologies and techniques for AIGC detection, highlighting the pivotal role this field plays in preserving the integrity of digital information in an era increasingly dominated by foundation models and AI technologies.

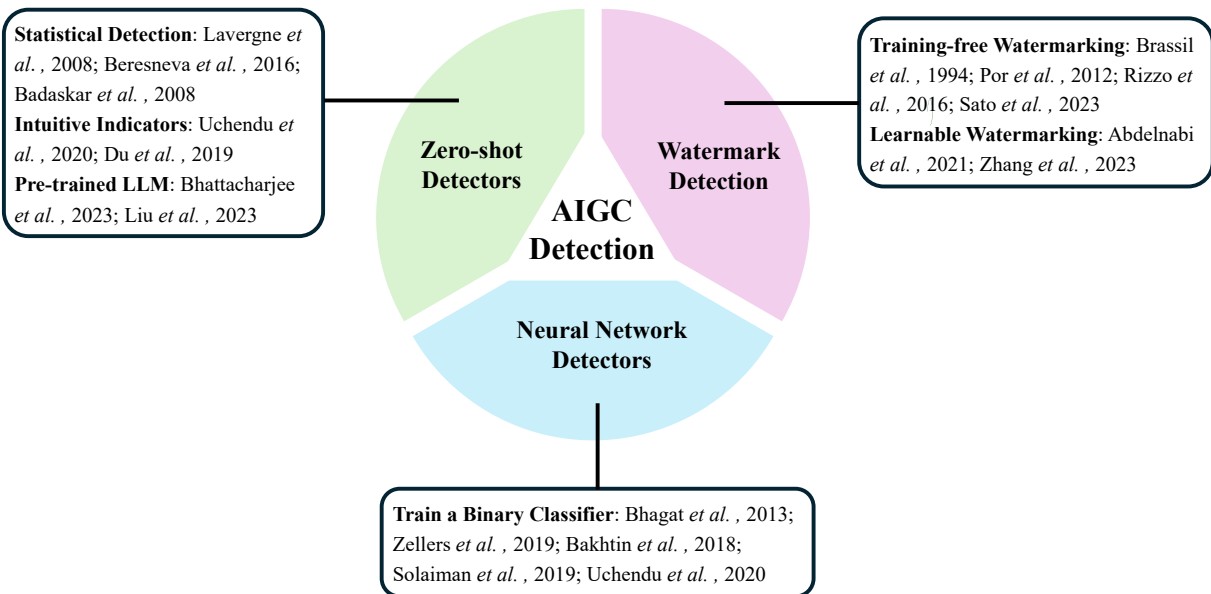

Figure 29: An overview of AIGC detection techniques. We group them into three categories: zero-shot detectors, watermark-based detection, and neural network detectors, each with further subdivisions.

### 11.1   The AIGC Detection Problem

The task of AIGC detection can be seen as a binary classification problem. In general, we aim to determine whether a given input $x \in \mathcal{X}$, such as an image, text, or audio, is generated by AI models. This can be achieved using a detector $D : \mathcal{X} \to \{0, 1\}$, which can be defined as follows:

$$D(x) = \begin{cases} 1 & \text{if } x \text{ is generated by AI or is partially generated by AI.} \\ 0 & \text{if } x \text{ is created by a human.} \end{cases} \tag{14}$$

The detector $D$ can be broadly categorized into the following types: (i) zero-shot detectors, (ii) watermark detectors, and (iii) learnable detectors. Furthermore, we summarize the representative work for all types in Figure 29, with examples for each type illustrated in Figure 31.

### 11.2   Zero-shot Detectors

The fundamental concept behind zero-shot detectors is to differentiate between AI- and human-generated content based on their intrinsic distinctions, such as the frequency of word occurrence in the generated text,

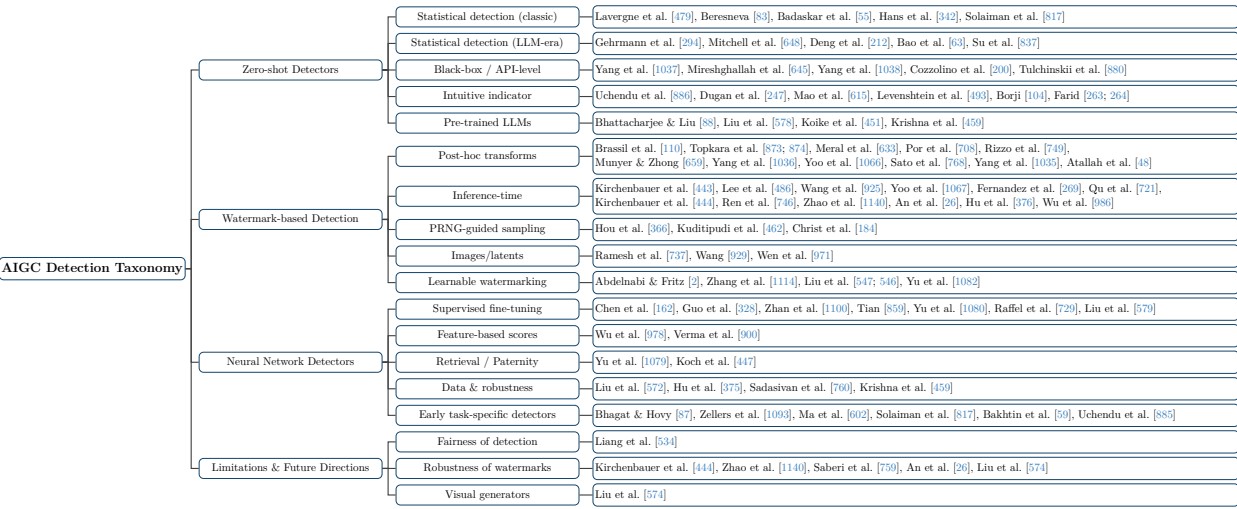

Figure 30: Taxonomy of AIGC detection in Foundation Models.

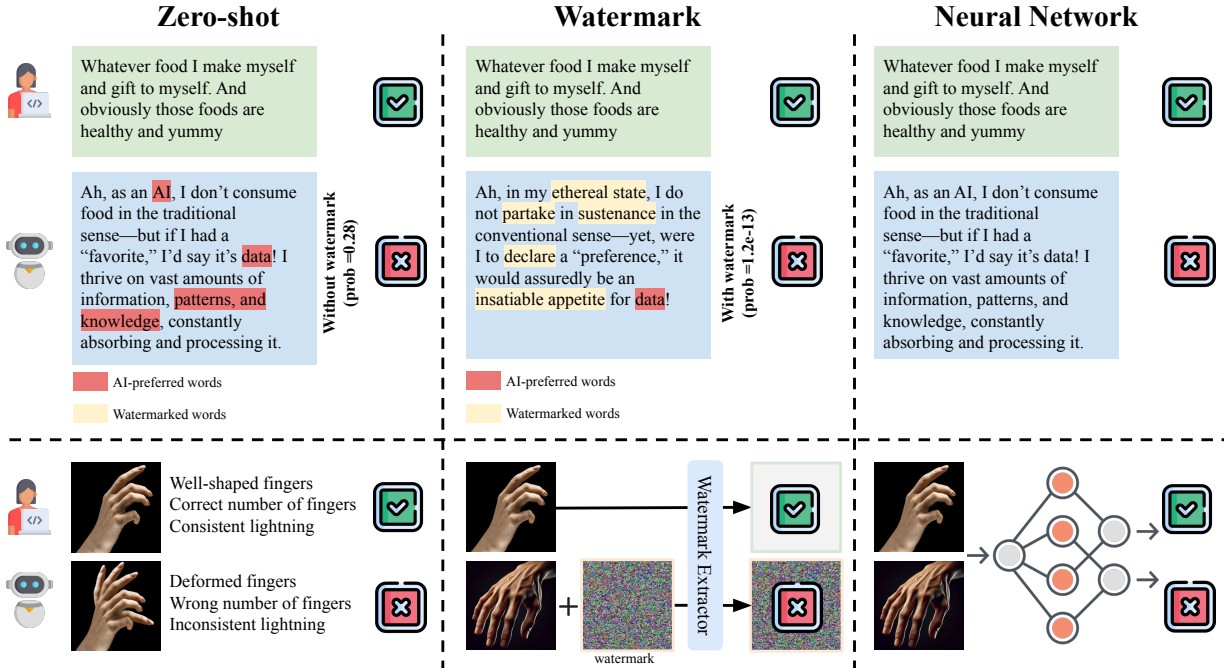

Figure 31: Examples of zero-shot, watermark, and neural network detectors for textual and visual inputs.

which can be identified and flagged by hand-crafted detectors. That said, zero-shot detectors are arguably the simplest to deploy since they do not require additional training of both the detectors and the foundation models that generate the content.

### 11.2.1 Statistical Detection

These detectors assume full, or at least partial (e.g., the token logits during generation), access to the foundation model that generated the content. In the text domain, traditional methods usually rely on statistical outlier detection based on different metrics, including entropy [479], perplexity [83], n-gram frequencies [55], the ratio of perplexity to cross-perplexity [342], which measures how surprising the next token predictions of

one model are to another model, and average per-token log probability [817]. We use them to evaluate the given text passage and apply thresholding to assess whether the content is likely AI-generated. However, these approaches are inadequate in the era of foundation models, where AI-generated content becomes more diverse and of high quality.

To this end, several recent studies improve upon these simple ideas and extend them to LLMs. [294] propose GLTR, which is centered on the underlying assumption that LLMs overgenerate from a limited subset of the true distribution of natural language, for which they have high confidence. This property is detected by computing, for each token in a text sequence: (i) the probability of generating the token, (ii) the rank of the word, and (iii) the entropy of the generated distribution. These metrics are then compared against those of human writers. In a similar vein, DetectGPT [648] leverages the empirical observation that AI-generated text tends to lie in negative curvature of the model's log probability function, i.e., the sequence sits at a locally concave region of $f(x) = \log p_\theta(x)$ such that small random paraphrases/perturbations $y \sim q(\cdot|x)$ systematically *decrease* $f$. Practically, DetectGPT scores a text by the average log-likelihood drop $\Delta = f(x) - \mathbb{E}_{y \sim q(\cdot|x)}[f(y)]$. A second-order Taylor expansion gives $\mathbb{E}[f(y)] \approx f(x) + \frac{1}{2}\mathrm{tr}(H_f(x)\,\mathrm{Cov}_q)$, so $\Delta > 0$ implies $\mathrm{tr}(H_f(x)\,\mathrm{Cov}_q) < 0$ (negative curvature) along the perturbation directions—an effect pronounced for model-generated text but weaker/inconsistent for human-written text. This observation led to follow-up investigations on improving detection efficiency [212] and utilizing conditional probability curvature [63]. DetectLLM [837] employs a similar principle, but scores with log-rank information. However, these approaches rely on thresholding the probability of a given sequence, which requires access to the model's token generation probability distribution. Such a requirement can be too restrictive in many practical scenarios.

To alleviate this, recent detection methods that require only API-level access to the unknown source model are proposed. For instance, [1037] utilize the N-Grad divergence between re-prompted and original text to identify AI-generated content in the biology domain. Additionally, recent research has shown that smaller surrogate models can serve as effective proxies for AIGC detection [645, 1038, 200]. By observing that AI-generated text exhibits lower intrinsic dimensionality compared to human-written text when measured in a representation space of fixed text embeddings (e.g., sentence- or token-level vectors produced by a pretrained encoder), [880] propose to employ a persistence-homology-based intrinsic dimension estimator (PHD) to exploit this property for AIGC detection, estimating local manifold dimension from neighborhoods within the embedding space. This approach does not require API-level access—i.e., it operates in a complete black-box setting.

### 11.2.2 Intuitive Indicators

These methods use the analytical abilities of humans to identify inconsistencies with prior knowledge in AIGC, thus achieving detection. As a result, these methods provide notable interpretability and credibility in the detection process.

For AI-generated text, [886] note that a lack of coherence and consistency serves as a strong indicator of AIGC, and emphasize the importance of collaboration among human detectors in improving detection accuracy. Similarly, [247] note the unreliability of relying solely on grammatical errors as a detection strategy. They further showcase that while LLMs frequently commit factual and logical errors, these mistakes are often overlooked by neural network-based detectors but are easily noticed by human detectors. More recently, [615] find that LLMs exhibit a greater propensity to alter human-written text compared to AI-generated text when tasked with rewriting. This tendency stems from LLMs' perception of AI-generated text as being of high quality, which results in fewer modifications. They then proposed "geneRative AI Detection viA Rewriting" (RAIDAR) to detect AI-generated content by instructing LLMs to rewrite text and then calculating the edit distance of the output by the Levenshtein Score [493].

In vision, the detection of AI-generated images can be done by examining inconsistency with reality. Numerous studies [104, 263] note that AI-generated images often violate physical rules in the real world, such as missing or unnatural reflections and shadows of objects that are inconsistent with natural lighting and environment. In addition, [264] has noticed that AI-generated images exhibit inconsistency in perspective, such as parallel lines cannot converge at a common vanishing point. For facial images, [104] outlines key cues

for detecting AI-generated faces, including symmetry, iris color, pupil shapes, skin, etc., where the generated images tend to depict physiological falsehood.

However, AIGC detection by intuitive indicators are becoming much harder as the capabilities of AIGC models continually improve.

### 11.2.3 Pre-trained LLMs

Without training, a few studies have investigated the use of pre-trained LLMs to directly identify generated texts either by themselves or by other LLMs. However, it has been observed that the performance of these detection methods is often inferior to statistical and neural network approaches. For example, [88, 578] formulate the AIGC detection task in a question-and-answer format, and prompt LLMs with the question to obtain an answer for detection. [88] note that neither ChatGPT nor GPT-4 could reliably identify text generated by various LLMs, while [578] reveal the poor zero-shot performance of GPT-3.5-turbo in AIGC detection which is close to random guessing.

A recent work [451] considers employing in-context-learning (ICL) with pre-trained LLMs for AIGC detection, in which a few labeled examples (context) are integrated into the question prompt as a single input to the model, thereby facilitating the learning of new tasks in context. The results in [451] show that using ICL outperforms both traditional zero-shot methods and RoBERTa-based detectors, however, [578] observe no significant improvement in using ICL with GPT-3.5-turbo. It is worth noting that while ICL methods are not strictly zero-shot, they do not require additional training of the detectors. Another recent work [459] proposes a detection mechanism based on retrieval, which involves creating a database of generated text and comparing the semantic similarity of the target text with all the text stored in the database to perform detection. Although this approach is effective and robust against paraphrasing, its requirement of storing LLMs generation may raise privacy concerns.

## 11.3 Watermark-based Detection

Watermarking injects algorithmically detectable patterns into the AI-generated content while ideally preserving the quality and diversity of AIGC. A watermarking algorithm for AI-generated content detection typically involves three components:

- The *watermark* or message, denoted as $m$, can be represented as a bit-string in the generated images or as a specific occurrence of words in the generated text. From now on, the term "watermark payload" will be used to refer to the amount of information conveyed by the watermark message.
- An *encoder*, denoted as $A$, is responsible for embedding the watermark message $m$ into an AI-generated content $x$, thereby transforming it into a watermarked content $\tilde{x}$.
- A *detector*, denoted as $D$, is capable of determining the presence of a watermark in either $\tilde{x}$ or $x$, provided that the content is generated by AI.

In *zero-bit watermarking*, the embedded message $m$ only signifies the presence or absence of a watermark, hence is only used to indicate whether $x$ is generated by AI; whereas in *multi-bit watermarking*, the embedded message $m$ can carry additional detailed, customized information, e.g., the name of the AI model or authorship attribution. We will primarily focus on the first case - using watermarking for AIGC detection.

A watermarking algorithm that is effective for detecting AI-generated content should possess the following key properties:

- It should be algorithmically easy to verify yet remain imperceptible to humans, where ease of verification can refer to the ability to open-sourcing, or a high success rate for detection.
- It should have minimal impact on the quality of AI-generated content. This means that foundation models incorporating the watermark algorithm, potentially during training, should still produce content of similar quality compared to the non-watermarked version.

- It should exhibit high robustness to attacks aimed at removing the watermark or applying semantically invariant transformations to AI-generated content with watermarks. These transformations can range from rephrasing generated text to distorting watermarked images.
- It should demand minimal effort to incorporate the watermark into AI-generated content.

### 11.3.1 Training-free Watermarking

In training-free watermarking algorithms, the watermark, encoding, and decoding algorithms are all designed based on heuristics, exploiting domain-specific characteristics of the generated content rather than learned through end-to-end training.

Several studies apply various kinds of semantically-invariant transformation directly to *existing* AI-generated text. These include visually imperceptible reformatting such as adding whitespace characters and replacing characters with similar ones in appearance but with a different Unicode representation [110, 708, 749, 768]; lexical-based modifications such as synonym substitution [659, 874, 1036, 1066, 1035]; syntax-based manipulation which alters the arrangement of words and phrases in the text through several predefined types of transformations [48, 633, 873]. Each distinct type of transformation corresponds to a specific message bit, therefore allowing the detection and extraction of watermarks. The immediate advantage of these approaches is that they do not require knowing the identity (i.e., the name of the model) or access to the AI models that generated the content. However, since these methods largely rely on simple semantically invariant transformation, they are easy to spot and hence are vulnerable to watermark attack or removal. Moreover, these manually defined modifications can create abrupt and unnatural modifications to the original text, hence significantly degrading the quality of the generated content.

Instead of encoding watermarks in the existing context *after* generation, it is also possible to encode training-free-based watermarks *during* the content generation process without the need for re-training the models. Consequently, unlike previous approaches discussed, the following methods assume at least the given access to controlling the generation process of the foundation models. The pioneering research of [443] first proposes a watermarking framework for LLMs by altering the *logits* for token sampling in a text sequence generation. The algorithm [443] works by selecting a randomized set of "green" tokens before generation, and then softly promoting the use of "green" tokens during generation by adding a small bias on the sampling logits of "green" tokens. Detection can be achieved by deploying statistical tests which are essentially based on identifying the unnatural occurrence of "green" tokens in the writing. Follow-up research works expand upon this idea in the directions of preserving quality and semantic meaning of generated content in low-entropy text generation scenarios [486, 925], where text quality is vulnerable to such tiny bias towards generating randomly selected "green" tokens; multi-bit watermarking [1067, 269, 721]; improving the robustness of watermarking against removal attack and post-processing [444, 746, 1140, 26]; and defending against forgeries of watermarks [376, 986]. In contrast altering the logits, a line of works [366, 462, 184] alternatively choose to manipulate the *token sampling* process itself directly by encoding a watermark in a pseudo-random number sequence as seeds to guide the sampling of each token or sentence in a text generation sequence. Detection therefore needs to access the correspondence between the tokens generated and the underlying pseudo-random numbers.

Beyond text generation, training-free watermarks have also been applied to AI-generated images. For instance, DaLL·E [737] always prints a tiny visible color pattern at the bottom right corner of its generated images. To better preserve the visual quality of the generated images, invisible-watermark [929], which is adopted by Stable Diffusion, encodes bits of the watermark message through modifying coefficients of a carefully selected subset of band frequencies of its generated images under discrete wavelet transforms. Detection and decoding of the watermark is thereon achieved through an inverse transformation. In addition, [971] introduce a training-free watermark for diffusion models by embedding watermark signals into the initial latent noise, creating a semantic watermark.

### 11.3.2 Learnable Watermarking

Although training-free watermarking and detection techniques are straightforward in concept and require minimal effort to deploy, the pre-defined watermarking rules may be too conspicuous, leading to a compromise

in the quality of the generated content or making them susceptible to watermark removal and forgery. In this survey, we use "learnable watermarking" to refer to methods that *modify the generation process* to encode a keyed *watermark payload* at training or inference time, and whose verification requires the corresponding key (or a public verifier). This distinguishes them from post-hoc detectors in Section 11.4, which do not assume any embedded signal. To address this issue, a couple of studies [2, 1114] propose using learning-based watermark encoding and decoding modules, in which the training pipeline involves an encoder that first embeds a binary "watermark payload" into the original text followed by decoding for the message from the watermarked text. To preserve coherence and consistency of the generated content, the modules from [2] are trained against an adversary that performs a classification between the original and watermarked text, whereas [1114] regularize the watermarked message by penalizing semantic difference with the original text. [547] embed watermarks into text by adding extra watermark logits to the LLM's sampling logits at each generation step, following [443]. To ensure both attack robustness and security robustness, each watermark logit is determined by applying a learned transformation (a trained watermark model) on the semantic embedding of all preceding tokens generated using another pre-trained LLM. Two similarity loss and normalization loss are minimized during training to prompt semantic consistency and unbiasedness in the generated watermark logits and facilitate statistical detection. Moreover, in a recent work, [546] propose an unforgettable publicly verifiable watermark algorithm utilizing two different neural networks for watermark generation and detection, thereby preventing exposing key information in the watermark generation phase when made accessible for public detection. Furthermore, the token embedding parameters are shared between the generation and detection networks which improves both training efficiency and detection accuracy. [1082] proposed SAEMARK, a user-specific watermarking method that embeds personalized signatures without altering logits. SAEMARK uses Sparse Autoencoder (SAE) to extract features from generated texts and selects outputs by matching key-derived feature distributions. Boundary to Section 11.4. Although methods such as ASI and publicly verifiable schemes train neural *verifiers*, we keep them in Learnable Watermarking because they *require* a payload that was intentionally embedded at generation time. By contrast, Section 11.4 covers detectors that operate *without* any embedded watermark or key, treating detection purely as post-hoc content classification.

## 11.4 Neural Network Detectors

Unlike Section 11.3 (learnable watermarking), approaches in this section do *not* modify the generator and do *not* assume any embedded payload/key. They train post-hoc classifiers—often on human vs. AI corpora—and can operate in black-box settings against unknown generators. Consequently, any methods whose verification relies on a generation-time watermark remain in Section 11.3 rather than here. Another line of work approaches the AIGC detection problem by training a binary classifier using labeled training samples containing both human and AI-generated content. Earlier work focuses on fake review [87], fake news [1093], fake images [602], or small AI models detection [817, 59, 885]. Subsequently, growing interest in this line of research turns to detecting high-quality content brought by foundation models. Detectors under this category do not require access to model parameters hence can operate under complete black-box settings.

Targeting the problem of machine-generated text detection, numerous studies [162, 328, 1100, 859, 1080] fine-tune a pre-trained LLM, such as T5 [729] or RoBERTa [579], on a dataset of pairs of human-written text and AI-written text from mixed sources as a simple solution. Alternatively, several works also consider training a classifier on top of a frozen pre-trained LLM [162, 328, 978, 900]. In particular, [162, 328] have attempted training a logistic regression classifier on text embedding obtained using a pre-trained LLM for detection, however, they find such a method often underperforms the fine-tuning approach. [978] propose LLMDet, which conducts binary classification utilizing a proxy score for perplexity, while [900] propose Ghostbuster, which is inspired by statistical detection methods based on analyzing token log-probabilities. Both methods train a logistic regression classifier on top of these selected and hand-crafted features to detect machine-generated text, therefore, no longer requiring direct access to the model token sampling logits, as in their zero-shot counterparts, at test time. Recognizing the similarities between the original AI-generated and the regenerated text produced with ChatGPT, [1079] introduce a novel GPT Paternity Test for AI-generated text detection. This method involves utilizing ChatGPT to infer a question based on the input text being examined, followed by supplying a response. Subsequently, a Siamese network [447] is trained to

assess the similarity between the original and regenerated text, aiding the detection using another trained binary classifier.

One major challenge in training a reliable binary classifier is data scarcity as collecting sufficient data to train the classifier can be challenging, especially in diverse domains where the availability of training samples is a major bottleneck. To alleviate this, [572] consider adopting contrastive learning approaches in addition to the supervised training for detection. Another significant challenge involves tackling paraphrasing attacks [760, 459]. To mitigate this problem, [375] propose to employ an adversarial learning approach to simultaneously train a detector and a paraphraser. Nevertheless, supervised training of a binary classifier tends to overfit their training data, resulting in a decline in performance when faced with cross-domain or unseen data. Additionally, fine-tuning LLM classifiers is limited in facing data generated by different models.

## 11.5 Current Limitations and Future Directions

Despite significant advancements in the domain of AIGC detection, several limitations still require future attention:

### 11.5.1 Fairness of AIGC Detection

Although state-of-the-art text detectors generally achieve high accuracy in experimental settings, as discussed by [534], perplexity-based text detectors exhibit a notable bias against text written by non-native speakers. Specifically, these detectors have been observed to misclassify TOEFL essays written by foreign writers more frequently than those by native speakers. This discrepancy may be due to the lower perplexity of non-native essays, which often display less linguistic diversity and richness. This issue may also affect minority languages, which tend to have higher perplexities compared to popular languages like English. Additionally, similar biases might exist in other modalities, such as image detection. Therefore, it is crucial to consider the fairness of detectors when designing future detection methods and to develop efficient methods for evaluating the fairness of AIGC detection methods.

Meanwhile, on the watermarking side, learnable watermarking methods might also exhibit biases toward out-of-distribution data points. For instance, if the watermarking encoder and decoder are trained on English text written by native speakers, the model might also have a higher misclassification rate on essays written by non-native speakers. Therefore, it is crucial to consider fairness in the development of learnable watermarking methods as well.

### 11.5.2 Robustness of Watermarks

Both text and image watermarks are susceptible to regeneration or post-processing attacks, such as paraphrasing [444] or diffusion purification [1140]. In contrast, semantic watermarks tend to be more robust against such attacks. However, because semantic watermarks typically require deep neural networks to decode the watermark signals, they are vulnerable to adversarial attacks [759, 26]. Adversarial perturbations can also be developed to prevent regeneration and post-processing attacks [574]. Adversarial attacks remain a significant challenge even for classification tasks. Therefore, designing robust watermarks that can withstand both attacks is challenging and crucial.

### 11.5.3 Origin Attribution of Generated Images

Recent advancements in visual generative models have significantly improved the quality of generated images, raising concerns about their potential misuse. It is critical to develop methods to accurately identify the origin model responsible for generating a given image [574]. Especially, the scenarios are especially important and practical where access to the source model is restricted and only a limited number of images from the source model are available [574].

## 12 Intersection and Conclusion

In this survey, we comprehensively examine the reliability and responsibility of foundation models, spanning technical and societal considerations. Given our overview of the current research landscape, we identify significant prior research that makes progress on these critical issues. However, outstanding challenges limit the extent to which current models are reliable or responsibly developed, indicating more research is needed as this technology has a broader societal impact.

Moreover, greater attention should be paid to the intersections between different research areas, as the areas covered by this survey are interconnected and influence each other. Instead of addressing challenges in isolation, we advocate for a more holistic approach to ensure the overall reliability and responsible development of foundation models. In conclusion, we emphasize a number of key points of intersection across these domains, highlighting the challenges at these crossroads and outlining potential directions for future research.

### 12.1 Summary of Recent Community Efforts

A growing body of community work has conducted the reliability and responsibility reviews from complementary perspectives. Table 3 shows a summary of relevant past surveys[4]. For example, there are hallucination and grounding reviews for LLMs [383, 410], cross-modal syntheses [762], safety-at-scale perspectives spanning security, robustness, and governance [603], comprehensive trustworthiness benchmarking [1128, 393], and explainability overviews [1136]. These efforts collectively advance shared terminology, taxonomies, and evaluation practices, while emphasizing that trustworthy AI requires both method innovation and standardized assessment.

Table 3: Coverage of modalities and trustworthy AI topics in related surveys. A ✓indicates the survey has a considerable proportion of contents that emphasizes the modality or task. T: Text-only LLMs, M: Multimodal LLMs, T2I: Text-to-Image, T2V: Text-to-Video, B: Bias and Fairness, A: Alignment by Post-Training, S: Security, P: Privacy, H: Hallucination, U: Uncertainty, D: Distribution Shift, E: Explainability, C: AIGC Detection, I: Intersection.

| Survey | T | M | T2I | T2V | B | A | S | P | H | U | D | E | C | I |
|---|---|---|---|---|---|---|---|---|---|---|---|---|---|---|
| Huang et al. [383] | ✓ | ✓ | | | | ✓ | | | ✓ | ✓ | | ✓ | | |
| Sahoo et al. [762] | ✓ | ✓ | ✓ | ✓ | | ✓ | | | ✓ | | | | | |
| Shen et al. [790] | ✓ | | | | | ✓ | ✓ | | | | | ✓ | | |
| Gallegos et al. [279] | ✓ | | | | ✓ | | | ✓ | | | | | | |
| Huang et al. [393] | ✓ | ✓ | ✓ | ✓ | ✓ | ✓ | ✓ | ✓ | ✓ | | | | | ✓ |
| Zhao et al. [1136] | ✓ | | | | | ✓ | | | | ✓ | | ✓ | | |
| Wen et al. [968] | ✓ | | | | | | | | | ✓ | ✓ | | | |
| Ma et al. [603] | ✓ | ✓ | ✓ | ✓ | | ✓ | ✓ | ✓ | | | | | | |
| Lin et al. [538] | ✓ | ✓ | ✓ | ✓ | | ✓ | | | | | | ✓ | ✓ | |
| Zhang et al. [1128] | | ✓ | | | ✓ | ✓ | ✓ | ✓ | ✓ | | | | | |
| Anwar et al. [37] | ✓ | | | | | ✓ | ✓ | | | | | ✓ | | |
| Bengio et al. [82] | ✓ | | ✓ | | ✓ | ✓ | ✓ | | ✓ | | | ✓ | ✓ | |
| Huang et al. [392] | ✓ | ✓ | | | ✓ | ✓ | ✓ | ✓ | ✓ | | ✓ | | | ✓ |
| Gallegos et al. [280] | ✓ | | | | ✓ | ✓ | | | | | | | | |
| Zhang et al. [1133] | ✓ | ✓ | | | ✓ | ✓ | | | ✓ | ✓ | | | | |

### 12.2 Bias, Fairness, and Security

To improve the security of foundation models, adversarial training [611, 56] is often leveraged, which aims to make models resistant to malicious manipulations by conducting bi-level adversarial games [851, 599]

---

[4]The table's mapping of each survey's primary scope (based on abstracts and dedicated sections) could be limited. We welcome corrections, suggestions, and insights from readers and survey authors and will revise the table accordingly.

during training. However, while adversarial training improves robustness, it can unintentionally exacerbate fairness issues [1017]. For example, robust models may focus on defending against a certain set of features without considering the fact that some demographic groups may be penalized by these features more than others, leading to disproportionate performance degradation for underrepresented or marginalized groups. For instance, [766] found that hate speech detection models were biased against African American English (AAE), disproportionately misclassifying non-offensive AAE utterances as hate speech.

On the other hand, security risks such as data poisoning, where malicious attackers corrupt training data, can introduce new biases into foundation models or exacerbate existing biases [627, 335]. Poisoned data can skew a model's learned representation, posing the dual challenge of protecting models from data poisoning while ensuring that the fairness of the model is not compromised, especially when training on large, uncurated datasets.

## 12.3   Bias, Fairness, and AI-generated Content Detection

Large multimodal and text-to-image models are often biased due to unbalanced or stereotype-laden training data, such as images that reinforce harmful stereotypes or misrepresent certain social or cultural groups [1127, 180, 591]. Detecting biased AI-generated content (AIGC) poses a significant challenge, as detection systems themselves can inherit or amplify biases found in the training data. For instance, zero-shot or neural network detectors might disproportionately mark content created by minority groups as *AI-generated* based on biased data patterns in the training set. It is important to ensure the fairness of these detection capabilities, as biased detection could result in unfair discrimination, such as unfairly enforcing restrictions on content produced by minority groups.

Similarly, watermark-based detectors can raise fairness concerns if watermarks are inconsistently applied across different types of content. For example, if T2I models excessively generate watermarked content associated with certain groups (e.g., images associated with a particular ethnic or gender identity), this content could be more easily flagged or blocked, suppressing content created or represented by those groups.

## 12.4   Security and Privacy

Privacy concerns often go hand-in-hand with issues of security. For example, models ought not to reveal users' private information, which can be found in pre-training data as well as in interactions with users. Furthermore, certain legal jurisdictions already impose privacy-related laws that impact technologies including foundation models. For example, the EU's GDPR covers a "Right to be Forgotten", which mandates that users have the ability to delete their private information [1104]; non-compliance with these regulations could result in legal penalties.

Several methods exist for preventing models from revealing private information, including unlearning (e.g., removing concepts from the model's parametric knowledge) [568, 1054, 408, 989] and introducing prompts [251] or RLHF [1005] to prevent models from revealing information. However, jailbreaking attacks (see Section 5.1.1) can often circumvent prompts and RLHF, and [695] show that model editing is also vulnerable to attack, finding that sensitive information could be recovered from models even after deletion when querying multiple times. Thus, making models compliant with existing and future legislation regarding private information is an open challenge, as is robustly defending against adversaries attempting to extract private information.

## 12.5   Security and AI-generated Content

AIGC detection is naturally framed as an adversarial task, with an attacker attempting to pass AIGC as real content, and a defender attempting to detect it. In such scenarios, the advantage usually lies with the attacker, who can make multiple attempts to test existing defenses. Existing work shows that the robustness of current detection methods is imperfect and they are vulnerable to adversarial attacks [759, 26, 444, 1140]; improving these detection algorithms remains an area of continuous future work.

Beyond the security of individual watermarking and detection methods, AIGC raises broader questions of *societal* security, i.e. the potential threats that AIGC poses to both public institutions and individuals. Here, extensive documentation exists regarding ongoing threats from AIGC. In the political sphere, AIGC has been employed to disseminate misinformation and erode public trust in political systems and elections [229, 418], where AIGC has been used to spread misinformation and sow distrust. Similarly, in public health settings, AIGC has been utilized to generate and spread health-related misinformation [632]. Other voices have also called attention to the risks associated with AIGC's interactions with individuals. For example, [315] raise the concern that AIGC could harm mental health through hyper-personalization. Moreover, individuals may become more susceptible to personalized fraud attempts, such as voice cloning or sophisticated phishing-style attacks [76, 261, 114, 133]. To counteract these malicious use-cases, larger-scale safeguards will likely be needed. This includes implementing public education initiatives to raise awareness about the risks of AIGC and developing strategies to combat its misuse across various domains.

## 12.6 Uncertainty and Alignment

Given the importance of correctly expressing model uncertainty (as described in Section 8), a growing area of interest is in aligning models to accurately predict their uncertainty [637, 833] or to abstain from answering in cases of uncertainty [968]. This work builds on past work finding that models' internal states often contain meta-knowledge about whether the model can correctly respond to a particular prompt [421, 637, 563]. Several past efforts have explored aligning models to predict this information based on internal states. [637] train models to express uncertainty linguistically by extracting control codes from their internal states and using them to adjust the model's output. [888] train a smaller LLM to predict the uncertainty of a larger LLM. [833] use a speaker-listener framework to supervise models, rewarding a generator or speaker model for getting a listener to accept correct answers and reject incorrect ones, while penalizing it for doing the opposite outcomes. Future research directions in this field include addressing uncertainty *not* reflected by the model's internal state, that is, the "unknown unknowns" where the model is unaware of its own ignorance, often when faced with out-of-distribution data or novel concepts, and enhancing models' capacity to resolve uncertainty effectively.

## 12.7 Hallucination, Uncertainty, Distribution Shift, and Alignment

Uncertainty and distribution shifts are deeply interconnected with hallucinations in foundation models [1006, 1169, 265, 484]. When these models encounter unfamiliar OOD data, they often lack the ability to detect OOD data accurately, leading to highly overconfident predictions. This unreliability is especially concerning in safety-critical applications. The risk of hallucination also increases as the model makes inferences based on its prior knowledge [747, 511], which may not be applicable in the new context [410, 383, 1133]. In multimodal settings, training a model on evolving image-text data presents significant challenges, as distribution shifts in training data increase the potential of the model to forget previously learned knowledge and disrupt modality alignment. This can exacerbate hallucinations of the model by incorrectly aligning the relationships between observations and prior knowledge [484, 1178, 417]. For example, changes in language usage or visual context can cause the model to lose its ability to associate specific textual descriptions with visual cues, resulting in errors when trying to generalize to unseen data, highlighting the need for effective memory retention mechanisms or continual learning strategies [584, 827, 1069, 199] to maintain seamless alignment between modalities.

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
