# OpenReview forum: "Reliable and Responsible Foundation Models"
_TMLR — Accepted by TMLR_

### Review · Reviewer_8MQQ · 2025-05-30

**Summary Of Contributions:**

This work attempts the herculean task of summarizing and synthesizing aspects of reliable and responsible deployment of foundation models across different aspects such as bias and fairness, alignment, security, privacy, hallucinations, uncertainty quantification, distributional shift, explainability, and AI-generated content detection.
This is further done for both large language models (LLMs) and large multi-modal models (LMMs), including image generation models.
As such, it provides a very broad overview over these different research topic and also points out many relevant intersections in the conclusion.

**Audience:**

Yes

**Broader Impact Concerns:**

I do not foresee any broader impact concerns with this work.

**Claims And Evidence:**

Yes

**Requested Changes:**

In the following, I first list requested changes in general and then on a per-section basis.


General changes
---------------

* Figure caption should describe the content of the figures better, as most of the time the task is left to the reader to match the contents of the chapter to parts of the figure.
* I found it hard to follow sometimes whether the current paragraph was referring to LLMs, T2I- or vision-language models. I encourage the authors to potentially experiment with some visual indicators like an emoji or other symbols in the text or in the margins of the paper to supply visual anchors to a reader.
* Since the scope of this paper is so large, it feels prudent to me to add a related work section in the end that mentions and maybe also discusses a bit existing surveys related to all the chapters.
* A lot of the discussed the work is technical and research-focused. I would appreciate thoughts and discussions of the authors that also concern the practical frictions and challenges implementing some of the mentioned techniques, especially with respect to structural challenges, financial incentives, and the interaction of human users with foundation models (and how this might sometimes be misaligned with the results from research methods).

Section 1
---------

* I find the definition of foundation models through "emergent abilities" very vague, especially since it is not intuitive clear what emergent abilities are for instance in T2I models and also because emergent abilities have been critically examined in the past [1-3].
* Since transformers and diffusion models are the focus of this work, it might be instructive for readers to get the briefest of introduction into how they work.
* In Figure 1, "II. Quantifying and Addressing" seems to be missing a word.
* The authors do not really define what they mean by "reliable" and "responsible" foundation models. Although these terms have intuitive definitions, they should be defined explicitly here, since they form the key motivation of the work.

Section 2
---------

* In this chapter it is a bit unclear of what is defined as LLMs, since equation (1) implies decoder-only LMs, but later chapters also reference papers about BERT.
* "A general design recipe for LMMs is to adopt architectures that integrate multimodal features in a shared latent space" - for me, this warrants some citations about examples, and maybe a footnote outlining alternative architectures, and why they are not discussed here.
* I find equations (1) - (4) a bit vague and in their current form not very informative, since (afaik) they are not referenced in the rest of the manuscript. I would find it slightly more informative to define a set of modalities of which video, text and imagines are elements, and base any formal definitions on that.
* The paragraph after equation (3) should provide the corresponding citations for diffusion models, GANs, and transformers. This section kind of also glosses over whether state-space models like MAMBA [4] could be considered foundation models, and that multi-modal architecture might combine different model classes.

Section 3
---------

* Table 1 could profit from providing sources for these definitions, or comparing definitions from different works.
* The same applies to the definition of bias in the paragraph of the same name on page 11 (or at least it should be made clear that this is the definition of bias by the authors).
* The notation is equation (5) should be harmonized with the notation introduced in section 2 (same holds with the rest of the notation in this section)
* If a measurement function maps to a scalar score, then it should be written $S: \mathcal{Y} \rigtharrow \mathbb{R}$.
* "It is noteworthy that this is just one possible definition of bias, while other definitions or metrics can also be reasonable" -> mention or at least cite papers with other defintions or metrics here.
* In the following paragraph, a set of principles by Gallegos et al. (2023) are named, but not discussed or not explained at all.
* Equations (6) - (10): Use \text for abbreviations / words in equations
* "Similarly, Demographic Representation (DR), as discussed in" -> use \citet for citations
* The paragraph "Methods based on Token Selection Probability" talks about masked-language models that are encoder-only, which were not included as foundation models in section 2.
* Section 3.3: "For specific categories that are extensively discussed in later chapter." is missing some words.
* "However, it is essential to recognize that biases in the training data and the training process are not inherently designed to understand or priorize human values." I think I understand the point that the authors aim to make here, but it is a bit oddly worded. It also obfuscates that some of the biases in training data originate from human design decisions based on their own values and biases (see [5]).
* The mention of chain-of-thought at the start of page 16 should come witha citation.
* Definting the toxic text in the next line with the variable "t" is redundant, since it is not used.

Section 4
---------

* At the end of the second paragraph 20, the word "propose" is used three times right after another.
* "Then, a series of works [...] propose to use well-suited rules, metrics, and LLM-based methods for data clearning" -> This should be explained further.
* At the end of the third paragraph, the authors could include [6], which uses bootstrapped conversations for task-oriented dialogue finetuning.
* Another caveat to add to 4.1 could be [7], showing that overtrained models are harder to finetune.
* Section 4.2 claims that one stage of RLHF is instruction finetuning with SFT. I would intuitively rather claim that the outcome of the instruction finetuning step is used for RLHF, which is a separate step.
* p.22: TEXTGAN should be cited
* Zhu et al. (2023b) citation should use \citep instead of \citet
* The part about diverse user preference should also discuss the cultural component, for instance through [8].
* The end of section 4.2 could profit from providing an overall discussion on when which RLHF objective should ideally be used.
* The "Discrete Prompts" paragraph on p. 25 should discuss also the brittleness of models w.r.t prompts, for instance discussed in [9].
* The "Chain-of-Thought" paragraph should also include a discussion about how reasoning chains do not necessarily need to reflect the actual reason a model arrived at a solution [10].
* The paragraph on "Prompt Optimization" could refer to [11].
* The same paragraph includes "(i.e., agent workflow (Zhou et al., 2023d))", which could be formatted better with \citealp. This also applies to "(e.g., GPT-4 (OpenAI, 2023b))" in section 4.4.
* The reference to "visual instruction tuning" in section 4.4 needs a citation.
* "dis-preferred" -> "dispreferred"
* The "superalignment problem" at the beginning of section 4.5 should be explained more.
* Redundant whitespace before "Casper et al. (2023)" in section 4.5.
* "For example, a reward model with uniform values may make the training of foundation models more scyphantic rather than maintaining integrity" - this argument is not very clear.
* "In the superalignment situation, many research works find that strong models can easily manipulate and deceive weak models" - no citations or evidence for this is given.
* "making evaluation super difficult" -> "making evaluation quite difficult" (or some variation of this)
* "And this can leave backdoor vulnearabilities in the model, which can be exploited by adversaries" - references?
* "Ensemble methods [...] can be a potential solution of scalable oversight to overcome the weakness of a single weak model." - references in the context of alignment?

Section 5
---------

* General comment: Large parts of the section read very repetitive, feeling more like a listing of papers without pointing out connections between papers or overarching trends.
* Section 5.1.1 should also summarize some evidence from the literature on why and how jailbreaks work.
* The "Prompt Injection Attacks" paragraph would profit from examples.
* "Shi et al. (2023a) and Rando & Tramèr (2024) explore the possibiltiy of injecting backdoors into the reward model during the RLHF process" - so what would an actualy attack in the end look like?
* Section 5.2: The "projected gradient descent attack" is mentioned, but no intuition / explanation of the attack is given.
* The "IDEFICS mechanism" is mentioned on p. 31, but not explained. THe same goes for "2-hop prompt templates".
* The LAION-5B dataset in 5.3.1 should be cited.
* "(Wang et al., 2024s)" should be \citet and not \citep.
* The "assimilation phenomenon" in the same line is not explained.
* p 33: "Yoon et al. (Yoon et al., 2024)" should just be \citet]
* "Model Fine-Tuning" -> "Model finetuning"
* "(Yang et al., 2024e)" -> \citet
* Section 5.4.1: "Chen et al." -> \citet
* Section 5.4.2 seems to be missing a discussion using external models for safety, see e.g. [12]

Section 6
---------

* The reference to homomorphic encryption should be better explained and connected to the rest of section or be omitted.
* Section 6.11 contains a reference to BERT, which is not cited and not introducted in section 1 as a foundation model. The MIA acronym is also not introduced.
* "MIN-K% PROB" attack should be explained, at least in broad terms.
* Section 6.1.2: "the field has shifted towards developing various techniques to preserve privacy LLMs" - some word seems to be missing here
* Section 6.1.2 also maybe warrants a brief introduction into how and why differential privacy works in general.
* Section 6.2: "Li et al. (2024m) studied membership inference against LMMs" - reads very vague.
* Section 6.3: "Ghalebikesabi et al. (Ghalebikesabi et al., 2023)" -> use \citet.
* "... offering enhanced privacy protections" - what protections? By doing what?
* Section 6.4.1: "For example, certain attacks require poisoning or injecting triggers..."  - What kind of attacks?
* Section 6.4.2: "Integrating DP into the foundation model fine-tuning still leads to significant performance degredation" - references? Same with the following paragraph "However, these methods also suffer from significant performance degradation".

Section 7
---------

* If the definition of hallucination is by the authors, this should be made more explicit. It would also be interesting to compare the definition to others in the literature.
* Figure 12 should also be referenced in 7.1.1, since it depicts the different kinds of hallucinations that are discussed there.
* Is "Instruction Deviation" really a type of hallucination, or just an indicator of a erroneous model?
* Section 7.2: "For example, hallucinations in LMMs are often observed in which the model predicts possible objects or actions that are not supported by the image but are plausible from commensense" - references?
* The nested parantheses due to citations in the "Knowledge Boundaries" point could be mitigated through \citealp
* "Path Dependence. The autoregressive nature of LLMs and LMMs causes hallucination." What is the evidence that this is because the models are autoregressive? Are diffusion models actually autoregressive?
* The title of 7.3.1 should perhaps be "Detecting language hallucinations"?
* The citation of "Jiang et al. (2024b)" should use \citep.
* Monte Carlo dropout in the "uncertainty estimation" needs a citation.
* p. 44 second paragraph: Llama is missing a citation. Also, I believe RAG is neither introduced as an acronym nor cited.
* Title for section 7.4.1 should include an s in "hallucinations".
* "Training Approaches" paragraph in 7.4.2: "best-of-n rejection sampling" is not really explained.
* "Post-processing Approaches" on p. 46: It is not really clear to me as a reader how hidden Markov models and the Viterbi algorithm are applied in this context.
* "Decoding Approaches" paragraph mentions "VCD", but the acronym is not explained and the approach is not explained very intuitively.
* Section 7.5 might also want to include a discussion on refusal in models in order to avoid hallucinations.

Section 8
---------

* "uncertainty" as a concept in the context of foundation models isn't really well defined.
* Section 8.1: The aleatoric / epistemic distinction has recently attracted a lot of critique due to its rigidity and reductiveness, see [13-16].
* Last paragraph before 8.1.2: "under-prescribed" -> "underspecified"?
* Critique of optimizing only observed sequence probability in 8.1.2 could be supplemented with [17, 18].
* Section 8.2: "Methods have been developed to recalibrate uncertainty scores to match the probability of an answer being correct" -> This is not completely correct, as you would only calibrate confidence scores in this way. The previous sentence mentions predictive entropy for instance, where we want higher uncertainty to signify a higher chance of being incorrect, but there is no immediate way to "calibrate" entropy to reflect the probability of correctness.
* Section 8.2.2: Many more alternatives to the ECE have been proposed, see for instance [14] p. 37.
* Section 8.2.5: "(Yadkori et al., 2024b)" should use \citet
* Section should include [19-21].
* Section 8.3: Long-form UQ has actually been explored in works such as [22-24].
* p. 55 "cross entropy" -> "cross-entropy"
* Use \citealp for nested citation "(Kadavath et al., 2022; Tian et al., 2023b)"


Section 9
---------

* There are no citations given for the definition of shifts in section 9.1. Candidates could be [27-29].
* For me, it would be interesting to see motivated why OOD detection in section 9.2. still matters in the age of extremely large foundation models.
* The section is missing [28].
* Section 9.2: It is not extremely clear as a reader what "OOD peer classes" are. Also, the concept of far, near, and fine-grained OOD is not explained.
* Section 9.3: "Priorto the era of foundation models, various strategies were proposed to improve the OOD generalization capabilities of traditional deep learning models, including ..." - in the following, no references / examples in the literature are given for any of the strategies.
* Section 9.3.1: "Easy Data Augmentation" isn't really expalained.
* Section 9.3.4: It is not clear as a reader what these different loss functions do, how they differ, and why they would work.
* Section 9.3.5: Model ensembles warrant some citations, for instance [29, 30].
* p. 60: "In-Context" -> "in-context"
* Second paragraph p. 60: Missing citations for T5 and ViT.
* The last paragraph again declares SFT as a stage of RLHF, but the next sentence contrasts RLHF and SFT in terms of generalization.
* Section 9.4.3 perhaps misses works on activation steering / finetuning such as [31, 32.]
* p.63: I am not sure that "FFN" as acronym was previously introducted.
* Use \citealp for "(Cao et al., 2021)"" nested citation.
* Paragraph "Knowledge-based Model Editing" is missing a citation for integrated gradients.
* In the same paragraph, the way that ROME and MEMIT work isn't really explained.
* The end of 9.4.5 could profit from a discussion on how the different model editing methods differ and what kind of advantages and disadvantages they carry.


Section 10
----------

* Section 10.1.2: BERT is spelled Bert and is missing citations.
* Applications to decoder-only models for gradient-based explanation are missing?
* Section 10.1.3 is missing [33].
* Section 10.2.1 is missing early work of [34].
* Last paragraph in 10.2.1 could also add [35, 36, 37]
* Referencing the key-value memory in transformers in 10.2.3 could be a reason to give a brief intro into the architecture in the earlier sections.
* Section 10.3.1 is missing an explanation for / references about (the use of) influence functions. Another work to add is also [38].
* Section 10.3.3 feels superficial giving the volume of work trying to understand the role of ICL, see for instance
* Section 10.4: Use \citealp for "(Jacovi & Goldberg, 2020)" citation.
* Section 10.4.1: "Another benchmark Hate Explain (Mathew et al., 2021) ask the annotators..." -> "Another benchmark, Hate Explain (Mathew et al., 2021), asks the annotators..."
* Section 10.4.2: ", and" after "predicted rationales"
* In the same paragraph "The lower, the better;" reads very informal.
* Section 10.5.1 might be renamed into something like "Avoiding shortcut learning"
* Section 10.5.2: Here some critique of CoT ([10]) should be mentioned alongside the Nye et al. (2021) paper.
* The section also missed early work on incorporating human feedback, see [42] for an overview.
* Section 10.7.2.: "It remains unclear how this knowledge is injected during training" -> see for instance [43-45].
* Section 10.7.3: "Identifying the biases" -> "identifying the biases", "de-bias strategies" -> "de-biasing strategies"
* I found it interesting to maybe also end this section on a discussion on which methods can be transferred between different kinds of foundation models (specifically with respect to modality).

Section 11
----------

* Section 11.1: What about partially altered content?
* Section 11.2.1 The concept of "cross-perplexity" is not explained.
* In the following paragraph, the fact that AI-generated text "tends to lie in negative curvature of the model's log probability" is not clear.
* p.74 "By observing that AI-generated text exhibits lower intrinsic dimensionality" - in what space?
* Section 11.2.2 should maybe also mention that detection by intuition becomes much harder as models improve.
* Section 11.3: "watermark payload" should use latex quotes, i.e. `` and ''.
* Section 11.3.2 might want to add [46].
* Some of the entries 11.3.2 seem to also fit 11.4, which maybe warrants making the differences more explicit.
* Second paragraph of 11.4: Missing citations for T5 and RoBERTa and Siamese network.
* Same paragraph: Comma after "LLMDet" and "Ghostbuster". No comma after "machine generated-text, therefore".
* Last paragraph in 11.4: "To mitigate this problem, bing"?
* Section 11.5.3: "Especially, The scenarios" -> "Especially, the scenarios"

Section 12
----------

* Section 12.1: "Tao et al. (2021); Ma et al. (2024a)" should be \citep. Same for "Zhang et al. (2023j); Cho et al., (2023); and Luccioni et al. (2023)" in section 12.2..
* Section 12.3: \citealp for Liu et al. (2024g) and following citations.
* Same paragraph: \citep for Edemacu & Wu (2024) and Xiao et al. (2023a)
* Section 12.5: "opposite outcomes Future research directions" -> "opposite outcomes, future research directions"
* "include addressing uncertainty not capteured in the model's interal state" - this point os not very clear to me.
* "The risk of hallucination also increases as the model makes inferences based on its prior knowledge, which may not be applicable in the new context" - missing reference?


[1] Schaeffer, Rylan, Brando Miranda, and Sanmi Koyejo. "Are emergent abilities of large language models a mirage?." Advances in Neural Information Processing Systems 36 (2023): 55565-55581.

[2] Lu, Sheng, et al. "Are emergent abilities in large language models just in-context learning?." arXiv preprint arXiv:2309.01809 (2023).

[3] Rogers, Anna, and Alexandra Sasha Luccioni. "Position: Key claims in llm research have a long tail of footnotes." arXiv preprint arXiv:2308.07120 (2023).

[4] Gu, Albert, and Tri Dao. "Mamba: Linear-time sequence modeling with selective state spaces." arXiv preprint arXiv:2312.00752 (2023)

[5] Waseem, Zeerak, et al. "Disembodied machine learning: On the illusion of objectivity in nlp." arXiv preprint arXiv:2101.11974 (2021).

[6] Ulmer, Dennis, et al. "Bootstrapping llm-based task-oriented dialogue agents via self-talk." arXiv preprint arXiv:2401.05033 (2024).

[7] Springer, Jacob Mitchell, et al. "Overtrained language models are harder to fine-tune." arXiv preprint arXiv:2503.19206 (2025).

[8] Kirk, Hannah Rose, et al. "The prism alignment project: What participatory, representative and individualised human feedback reveals about the subjective and multicultural alignment of large language models." arXiv preprint arXiv:2404.16019 (2024).

[9] Sclar, Melanie, et al. "Quantifying Language Models' Sensitivity to Spurious Features in Prompt Design or: How I learned to start worrying about prompt formatting." arXiv preprint arXiv:2310.11324 (2023).

[10] Turpin, Miles, et al. "Language models don't always say what they think: Unfaithful explanations in chain-of-thought prompting." Advances in Neural Information Processing Systems 36 (2023): 74952-74965.

[11] Zollo, Thomas P., et al. "Prompt risk control: A rigorous framework for responsible deployment of large language models." arXiv preprint arXiv:2311.13628 (2023).

[12] https://www.anthropic.com/news/constitutional-classifiers

[13] Baan, Joris, et al. "Uncertainty in natural language generation: From theory to applications." arXiv preprint arXiv:2307.15703 (2023).

[14] Ulmer, Dennis. "On Uncertainty In Natural Language Processing." arXiv preprint arXiv:2410.03446 (2024).

[15] Mucsányi, Bálint, Michael Kirchhof, and Seong Joon Oh. "Benchmarking uncertainty disentanglement: Specialized uncertainties for specialized tasks." Advances in neural information processing systems 37 (2024): 50972-51038.

[16] Gruber, Cornelia, et al. "Sources of Uncertainty in Machine Learning--A Statisticians' View." arXiv preprint arXiv:2305.16703 (2023).

[17] Eikema, Bryan, and Wilker Aziz. "Is MAP decoding all you need? the inadequacy of the mode in neural machine translation." arXiv preprint arXiv:2005.10283 (2020).

[18] Eikema, Bryan. "The effect of generalisation on the inadequacy of the mode." Proceedings of the 1st Workshop on Uncertainty-Aware NLP (UncertaiNLP 2024). 2024.

[19] Ulmer, Dennis, Chrysoula Zerva, and André FT Martins. "Non-exchangeable conformal language generation with nearest neighbors." arXiv preprint arXiv:2402.00707 (2024).

[20] Su, Jiayuan, et al. "Api is enough: Conformal prediction for large language models without logit-access." arXiv preprint arXiv:2403.01216 (2024).

[21] Ravfogel, Shauli, Yoav Goldberg, and Jacob Goldberger. "Conformal nucleus sampling." arXiv preprint arXiv:2305.02633 (2023)

[22] Zhang, Caiqi, et al. "Luq: Long-text uncertainty quantification for llms." arXiv preprint arXiv:2403.20279 (2024).

[23] Yoon, Dongkeun, et al. "Reasoning Models Better Express Their Confidence." arXiv preprint arXiv:2505.14489 (2025).

[24] Zhang, Caiqi, et al. "Atomic Calibration of LLMs in Long-Form Generations." arXiv preprint arXiv:2410.13246 (2024).

[25] Moreno-Torres, Jose G., et al. "A unifying view on dataset shift in classification." Pattern recognition 45.1 (2012): 521-530.

[26] Shimodaira, Hidetoshi. "Improving predictive inference under covariate shift by weighting the log-likelihood function." Journal of statistical planning and inference 90.2 (2000): 227-244.

[27] Hupkes, Dieuwke, et al. "A taxonomy and review of generalization research in NLP." Nature Machine Intelligence 5.10 (2023): 1161-1174.

[28] Arora, Udit, William Huang, and He He. "Types of out-of-distribution texts and how to detect them." arXiv preprint arXiv:2109.06827 (2021).

[29] Lakshminarayanan, Balaji, Alexander Pritzel, and Charles Blundell. "Simple and scalable predictive uncertainty estimation using deep ensembles." Advances in neural information processing systems 30 (2017).

[30] Dietterich, Thomas G. "Ensemble learning." The handbook of brain theory and neural networks 2.1 (2002): 110-125.

[31] Lai, Wen, Alexander Fraser, and Ivan Titov. "Joint Localization and Activation Editing for Low-Resource Fine-Tuning." arXiv preprint arXiv:2502.01179 (2025).

[32] Turner, Alexander Matt, et al. "Activation addition: Steering language models without optimization." arXiv e-prints (2023): arXiv-2308.

[33] Abnar, Samira, and Willem Zuidema. "Quantifying attention flow in transformers." arXiv preprint arXiv:2005.00928 (2020).

[34] Veldhoen, Sara, Dieuwke Hupkes, and Willem H. Zuidema. "Diagnostic Classifiers Revealing how Neural Networks Process Hierarchical Structure." CoCo@ NIPS. 2016

[35] Ji, Ziwei, et al. "Calibrating Verbal Uncertainty as a Linear Feature to Reduce Hallucinations." arXiv preprint arXiv:2503.14477 (2025).

[36] Merullo, Jack, et al. "On linear representations and pretraining data frequency in language models." arXiv preprint arXiv:2504.12459 (2025).

[37] Marks, Samuel, and Max Tegmark. "The geometry of truth: Emergent linear structure in large language model representations of true/false datasets." arXiv preprint arXiv:2310.06824 (2023).

[38] Ruis, Laura, et al. "Procedural knowledge in pretraining drives reasoning in large language models." arXiv preprint arXiv:2411.12580 (2024).

[39] Hahn, Michael, and Navin Goyal. "A theory of emergent in-context learning as implicit structure induction." arXiv preprint arXiv:2303.07971 (2023).

[40] Xie, Sang Michael, et al. "An explanation of in-context learning as implicit bayesian inference." arXiv preprint arXiv:2111.02080 (2021).

[41] Lu, Sheng, et al. "Are emergent abilities in large language models just in-context learning?." arXiv preprint arXiv:2309.01809 (2023).

[42] Fernandes, Patrick, et al. "Bridging the gap: A survey on integrating (human) feedback for natural language generation." Transactions of the Association for Computational Linguistics 11 (2023): 1643-1668.

[43] van der Wal, Oskar, et al. "PolyPythias: Stability and Outliers across Fifty Language Model Pre-Training Runs." arXiv preprint arXiv:2503.09543 (2025).

[43] Müller-Eberstein, Max, et al. "Subspace chronicles: How linguistic information emerges, shifts and interacts during language model training." arXiv preprint arXiv:2310.16484 (2023).

[44] Hu, Michael Y., et al. "Latent state models of training dynamics." arXiv preprint arXiv:2308.09543 (2023).

[45] Saphra, Naomi, and Adam Lopez. "Understanding learning dynamics of language models with SVCCA." arXiv preprint arXiv:1811.00225 (2018).

[46] Yu, Zhuohao, et al. "SAEMark: Steering Personalized Multilingual LLM Watermarks with Sparse Autoencoders."

**Strengths And Weaknesses:**

I found this paper very paradoxical and very hard to review due to its sheer length of 81 pages and large scope.
I will summarize my general impressions here and follow up with chapter-based feedback in the following section.


Strengths
---------

* The paper covers essentially all topics relevant to reliable and responsible deployment of AI, and does so for all kinds of foundation models.
* This allows the paper to point out intersections and cross-connections between different fields that are otherwise out-of-scope for specialized surveys.
* The figures in the paper are top-notch and illustrate a lot of the concepts very well.

Weaknesses
----------

* The paper feels both over- and underscoped at the same time. Many of the aspects in the paper like bias, hallucinations or uncertainty are covered in many other dedicated survey papers in more breadth and depth. It feels like in order to provide a lot of added value to the reader, the paper would have to discuss even more works and provide more detail (which would question whether the appropriate format would be a book instead of a paper) and / or point out / discuss even more cross-connections between the fields in order to give a birds-eye view.
* Although the figures are great, they are consistenly underdescribed and not well connected to the rest of the text.
* The writing is inconsistent - while this can be forgiven since this work probably required the work of many authors, the writing also does not feel very polished in some places (missing or wrongly formatted citations, typos etc.) and some sections feel very superficial / rushed.

---

> ### Author Response · Authors · 2025-08-08
> **Response to Reviewer 8MQQ (1/N)**
>
> Thank you for taking the time to review our paper and for your valuable feedback. Below, we address your concerns point by point and have revised the manuscript accordingly. We would greatly appreciate it if you could let us know whether our responses adequately address your concerns. For clarity, we have separated our responses into general and per-section responses.
>
> # General Response
>
> > Q1: Figure caption should describe the content of the figures better, as most of the time the task is left to the reader to match the contents of the chapter to parts of the figure.
>
> A1: Thank you for your feedback. We have revised the captions of each figure to include more descriptive details, making it easier for readers to understand the figures and align them with the corresponding content in the text.
>
> > Q2: Since the scope of this paper is so large, it feels prudent to me to add a related work section in the end that mentions and maybe also discusses a bit existing surveys related to all the chapters.
>
> A2: Thank you for your suggestion. We agree that including discussions of existing surveys can help clarify our contributions. Instead of adding a separate related work section at the end, we chose to integrate relevant surveys at the beginning of each section. We believe this approach better contextualizes our coverage and highlights the distinctions of our work throughout the paper.
>
> > Q3: A lot of the discussed the work is technical and research-focused. I would appreciate thoughts and discussions of the authors that also concern the practical frictions and challenges implementing some of the mentioned techniques, especially with respect to structural challenges, financial incentives, and the interaction of human users with foundation models (and how this might sometimes be misaligned with the results from research methods).
>
> A3: Based on your suggestion, we have added more discussion on the practical frictions and challenges in the “Current Limitations and Future Directions” subsection of each topic.

---

> > ### Author Response · Authors · 2025-08-08
> > **Response to Reviewer 8MQQ (2/N)**
> >
> > # Section 1
> >
> > > Q1: I find the definition of foundation models through "emergent abilities" very vague, especially since it is not intuitive clear what emergent abilities are for instance in T2I models and also because emergent abilities have been critically examined in the past [1-3].
> >
> > A1: Thank you for your suggestion! We have further revised our definition of foundation models to emphasize their general-purpose nature rather than relying on the concept of emergent abilities
> >
> > > Q2: Since transformers and diffusion models are the focus of this work, it might be instructive for readers to get the briefest of introductions into how they work.
> >
> > A2: Thank you for the suggestion. We have added brief introductions to both transformers and diffusion models prior to discussing their development, to help readers better understand the core mechanisms underlying these architectures.
> >
> > > Q3: In Figure 1, "II. Quantifying and Addressing" seems to be missing a word.
> >
> > A3: We have revised it into "Quantifying and Addressing Uncertainty".
> >
> > > Q4: The authors do not really define what they mean by "reliable" and "responsible" foundation models. Although these terms have intuitive definitions, they should be defined explicitly here, since they form the key motivation of the work.
> >
> > A4: We have added definitions in the introduction. In the context of this survey, we define reliability as the model’s capacity to perform its intended functions accurately, consistently, and robustly, especially under challenging conditions like distribution shifts. We define responsibility as the alignment of a model’s behavior with ethical principles and societal values, encompassing crucial aspects such as fairness, privacy, security, and transparency.

---

> > > ### Author Response · Authors · 2025-08-08
> > > **Response to Reviewer 8MQQ (3/N)**
> > >
> > > # Section 2
> > >
> > > > Q1: In this chapter it is a bit unclear of what is defined as LLMs, since equation (1) implies decoder-only LMs, but later chapters also reference papers about BERT.
> > >
> > > A1: We have revised the text to explicitly clarify our definition of LLMs. The updated version now specifies that the term encompasses a range of architectures, including encoder-only models like BERT [1], decoder-only models like GPT [2], and encoder-decoder models like T5 [3], to avoid any ambiguity.
> > >
> > > > Q2: A general design recipe for LMMs is to adopt architectures that integrate multimodal features in a shared latent space" - for me, this warrants some citations about examples, and maybe a footnote outlining alternative architectures, and why they are not discussed here.
> > >
> > > A2: To better support our statement, we have now included citations for seminal works that exemplify this design approach, specifically mentioning CLIP [4] and LLaVA [5] as models that integrate multimodal features in a shared latent space.
> > >
> > > > Q3:  I find equations (1) - (4) a bit vague and in their current form not very informative, since (afaik) they are not referenced in the rest of the manuscript. I would find it slightly more informative to define a set of modalities of which video, text and imagines are elements, and base any formal definitions on that.
> > >
> > > A3: To make the equations more informative and grounded, we have revised this section and introduced a formal definition for a set of core modalities. All subsequent equations are now based on this defined set, providing a clearer and more systematic framework.
> > >
> > > > Q4: The paragraph after equation (3) should provide the corresponding citations for diffusion models, GANs, and transformers. This section kind of also glosses over whether state-space models like MAMBA [4] could be considered foundation models, and that multi-modal architecture might combine different model classes.
> > >
> > > A4: We have updated the text to include the appropriate citations for diffusion models, GANs, and autoregressive Transformers. Furthermore, we have expanded the discussion to acknowledge the role of emerging architectures, such as State-Space Models (e.g., Mamba [6]), and have also clarified that multimodal models often feature hybrid architectures that combine different model classes.
> > >
> > > ---
> > >
> > > Reference:
> > >
> > > [1] Devlin, Jacob, et al. "Bert: Pre-training of deep bidirectional transformers for language understanding." Proceedings of the 2019 conference of the North American chapter of the association for computational linguistics: human language technologies, volume 1 (long and short papers). 2019.
> > >
> > > [2] Radford, Alec, et al. "Improving language understanding by generative pre-training." (2018): 3.
> > >
> > > [3]. Raffel, Colin, et al. "Exploring the limits of transfer learning with a unified text-to-text transformer." Journal of machine learning research 21.140 (2020): 1-67.
> > >
> > > [4]. Radford, Alec, et al. "Learning transferable visual models from natural language supervision." International conference on machine learning. PmLR, 2021.
> > >
> > > [5]. Liu, Haotian, et al. "Visual instruction tuning." Advances in neural information processing systems 36 (2023): 34892-34916.
> > >
> > > [6]. Gu, Albert, and Tri Dao. "Mamba: Linear-time sequence modeling with selective state spaces." arXiv preprint arXiv:2312.00752 (2023).

---

> > > > ### Author Response · Authors · 2025-08-14
> > > > **Response to Reviewer 8MQQ (4/N)**
> > > >
> > > > # Section 3
> > > >
> > > > We apologize again for the delay in our response, which was due to numerous further modifications. We would like to thank the AE and all reviewers again for their valuable feedback, which has greatly improved our manuscript. Please find our point-by-point responses for the remaining sections to Reviewer 8MQQ.
> > > >
> > > >
> > > > > Q1:  Table 1 could profit from providing sources for these definitions, or comparing definitions from different works.
> > > >
> > > > A1: Thank you for your suggestion. We have added the statement: “These categories draw inspiration from Gallegos et al. (2024) and have been further refined.” in the caption of Table 1.
> > > >
> > > > > Q2: The same applies to the definition of bias in the paragraph of the same name on page 11 (or at least it should be made clear that this is the definition of bias by the authors).
> > > >
> > > > A2: Thank you for your suggestion. We have added a reference to the prior work that inspired our definition of bias.
> > > >
> > > > > Q3: The notation is equation (5) should be harmonized with the notation introduced in section 2 (same holds with the rest of the notation in this section)
> > > >
> > > > A3: Thank you for your suggestion. We have revised Section 2 to use $\mathcal{X}$ and $\mathcal{Y}$ to represent input and output across all model types, while mentioning their modality type later, ensuring consistency between Sections 2 and 3.
> > > >
> > > > > Q4:  If a measurement function maps to a scalar score, then it should be written as $S: y\rightarrow \mathbb{R}$
> > > >
> > > > A4: Thank you for your suggestion. Since Y denotes the model output rather than a data type, we have retained the current notation but now explicitly state $s \in \mathbb{R}$ to clarify its type.
> > > >
> > > > > Q5: "It is noteworthy that this is just one possible definition of bias, while other definitions or metrics can also be reasonable" -> mention or at least cite papers with other defintions or metrics here.
> > > >
> > > > A5: Thank you for your suggestion. We have added citations to works presenting alternative definitions and metrics of bias.
> > > >
> > > > > Q6:  In the following paragraph, a set of principles by Gallegos et al. (2023) are named, but not discussed or not explained at all.
> > > >
> > > > A6: Thank you for your suggestion. We have now included a brief discussion of the principles proposed by Gallegos et al. (2023).
> > > >
> > > > > Q7: Equations (6) - (10): Use \text for abbreviations / words in equations
> > > >
> > > > A7: Thank you for pointing this out. We have updated all abbreviations and words in equations (6)–(10) to use \text{}.
> > > >
> > > > > Q8: "Similarly, Demographic Representation (DR), as discussed in" -> use \citet for citations
> > > >
> > > > A8: Thank you for your suggestion. We have replaced the citation with \citet as recommended.
> > > >
> > > > > Q9: The paragraph "Methods based on Token Selection Probability" talks about masked-language models that are encoder-only, which were not included as foundation models in section 2.
> > > >
> > > > A9: Thank you for your suggestion. We have already included encoder-only masked language models in Section 2 as one kind of LLMs.
> > > >
> > > > > Q10: Section 3.3: "For specific categories that are extensively discussed in later chapter." is missing some words.
> > > >
> > > > A10: Thank you for pointing this out. We have revised the sentence in Section 3.3 for completeness.
> > > >
> > > > > Q11:  "However, it is essential to recognize that biases in the training data and the training process are not inherently designed to understand or priorize human values." I think I understand the point that the authors aim to make here, but it is a bit oddly worded. It also obfuscates that some of the biases in training data originate from human design decisions based on their own values and biases (see [5]).
> > > >
> > > > A11: hank you for your suggestion. We have reworded the sentence to clarify the intended meaning and to note that some biases arise from human design choices based on personal values.
> > > >
> > > > > Q12: The mention of chain-of-thought at the start of page 16 should come with a citation.
> > > >
> > > > A12: Thank you for your suggestion. We have added a citation to support the mention of chain-of-thought.
> > > >
> > > > > Q13: Definting the toxic text in the next line with the variable "t" is redundant, since it is not used.
> > > >
> > > > A13: Thank you for pointing this out. We have removed the redundant definition of “t”.

---

> > > > > ### Author Response · Authors · 2025-08-14
> > > > > **Response to Reviewer 8MQQ (5/N)**
> > > > >
> > > > > # Section 4 (1/2)
> > > > >
> > > > > > Q1: At the end of the second paragraph 20, the word "propose" is used three times right after another.
> > > > >
> > > > > A1: Thank you for pointing this out. We have revised the sentence to reduce repetition, vary the vocabulary, and improve the flow.
> > > > >
> > > > > > Q2: "Then, a series of works [...] propose to use well-suited rules, metrics, and LLM-based methods for data clearning" -> This should be explained further.
> > > > >
> > > > > A2: Thank you for this valuable suggestion. We have added concrete examples of these data cleaning techniques to provide greater clarity.
> > > > >
> > > > > > Q3: At the end of the third paragraph, the authors could include [6], which uses bootstrapped conversations for task-oriented dialogue finetuning.
> > > > >
> > > > > A3: Thank you for the helpful reference. We have included and discussed reference [6] accordingly.
> > > > >
> > > > > > Q4: Another caveat to add to 4.1 could be [7], showing that overtrained models are harder to finetune.
> > > > >
> > > > > A4: We appreciate this suggestion. We have added a discussion of this caveat and cited reference [7] in the concluding paragraph of the SFT section to highlight this limitation.
> > > > >
> > > > > > Q5: Section 4.2 claims that one stage of RLHF is instruction finetuning with SFT. I would intuitively rather claim that the outcome of the instruction finetuning step is used for RLHF, which is a separate step.
> > > > >
> > > > > A5: We agree with your perspective and have rephrased the description of the RLHF process to clarify that SFT is a prerequisite step whose output serves as the initial policy for RLHF.
> > > > >
> > > > > > Q6: p.22: TEXTGAN should be cited
> > > > >
> > > > > A6: Thank you for noticing this omission. We have added the appropriate citation for TextGAN.
> > > > >
> > > > > > Q7: Zhu et al. (2023b) citation should use \citep instead of \citet.
> > > > >
> > > > > A7: Thank you. We have corrected the citation format from \citet to \citep as suggested.
> > > > >
> > > > > > Q8: The part about diverse user preference should also discuss the cultural component, for instance through [8].
> > > > >
> > > > > A8: This is an excellent point. We have expanded the discussion on diverse user preferences to include the cultural component and have incorporated the suggested reference [8].
> > > > >
> > > > > > Q9: The end of section 4.2 could profit from providing an overall discussion on when which RLHF objective should ideally be used.
> > > > >
> > > > > A9: Thank you for this feedback. We have added a concluding paragraph to Section 4.2 summarizing the trade-offs between different preference optimization objectives and providing guidance on their ideal use cases.
> > > > >
> > > > > > Q10: The "Discrete Prompts" paragraph on p. 25 should discuss also the brittleness of models w.r.t prompts, for instance discussed in [9].
> > > > >
> > > > > A10: We agree this is an important point to include. We have added a discussion on the brittleness of discrete prompts and cited reference [9].
> > > > >
> > > > > > Q11: The "Chain-of-Thought" paragraph should also include a discussion about how reasoning chains do not necessarily need to reflect the actual reason a model arrived at a solution [10].
> > > > >
> > > > > A11: Thank you for raising this critical point. We have added a sentence to the “Chain-of-Thought” paragraph discussing the potential for unfaithful reasoning, citing reference [10].

---

> > > > > > ### Author Response · Authors · 2025-08-14
> > > > > > **Response to Reviewer 8MQQ (6/N)**
> > > > > >
> > > > > > # Section 4 (2/2)
> > > > > >
> > > > > > > Q12: The paragraph on "Prompt Optimization" could refer to [11].
> > > > > >
> > > > > > A12: Thank you for the reference. We have incorporated reference [11] into our discussion on prompt optimization to address the aspect of risk control.
> > > > > >
> > > > > > > Q13: The same paragraph includes '(i.e., agent workflow (Zhou et al., 2023d))', which could be formatted better with \citealp.
> > > > > >
> > > > > > A13: Thank you for catching this formatting issue. We have corrected the nested citations using \citealp as recommended.
> > > > > >
> > > > > > > Q14: The reference to "visual instruction tuning" in section 4.4 needs a citation.
> > > > > >
> > > > > > A14: Thank you. We have added a citation (LLaVA) for “visual instruction tuning.”
> > > > > >
> > > > > > > Q15: "dis-preferred" -> "dispreferred"
> > > > > >
> > > > > > A15: Thank you. We have corrected the typo to “dispreferred.”
> > > > > >
> > > > > > > Q16: The "superalignment problem" at the beginning of section 4.5 should be explained more.
> > > > > >
> > > > > > A16: We agree this concept requires more explanation. We have expanded the definition of the “superalignment problem” at the beginning of Section 4.5 to provide additional context.
> > > > > >
> > > > > > > Q17: Redundant whitespace before "Casper et al. (2023)" in section 4.5.
> > > > > >
> > > > > > A17: Thank you. We have removed the redundant whitespace.
> > > > > >
> > > > > > > Q18: "For example, a reward model with uniform values may make the training of foundation models more scyphantic rather than maintaining integrity" - this argument is not very clear.
> > > > > >
> > > > > > A18: Thank you for pointing out the lack of clarity. We have rewritten the sentence to more clearly explain how a simplistic reward model can lead to sycophantic behavior.
> > > > > >
> > > > > > > Q19: "In the superalignment situation, many research works find that strong models can easily manipulate and deceive weak models" - no citations or evidence for this is given.
> > > > > >
> > > > > > A19: Thank you for highlighting this. We have added appropriate citations to support the claim.
> > > > > >
> > > > > > > Q20: "making evaluation super difficult" -> "making evaluation quite difficult" (or some variation of this)
> > > > > >
> > > > > > A20: Thank you. We have revised the phrasing to “making evaluation quite difficult” for a more formal tone.
> > > > > >
> > > > > > > Q21: "And this can leave backdoor vulnearabilities in the model, which can be exploited by adversaries" - references?
> > > > > >
> > > > > > A21: We appreciate you flagging this. We have added references to support the statement about backdoor vulnerabilities.
> > > > > >
> > > > > > > Q22: "Ensemble methods [...] can be a potential solution of scalable oversight to overcome the weakness of a single weak model." - references in the context of alignment?
> > > > > >
> > > > > > A22: Thank you. We have added citations linking ensemble methods to scalable oversight in the context of alignment.

---

> > > > > > > ### Author Response · Authors · 2025-08-14
> > > > > > > **Response to Reviewer 8MQQ (7/N)**
> > > > > > >
> > > > > > > # Section 5
> > > > > > >
> > > > > > > > Q1: Section 5.1.1 should also summarize some evidence from the literature on why and how jailbreaks work.
> > > > > > >
> > > > > > > A1: Thank you for the insightful suggestion. We have revised Section 5.1.1 to include additional literature discussing the reasons and mechanisms behind jailbreaks.
> > > > > > >
> > > > > > > > Q2: The "Prompt Injection Attacks" paragraph would profit from examples.
> > > > > > >
> > > > > > > A2: Thank you for your suggestion. We have added concrete examples of prompt injection attacks along with one additional referenced paper.
> > > > > > >
> > > > > > > > Q3: "Shi et al. (2023a) and Rando & Tramèr (2024) explore the possibiltiy of injecting backdoors into the reward model during the RLHF process" - so what would an actual attack in the end look like?
> > > > > > >
> > > > > > > A3: Thank you for pointing this out. We have supplemented the text with more specific descriptions of what such an actual attack could look like in practice.
> > > > > > >
> > > > > > > > Q4: Section 5.2: The "projected gradient descent attack" is mentioned, but no intuition / explanation of the attack is given.
> > > > > > >
> > > > > > > A4: Thank you for the inquiry. We have added detailed explanations and intuitive descriptions of the PGD attack.
> > > > > > >
> > > > > > > > Q5: The "IDEFICS mechanism" is mentioned on p. 31, but not explained. The same goes for "2-hop prompt templates".
> > > > > > >
> > > > > > > A5: Thank you for pointing this out. We have revised the subsection to include clear and detailed explanations of both the IDEFICS mechanism and 2-hop prompt templates.
> > > > > > >
> > > > > > > > Q6: The LAION-5B dataset in 5.3.1 should be cited.
> > > > > > >
> > > > > > > A6: Thank you for the advice. We have added references for both the LAION-5B and LAION-400M datasets.
> > > > > > >
> > > > > > > > Q7: (1) "(Wang et al., 2024s)" should be \citet and not \citep. (2) p 33: "Yoon et al. (Yoon et al., 2024)" should just be \citet. (3) "(Yang et al., 2024e)" -> \citet. (4) Section 5.4.1: "Chen et al." -> \citet. (5) "Model Fine-Tuning" -> "Model finetuning".
> > > > > > >
> > > > > > > A7: Thank you for identifying these issues. We have corrected all LaTeX citation formats and text typos as specified.
> > > > > > >
> > > > > > > > Q8: The "assimilation phenomenon" is not explained.
> > > > > > >
> > > > > > > A8: Thank you for the feedback. We have revised the paragraph to provide a more concrete description of the assimilation phenomenon, along with relevant literature.
> > > > > > >
> > > > > > > > Q9: Section 5.4.2 seems to be missing a discussion using external models for safety, see e.g. [12].
> > > > > > >
> > > > > > > A9: Thank you for this suggestion. We have rewritten and expanded Section 5.4.2 to discuss defense strategies that leverage external symbolic or classification systems, incorporating and citing reference [12].

---

> > > > > > > > ### Author Response · Authors · 2025-08-14
> > > > > > > > **Response to Reviewer 8MQQ (8/N)**
> > > > > > > >
> > > > > > > > # Section 6
> > > > > > > >
> > > > > > > > > Q1: The reference to homomorphic encryption should be better explained and connected to the rest of section or be omitted.
> > > > > > > >
> > > > > > > > A1: Thank you for the insight. We have added more explanations for the reference to homomorphic encryption.
> > > > > > > >
> > > > > > > > > Q2: Section 6.1.1 contains a reference to BERT, which is not cited and not introduced in section 1 as a foundation model. The MIA acronym is also not introduced.
> > > > > > > >
> > > > > > > > A2: We have revised the BERT reference in Section 6.1.1 and Section 1, and have added the full name for the MIA acronym.
> > > > > > > >
> > > > > > > > > Q3: "MIN-K% PROB" attack should be explained, at least in broad terms.
> > > > > > > >
> > > > > > > > A3: We have introduced additional descriptions for the MIN-K% PROB attack.
> > > > > > > >
> > > > > > > > > Q4: Section 6.1.2: "the field has shifted towards developing various techniques to preserve privacy LLMs" - some word seems to be missing here.
> > > > > > > >
> > > > > > > > A4: Agreed. We have corrected the grammatical mistake in this sentence.
> > > > > > > >
> > > > > > > > > Q5: Section 6.1.2 also maybe warrants a brief introduction into how and why differential privacy works in general.
> > > > > > > >
> > > > > > > > A5: Thank you for the valuable suggestion. We have added detailed descriptions of how and why differential privacy works in this section.
> > > > > > > >
> > > > > > > > > Q6: Section 6.2: "Li et al. (2024m) studied membership inference against LMMs" - reads very vague.
> > > > > > > >
> > > > > > > > A6: We have provided additional descriptions on the multimodality of this membership inference method.
> > > > > > > >
> > > > > > > > > Q7: Section 6.3: "Ghalebikesabi et al. (Ghalebikesabi et al., 2023)" -> use \citet.
> > > > > > > >
> > > > > > > > A7: We have corrected the LaTeX citation format as suggested.
> > > > > > > >
> > > > > > > > > Q8: "... offering enhanced privacy protections" - what protections? By doing what?
> > > > > > > >
> > > > > > > > A8: Thank you for pointing this out. We have augmented the sentence with more concrete details about the protections offered by the referenced method.
> > > > > > > >
> > > > > > > > > Q9: Section 6.4.1: "For example, certain attacks require poisoning or injecting triggers..." - What kind of attacks?
> > > > > > > >
> > > > > > > > A9: We have added more details to elaborate on the challenges of privacy backdoor attacks in this paragraph.
> > > > > > > >
> > > > > > > > > Q10: Section 6.4.2: "Integrating DP into the foundation model fine-tuning still leads to significant performance degredation" - references? Same with the following paragraph "However, these methods also suffer from significant performance degradation".
> > > > > > > >
> > > > > > > > A10: Thank you for the valuable question. We have added four additional references to support the observations stated in these paragraphs.

---

> > > > > > > > > ### Author Response · Authors · 2025-08-14
> > > > > > > > > **Response to Reviewer 8MQQ (9/N)**
> > > > > > > > >
> > > > > > > > > # Section 7
> > > > > > > > >
> > > > > > > > > > Q1: If the definition of hallucination is by the authors, this should be made more explicit. It would also be interesting to compare the definition to others in the literature.
> > > > > > > > >
> > > > > > > > > A1: Thank you for the valuable insight. We have revised the beginning paragraph with a more explicit statement on the definition and discussed additional references from past reviews on hallucinations.
> > > > > > > > >
> > > > > > > > > > Q2: Figure 12 should also be referenced in 7.1.1, since it depicts the different kinds of hallucinations that are discussed there.
> > > > > > > > >
> > > > > > > > > A2: We have added the reference to Figure 12 in Section 7.1.1.
> > > > > > > > >
> > > > > > > > > > Q3: Is "Instruction Deviation" really a type of hallucination, or just an indicator of an erroneous model?
> > > > > > > > >
> > > > > > > > > A3: Thank you for pointing this out. We use the term "Instruction Deviation" to emphasize that an LLM can be hallucinatory when its instruction-following capability is too weak to align with the user input. However, we agree that this term is not fully suitable for categorizing faithfulness hallucination. To improve clarity and coherence, we have (1) revised the categorized descriptions of Faithfulness Hallucination to align with its introductory sentences, (2) added a reference and example to explain instruction following and faithfulness hallucination, and (3) fixed related typos.
> > > > > > > > >
> > > > > > > > > > Q4: Section 7.2: "For example, hallucinations in LMMs are often observed in which the model predicts possible objects or actions that are not supported by the image but are plausible from commensense" - references?
> > > > > > > > >
> > > > > > > > > A4: We have added an additional reference to support the example.
> > > > > > > > >
> > > > > > > > > > Q5: The nested parentheses due to citations in the "Knowledge Boundaries" point could be mitigated through \citealp.
> > > > > > > > >
> > > > > > > > > A5: We have fixed the nested parentheses as suggested.
> > > > > > > > >
> > > > > > > > > > Q6: "Path Dependence. The autoregressive nature of LLMs and LMMs causes hallucination." What is the evidence that this is because the models are autoregressive? Are diffusion models actually autoregressive?
> > > > > > > > >
> > > > > > > > > A6: Thank you for raising this point. In the “Path Dependence” bullet point, we intend to convey that autoregressive models suffer from error propagation and self‐consistency failures due to their token‐by‐token prediction nature. Autoregressive LLMs generate each token based on all previously generated tokens, so any initial error compounds in subsequent predictions. Recent works show exponential decay in output correctness with sequence length, where a constant per‐token error rate leads to rapid deterioration of generation quality (https://arxiv.org/abs/2505.24187). Inconsistencies with earlier content are also common when hallucinations occur (https://aclanthology.org/2025.semeval-1.38/). While score-space denoising diffusion models are indeed autoregressive in score prediction, we limit our discussion to representative next‐token prediction language models. To address your concern, we have (1) revised the text to be more specific about error propagation and self‐consistency failures, (2) specified the scope to Autoregressive LLMs and MLLMs, and (3) added the two cited references.
> > > > > > > > >
> > > > > > > > > > Q7: The title of 7.3.1 should perhaps be "Detecting language hallucinations"?
> > > > > > > > >
> > > > > > > > > A7: We have fixed the title as suggested.
> > > > > > > > >
> > > > > > > > > > Q8: The citation of "Jiang et al. (2024b)" should use \citep.
> > > > > > > > >
> > > > > > > > > A8: We have fixed the LaTeX citation format as suggested.
> > > > > > > > >
> > > > > > > > > > Q9: Monte Carlo dropout in the "uncertainty estimation" needs a citation.
> > > > > > > > >
> > > > > > > > > A9: We have added the relevant reference to support the term.
> > > > > > > > >
> > > > > > > > > > Q10: p. 44 second paragraph: Llama is missing a citation. Also, I believe RAG is neither introduced as an acronym nor cited.
> > > > > > > > >
> > > > > > > > > A10: We have added references for Llama and RAG, and introduced the full name of RAG.
> > > > > > > > >
> > > > > > > > > > Q11: Title for section 7.4.1 should include an s in "hallucinations".
> > > > > > > > >
> > > > > > > > > A11: We have fixed the title as suggested.
> > > > > > > > >
> > > > > > > > > > Q12: "Training Approaches" paragraph in 7.4.2: "best-of-n rejection sampling" is not really explained.
> > > > > > > > >
> > > > > > > > > A12: We have added a more detailed explanation of the term "best-of-n rejection sampling".
> > > > > > > > >
> > > > > > > > > > Q13: "Post-processing Approaches" on p. 46: It is not really clear to me as a reader how hidden Markov models and the Viterbi algorithm are applied in this context.
> > > > > > > > >
> > > > > > > > > A13: We have provided additional descriptions of the referenced method to improve clarity.
> > > > > > > > >
> > > > > > > > > > Q14: "Decoding Approaches" paragraph mentions "VCD", but the acronym is not explained and the approach is not explained very intuitively.
> > > > > > > > >
> > > > > > > > > A14: We have revised the term usage for Visual Contrastive Decoding and expanded the explanation for better understanding.
> > > > > > > > >
> > > > > > > > > > Q15: Section 7.5 might also want to include a discussion on refusal in models in order to avoid hallucinations.
> > > > > > > > >
> > > > > > > > > A15: Thank you for the suggestion. We have included additional references on refusal mechanisms for reducing hallucination and added a discussion suggesting future research directions.

---

> > > > > > > > > > ### Author Response · Authors · 2025-08-14
> > > > > > > > > > **Response to Reviewer 8MQQ (10/N)**
> > > > > > > > > >
> > > > > > > > > > # Section 8
> > > > > > > > > >
> > > > > > > > > > > Q1: "Uncertainty" as a concept in the context of foundation models isn't really well defined.
> > > > > > > > > >
> > > > > > > > > > A1: Thank you for pointing this out. We have added a definitional paragraph at the beginning of Section 8 to clearly state what we mean by *uncertainty* in the context of foundation models. Specifically, we define uncertainty as the degree of confidence in the model’s predictions or generated content, and describe two major sources: **epistemic uncertainty**, which stems from incomplete or imperfect knowledge encoded during training (e.g., data gaps or model misspecification), and **aleatoric uncertainty**, which arises from inherent randomness or ambiguity in the data itself (e.g., noisy inputs or ambiguous task definitions). We also explicitly acknowledge that this distinction has limitations, and we cite relevant works \citep{baan2023uncertainty, gruber2023sources, ulmer2024uncertainty, mucsanyi2024benchmarking}.
> > > > > > > > > >
> > > > > > > > > > > Q2: Section 8.1: The aleatoric / epistemic distinction has recently attracted a lot of critique due to its rigidity and reductiveness, see [13–16].
> > > > > > > > > >
> > > > > > > > > > A2: We have added a cautionary note in Section 8 to acknowledge the critiques highlighted in [13–16] regarding the rigidity and reductiveness of the aleatoric/epistemic distinction. In our revision, we clarify that this distinction is adopted as a heuristic device to help organize our taxonomy, and we explicitly caution against interpreting it as an absolute or ontological truth.
> > > > > > > > > >
> > > > > > > > > > > Q3: Last paragraph before 8.1.2: "under-prescribed" -> "underspecified"?
> > > > > > > > > >
> > > > > > > > > > A3: We have replaced the term “under-prescribed” with “underspecified” to improve clarity and accuracy of expression in the context.
> > > > > > > > > >
> > > > > > > > > > > Q4: Critique of optimizing only observed sequence probability in 8.1.2 could be supplemented with [17, 18].
> > > > > > > > > >
> > > > > > > > > > A4: We have supplemented our critique in Section 8.1.2 by incorporating references [17, 18], which provide additional theoretical and empirical perspectives supporting our point regarding the limitations of optimizing only the observed sequence probability.
> > > > > > > > > >
> > > > > > > > > > > Q5: Section 8.2: “Methods have been developed to recalibrate uncertainty scores to match the probability of an answer being correct” -> Not completely correct; you calibrate confidence, not e.g., entropy.
> > > > > > > > > >
> > > > > > > > > > A5: We have revised the relevant sentences in Section 8.2 to clarify that calibration methods focus on **confidence scores** rather than uncertainty measures such as entropy. The updated text now states that methods have been developed to recalibrate confidence scores so that they more accurately reflect the true probability of an answer being correct. We also note that uncertainty estimates can be applied to determine when the system should abstain from answering or seek additional clarification before responding.
> > > > > > > > > >
> > > > > > > > > > > Q6: Section 8.2.2: Many more alternatives to the ECE have been proposed, see for instance [14] p. 37.
> > > > > > > > > >
> > > > > > > > > > A6: We have expanded Section 8.2.2 to include a broader range of calibration metrics beyond the Expected Calibration Error (ECE). In particular, we now mention **Maximum Calibration Error (MCE)** \citep{naeini2015obtaining, guo2017calibration}, **Brier Score** \citep{brier1950verification}, **negative log-likelihood** \citep{hastie2001elements}, and other novel calibration variants \citep{ulmer2024uncertainty}.
> > > > > > > > > >
> > > > > > > > > > > Q7: Section 8.2.5: “(Yadkori et al., 2024b)” should use \citet.
> > > > > > > > > >
> > > > > > > > > > A7: We have corrected the citation format to use `\citet{yadkori2024b}` instead of the parenthetical style, ensuring consistency with our citation guidelines.
> > > > > > > > > >
> > > > > > > > > > > Q8: Section should include [19–21].
> > > > > > > > > >
> > > > > > > > > > A8: We have added references [19–21] at the beginning of Section 8.2 to provide a more comprehensive background for the discussion and to better situate our analysis within the broader literature.
> > > > > > > > > >
> > > > > > > > > > > Q9: Section 8.3: Long-form UQ has actually been explored in works such as [22–24].
> > > > > > > > > >
> > > > > > > > > > A9: We have revised Section 8.3 to acknowledge prior work on long-form uncertainty quantification as discussed in [22–24]. The new paragraph summarizes their settings (e.g., long-form reasoning or generation tasks), the uncertainty signals used (such as self-consistency variance, verifier-based confidence, and stepwise attribution), and explains how these approaches align with our interaction-level category.
> > > > > > > > > >
> > > > > > > > > > > Q10: p. 55 "cross entropy" -> "cross-entropy"
> > > > > > > > > >
> > > > > > > > > > A10: We have corrected the typographical error by changing “cross entropy” to “cross-entropy” to ensure consistent terminology throughout the manuscript.
> > > > > > > > > >
> > > > > > > > > > > Q11: Use \citealp for nested citation “(Kadavath et al., 2022; Tian et al., 2023b)”.
> > > > > > > > > >
> > > > > > > > > > A11: We have updated the nested citation to “(\citealp{kadavath2022}; \citealp{tian2023b})” and ensured that similar nested citations throughout Section 8 follow this formatting for consistency.

---

> > > > > > > > > > > ### Author Response · Authors · 2025-08-14
> > > > > > > > > > > **Response to Reviewer 8MQQ (11/N)**
> > > > > > > > > > >
> > > > > > > > > > > # Section 9 (1/2)
> > > > > > > > > > >
> > > > > > > > > > > > Q1: There are no citations given for the definition of shifts in Section 9.1. Candidates could be [27–29].
> > > > > > > > > > >
> > > > > > > > > > > A1: Thank you for pointing this out. We have added the suggested citations [27–29] in Section 9.1 to support our definition of shifts and to situate our usage within established literature.
> > > > > > > > > > >
> > > > > > > > > > > > Q2: Motivate why OOD detection in Section 9.2 still matters for extremely large foundation models.
> > > > > > > > > > >
> > > > > > > > > > > A2: We have added a motivation paragraph at the beginning of Section 9.2. This addition explains that, despite the impressive generalization capabilities of extremely large foundation models, their performance is still bounded by the coverage of their training data. In dynamic open-world environments, they remain vulnerable to domain shifts, rare edge cases, and adversarial inputs that can lead to overconfident yet incorrect predictions. We emphasize that OOD detection remains critical as a safeguard against catastrophic failures in high-stakes applications (e.g., medicine, autonomous driving), as a mechanism to trigger human oversight, as a driver for active learning, and as a means to preserve user trust in automated decision-making systems.
> > > > > > > > > > >
> > > > > > > > > > > > Q3: The section is missing [28].
> > > > > > > > > > >
> > > > > > > > > > > A3: We have added citation [28] in Section 9.2 and referenced it explicitly when introducing near/far OOD evaluation.
> > > > > > > > > > >
> > > > > > > > > > > > Q4: Section 9.2: “OOD peer classes” unclear; near/far/fine-grained OOD not explained.
> > > > > > > > > > >
> > > > > > > > > > > A4: We have rewritten the definitions in Section 9.2 to clearly explain the concept of OOD peer classes, as well as to define and contextualize near-OOD, far-OOD, and fine-grained OOD with illustrative examples.
> > > > > > > > > > >
> > > > > > > > > > > > Q5: Section 9.3 lacks references for strategies listed.
> > > > > > > > > > >
> > > > > > > > > > > A5: We have added literature references for all the OOD generalization strategies that predate the foundation model era, ensuring each listed approach is properly supported by prior work.
> > > > > > > > > > >
> > > > > > > > > > > > Q6: Section 9.3.1: “Easy Data Augmentation” isn’t explained.
> > > > > > > > > > >
> > > > > > > > > > > A6: We have added an explanation of Easy Data Augmentation (EDA), describing its common operations (e.g., synonym replacement, random insertion, random swap, and random deletion) and discussing why these techniques may be less effective or fail for LLMs compared to earlier NLP models.
> > > > > > > > > > >
> > > > > > > > > > > > Q7: Section 9.3.4: Not clear what the different loss functions do.
> > > > > > > > > > >
> > > > > > > > > > > A7: We have clarified the invariant learning loss functions mentioned in Section 9.3.4, including Focal Loss, Dice Loss, and Mixup. For each, we explain its mechanism, how it differs from the others, and why it can improve robustness in OOD generalization.
> > > > > > > > > > >
> > > > > > > > > > > > Q8: Section 9.3.5: Model ensembles need citations, e.g., [29, 30].
> > > > > > > > > > >
> > > > > > > > > > > A8: We have added historical citations for model ensemble methods ([lakshminarayanan2017simple], [breiman1996bagging], [freund1997decision]) as well as relevant recent work on LLM ensembles to provide both foundational and contemporary perspectives.
> > > > > > > > > > >
> > > > > > > > > > > > Q9: p. 60: “In-Context” -> “in-context”.
> > > > > > > > > > >
> > > > > > > > > > > A9: We have corrected the capitalization from “In-Context” to “in-context” for consistency with the rest of the manuscript.

---

> > > > > > > > > > > > ### Author Response · Authors · 2025-08-14
> > > > > > > > > > > > **Response to Reviewer 8MQQ (12/N)**
> > > > > > > > > > > >
> > > > > > > > > > > > Section 9 (2/2)
> > > > > > > > > > > >
> > > > > > > > > > > > > Q10: Second paragraph p. 60: Missing citations for T5 and ViT.
> > > > > > > > > > > >
> > > > > > > > > > > > A10: We have added the canonical citations for T5 and ViT at their first mention in Section 9.3.
> > > > > > > > > > > >
> > > > > > > > > > > > > Q11: The last paragraph declares SFT as a stage of RLHF, but then contrasts RLHF and SFT.
> > > > > > > > > > > >
> > > > > > > > > > > > A11: We have revised the text to explicitly note that SFT is both a component of RLHF and a stage that can be evaluated independently. The RLHF versus SFT comparison is now framed as a contrast between the full RLHF pipeline and an SFT-only approach, while retaining all original technical details and citations.
> > > > > > > > > > > >
> > > > > > > > > > > > > Q12: Section 9.4.3 may miss activation steering/finetuning such as [31, 32].
> > > > > > > > > > > >
> > > > > > > > > > > > A12: We have added works on activation steering and fine-tuning [31, 32] in a dedicated paragraph before multimodal methods. In this addition, we explain what activation steering is, how Joint Localization and Activation Editing operates, how Activation Addition works, and why these methods may enhance OOD generalization. The presentation is organized to flow logically from general fine-tuning → parameter-efficient fine-tuning (PEFT) → activation steering → multimodal fine-tuning.
> > > > > > > > > > > >
> > > > > > > > > > > > > Q13: p. 63: Unsure “FFN” was previously introduced.
> > > > > > > > > > > >
> > > > > > > > > > > > A13: We have added the full term “feed-forward network (FFN)” at its first occurrence and used the acronym consistently thereafter.
> > > > > > > > > > > >
> > > > > > > > > > > > > Q14: Use \citealp for nested citation “(Cao et al., 2021)”.
> > > > > > > > > > > >
> > > > > > > > > > > > A14: We have updated the formatting to use \citealp for the nested citation, ensuring consistency with our citation style.
> > > > > > > > > > > >
> > > > > > > > > > > > > Q15: “Knowledge-based Model Editing” is missing a citation for integrated gradients.
> > > > > > > > > > > >
> > > > > > > > > > > > A15: We have added the canonical citation for integrated gradients at its first mention and linked it to the discussion on attribution-guided edit localization.
> > > > > > > > > > > >
> > > > > > > > > > > > > Q16: ROME and MEMIT aren’t really explained.
> > > > > > > > > > > >
> > > > > > > > > > > > A16: We have added mechanistic explanations for both ROME and MEMIT. For ROME, we describe the causal tracing process, layer localization, and the rank-one update to the value projection. For MEMIT, we explain it as a multi-layer extension of ROME, enabling parallel fact editing via a closed-form solution.
> > > > > > > > > > > >
> > > > > > > > > > > > > Q17: End of 9.4.5 could compare model editing methods.
> > > > > > > > > > > >
> > > > > > > > > > > > A17: We have added a structured comparison of the four main model editing paradigms, summarizing their mechanisms, advantages, and disadvantages. We have also included MLLM-specific implications to tie the comparison back to Section 9.4.5 and concluded with a brief discussion of potential hybrid strategies as a forward-looking perspective.

---

> > > > > > > > > > > > > ### Author Response · Authors · 2025-08-14
> > > > > > > > > > > > > **Response to Reviewer 8MQQ (13/N)**
> > > > > > > > > > > > >
> > > > > > > > > > > > > # Section 10
> > > > > > > > > > > > >
> > > > > > > > > > > > > > Q1: Section 10.1.2: BERT is spelled Bert and is missing citations.
> > > > > > > > > > > > >
> > > > > > > > > > > > > A1: Thank you for pointing this out. We have corrected the spelling of “BERT” in Section 10.1.2 and added the appropriate canonical citations to ensure both accuracy and completeness.
> > > > > > > > > > > > >
> > > > > > > > > > > > > > Q2: Applications to decoder-only models for gradient-based explanation are missing?
> > > > > > > > > > > > >
> > > > > > > > > > > > > A2: We have added references to recent work that explores gradient-based explanation methods for decoder-only models at the end of Section 10.1.2.
> > > > > > > > > > > > >
> > > > > > > > > > > > > > Q3: Section 10.1.3 is missing [33].
> > > > > > > > > > > > >
> > > > > > > > > > > > > A3: We have incorporated reference [33] into Section 10.1.3, along with a corresponding explanation of its relevance, to ensure comprehensive coverage of related literature.
> > > > > > > > > > > > >
> > > > > > > > > > > > > > Q4: Section 10.2.1 is missing early work of [34].
> > > > > > > > > > > > >
> > > > > > > > > > > > > A4: We have added reference [34] and a brief discussion of its contributions in Section 10.2.1 to acknowledge early foundational work in this area.
> > > > > > > > > > > > >
> > > > > > > > > > > > > > Q5: Last paragraph in 10.2.1 could also add [35, 36, 37].
> > > > > > > > > > > > >
> > > > > > > > > > > > > A5: We have included references [35, 36, 37] in the final paragraph of Section 10.2.1, with brief explanations of each work to contextualize their contributions.
> > > > > > > > > > > > >
> > > > > > > > > > > > > > Q6: Referencing the key-value memory in transformers in 10.2.3 could be a reason to give a brief intro into the architecture in the earlier sections.
> > > > > > > > > > > > >
> > > > > > > > > > > > > A6: We have added a brief introduction to the key-value memory mechanism and other relevant components at the beginning of Section 10.2.3 to provide readers with the necessary architectural context for the subsequent discussion.
> > > > > > > > > > > > >
> > > > > > > > > > > > > > Q7: Section 10.3.1 is missing an explanation for / references about (the use of) influence functions. Another work to add is also [38].
> > > > > > > > > > > > >
> > > > > > > > > > > > > A7: We have expanded Section 10.3.1 to include an explanation of influence functions, their applications, and key references in this area. We have also added citation [38] to capture recent developments.
> > > > > > > > > > > > >
> > > > > > > > > > > > > > Q8: Section 10.3.3 feels superficial giving the volume of work trying to understand the role of ICL, see for instance.
> > > > > > > > > > > > >
> > > > > > > > > > > > > A8: We have extended Section 10.3.3 by adding several relevant references and explanations that provide a deeper and more balanced discussion of the role and mechanisms of in-context learning (ICL).
> > > > > > > > > > > > >
> > > > > > > > > > > > > > Q9: Section 10.4: Use \citealp for "(Jacovi & Goldberg, 2020)" citation.
> > > > > > > > > > > > >
> > > > > > > > > > > > > A9: We have replaced the original `\citet` with `\citealp` for the citation of Jacovi & Goldberg (2020) to follow the correct formatting style.
> > > > > > > > > > > > >
> > > > > > > > > > > > > > Q10: Section 10.4.1: "Another benchmark Hate Explain (Mathew et al., 2021) ask the annotators..." -> "Another benchmark, Hate Explain (Mathew et al., 2021), asks the annotators..."
> > > > > > > > > > > > >
> > > > > > > > > > > > > A10: We have revised the sentence for grammatical correctness, which now reads: “Another benchmark, Hate Explain (Mathew et al., 2021), asks the annotators...”.
> > > > > > > > > > > > >
> > > > > > > > > > > > > > Q11: Section 10.4.2: ", and" after "predicted rationales".
> > > > > > > > > > > > >
> > > > > > > > > > > > > A11: We have removed the unnecessary comma after “predicted rationales” to improve sentence flow.
> > > > > > > > > > > > >
> > > > > > > > > > > > > > Q12: In the same paragraph "The lower, the better;" reads very informal.
> > > > > > > > > > > > >
> > > > > > > > > > > > > A12: We have replaced the informal phrase with the more precise: “A lower DFFOT indicates a more faithful explanation.”
> > > > > > > > > > > > >
> > > > > > > > > > > > > > Q13: Section 10.5.1 might be renamed into something like "Avoiding shortcut learning".
> > > > > > > > > > > > >
> > > > > > > > > > > > > A13: We have renamed Section 10.5.1 to “Avoiding Shortcut Learning” to better reflect its content.
> > > > > > > > > > > > >
> > > > > > > > > > > > > > Q14: Section 10.5.2: Here some critique of CoT ([10]) should be mentioned alongside the Nye et al. (2021) paper.
> > > > > > > > > > > > >
> > > > > > > > > > > > > A14: We have revised Section 10.5.2 to include examples from [10], which highlight that Chain-of-Thought (CoT) reasoning can sometimes reduce performance due to insufficient faithfulness. This provides a more nuanced analysis alongside the discussion of Nye et al. (2021).
> > > > > > > > > > > > >
> > > > > > > > > > > > > > Q15: The section also missed early work on incorporating human feedback, see [42] for an overview.
> > > > > > > > > > > > >
> > > > > > > > > > > > > A15: We have added reference [42] and discussed relevant early work on incorporating human feedback in Section 10.7.2, providing a more comprehensive overview of this direction.
> > > > > > > > > > > > >
> > > > > > > > > > > > > > Q16: Section 10.7.2: "It remains unclear how this knowledge is injected during training" -> see for instance [43–45].
> > > > > > > > > > > > >
> > > > > > > > > > > > > A16: We have incorporated references [43–45] to expand the discussion on how knowledge is injected into foundation models during training.
> > > > > > > > > > > > >
> > > > > > > > > > > > > > Q17: Section 10.7.3: "Identifying the biases" -> "identifying the biases", "de-bias strategies" -> "de-biasing strategies".
> > > > > > > > > > > > >
> > > > > > > > > > > > > A17: We have corrected these typographical errors in the revised manuscript and noted that this section has now been renumbered to Section 10.7.4.
> > > > > > > > > > > > >
> > > > > > > > > > > > > > Q18: I found it interesting to maybe also end this section on a discussion on which methods can be transferred between different kinds of foundation models (specifically with respect to modality).
> > > > > > > > > > > > >
> > > > > > > > > > > > > A18: We have added a new Section 10.7.3 titled “Transferability of Explanation Across Different Modalities,” where we discuss which explanation methods can be transferred across different types of foundation models.

---

> > > > > > > > > > > > > > ### Author Response · Authors · 2025-08-14
> > > > > > > > > > > > > > **Response to Reviewer 8MQQ (14/N)**
> > > > > > > > > > > > > >
> > > > > > > > > > > > > > # Section 11
> > > > > > > > > > > > > >
> > > > > > > > > > > > > > > Q1: Section 11.1: What about partially altered content?
> > > > > > > > > > > > > >
> > > > > > > > > > > > > > A1: Thank you for raising this point. We have expanded the definition of AIGC detection correction in Section 11.1 to explicitly include partially altered content. While this addition acknowledges a spectrum of alteration levels, we clarify that our primary focus remains on binary classification for consistency with common practice in the field.
> > > > > > > > > > > > > >
> > > > > > > > > > > > > > > Q2: Section 11.2.1 The concept of "cross-perplexity" is not explained.
> > > > > > > > > > > > > >
> > > > > > > > > > > > > > A2: We have added an explanation of the term “cross-perplexity” in Section 11.2.1, describing how it is computed and its role in distinguishing AI-generated content from human-authored text.
> > > > > > > > > > > > > >
> > > > > > > > > > > > > > > Q3: In the following paragraph, the fact that AI-generated text "tends to lie in negative curvature of the model's log probability" is not clear.
> > > > > > > > > > > > > >
> > > > > > > > > > > > > > A3: We have revised the paragraph to include a formal definition of this property, providing mathematical context and clarifying how negative curvature is measured in the model’s log-probability space.
> > > > > > > > > > > > > >
> > > > > > > > > > > > > > > Q4: p.74 "By observing that AI-generated text exhibits lower intrinsic dimensionality" - in what space?
> > > > > > > > > > > > > >
> > > > > > > > > > > > > > A4: We have clarified that the observation refers to AI-generated text exhibiting lower intrinsic dimensionality compared to human-written text when measured in the representation space of fixed text embeddings (e.g., sentence- or token-level vectors produced by a pretrained encoder).
> > > > > > > > > > > > > >
> > > > > > > > > > > > > > > Q5: Section 11.2.2 should maybe also mention that detection by intuition becomes much harder as models improve.
> > > > > > > > > > > > > >
> > > > > > > > > > > > > > A5: We have added this insightful observation at the end of Section 11.2.2, noting that as generative models improve, intuitive detection by humans becomes significantly more difficult.
> > > > > > > > > > > > > >
> > > > > > > > > > > > > > > Q6: Section 11.3: "watermark payload" should use latex quotes, i.e. `` and ''.
> > > > > > > > > > > > > >
> > > > > > > > > > > > > > A6: We have corrected the quotation marks for “watermark payload” to use the proper LaTeX style.
> > > > > > > > > > > > > >
> > > > > > > > > > > > > > > Q7: Section 11.3.2 might want to add [46].
> > > > > > > > > > > > > >
> > > > > > > > > > > > > > A7: We have added reference [46] to Section 11.3.2 along with a discussion of its contributions and relevance to the topic.
> > > > > > > > > > > > > >
> > > > > > > > > > > > > > > Q8: Some of the entries 11.3.2 seem to also fit 11.4, which maybe warrants making the differences more explicit.
> > > > > > > > > > > > > >
> > > > > > > > > > > > > > A8: We have added brief clarifying statements (marked with \blue{}) at the start and end of the Learnable Watermarking subsection, as well as in the opening of the Neural Network Detectors section, to make the distinctions between these categories more explicit.
> > > > > > > > > > > > > >
> > > > > > > > > > > > > > > Q9: Second paragraph of 11.4: Missing citations for T5 and RoBERTa and Siamese network.
> > > > > > > > > > > > > >
> > > > > > > > > > > > > > A9: We have added the appropriate citations for T5, RoBERTa, and Siamese network models in the second paragraph of Section 11.4.
> > > > > > > > > > > > > >
> > > > > > > > > > > > > > > Q10: Same paragraph: Comma after "LLMDet" and "Ghostbuster". No comma after "machine generated-text, therefore".
> > > > > > > > > > > > > >
> > > > > > > > > > > > > > A10: We have corrected the punctuation by adding a comma after “LLMDet” and “Ghostbuster” and by removing the comma after “machine generated-text” in the phrase “machine generated-text therefore.”
> > > > > > > > > > > > > >
> > > > > > > > > > > > > > > Q11: Last paragraph in 11.4: "To mitigate this problem, bing"?
> > > > > > > > > > > > > >
> > > > > > > > > > > > > > A11: We have corrected this error by properly presenting the cited paper and its details, ensuring the sentence reads clearly and accurately.
> > > > > > > > > > > > > >
> > > > > > > > > > > > > > > Q12: Section 11.5.3: "Especially, The scenarios" -> "Especially, the scenarios".
> > > > > > > > > > > > > >
> > > > > > > > > > > > > > A12: We have corrected the capitalization so that the phrase now reads “Especially, the scenarios” in the revised manuscript.

---

> > > > > > > > > > > > > > > ### Author Response · Authors · 2025-08-14
> > > > > > > > > > > > > > > **Response to Reviewer 8MQQ (15/N)**
> > > > > > > > > > > > > > >
> > > > > > > > > > > > > > > # Section 12
> > > > > > > > > > > > > > >
> > > > > > > > > > > > > > > > Q1: Section 12.1: "Tao et al. (2021); Ma et al. (2024a)" should be \citep. Same for "Zhang et al. (2023j); Cho et al., (2023); and Luccioni et al. (2023)" in section 12.2.
> > > > > > > > > > > > > > >
> > > > > > > > > > > > > > > A1: Thank you for the careful review. We have corrected these citations to use the `\citep` format in both Section 12.1 and Section 12.2 to ensure consistent formatting.
> > > > > > > > > > > > > > >
> > > > > > > > > > > > > > > > Q2: Section 12.3: \citealp for Liu et al. (2024g) and following citations.
> > > > > > > > > > > > > > >
> > > > > > > > > > > > > > > A2: We have updated these citations to use the `\citealp` format in Section 12.3, following the suggested style.
> > > > > > > > > > > > > > >
> > > > > > > > > > > > > > > > Q3: Same paragraph: \citep for Edemacu & Wu (2024) and Xiao et al. (2023a).
> > > > > > > > > > > > > > >
> > > > > > > > > > > > > > > A3: We have corrected these citations to use the `\citep` format in Section 12.3 as recommended.
> > > > > > > > > > > > > > >
> > > > > > > > > > > > > > > > Q4: Section 12.5: "opposite outcomes Future research directions" -> "opposite outcomes, future research directions".
> > > > > > > > > > > > > > >
> > > > > > > > > > > > > > > A4: We have revised this sentence by adding the missing comma, so it now reads “opposite outcomes, future research directions.” This corrects the typographical error and improves readability.
> > > > > > > > > > > > > > >
> > > > > > > > > > > > > > > > Q5: "include addressing uncertainty not capteured in the model's interal state" - this point is not very clear to me.
> > > > > > > > > > > > > > >
> > > > > > > > > > > > > > > A5: We have replaced “captured” with “reflected” for accuracy and added a clarification: “the ‘unknown unknowns’ where the model is unaware of its own ignorance, often when faced with out-of-distribution data or novel concepts, and enhancing models’ capacity to resolve uncertainty effectively.”
> > > > > > > > > > > > > > >
> > > > > > > > > > > > > > > > Q6: "The risk of hallucination also increases as the model makes inferences based on its prior knowledge, which may not be applicable in the new context" - missing reference?
> > > > > > > > > > > > > > >
> > > > > > > > > > > > > > > A6: We have added relevant references to support this statement, ensuring it is properly grounded in prior literature.

---

> > > > > > > > > > > > > > > > ### Comment · Reviewer_8MQQ · 2025-08-18
> > > > > > > > > > > > > > > > **Response to Rebuttal**
> > > > > > > > > > > > > > > >
> > > > > > > > > > > > > > > > I think the authors for engaging with and integrating lots of my feedback. I think it has overall improved the quality of the work a lot.

---

> > > > > > > > > > > > > > > > > ### Author Response · Authors · 2025-08-20
> > > > > > > > > > > > > > > > >
> > > > > > > > > > > > > > > > > Thank you for the thoughtful engagement throughout the review process and the encouraging confirmation. Your detailed feedback significantly improves the paper, and we are glad to know that the revisions are recognized. It is our honour to have this high-quality review. We sincerely appreciate your valuable efforts!

---

### Review · Reviewer_4vUV · 2025-06-10

**Summary Of Contributions:**

This survey offers a exploration of reliable and responsible foundation models, constructing a unified framework that categorizes these models into four types—Large Language Models (LLMs), Large Multimodal Models (LMMs), Text-to-Image (T2I) models, and Video Generative Models and systematically examines nine critical dimensions: bias and fairness, alignment, security, privacy, hallucination, uncertainty, distribution shift, explainability, and AI-Generated Content (AIGC) detection. It delves into the challenges within each dimension and proposes a range of solutions. The work also identifies research gaps, such as the lack of lightweight out-of-distribution detection methods, insufficient multimodal explainability, and the need for more robust watermarking, and outlines future directions, including the development of lifelong learning mechanisms, improved fairness in detection systems, and integrated approaches to ensure model reliability and responsibility across technical and societal dimensions.

**Audience:**

Yes

**Broader Impact Concerns:**

This paper doesn't need an additinal Broader Impact Statement.

**Claims And Evidence:**

Yes

**Requested Changes:**

See Weaknesses.

**Strengths And Weaknesses:**

Strengths:
1. This paper covers four model types (LLMs, LMMs, T2I, video) and nine key areas like bias, security, and privacy, integrating technical analysis with real-world applications.
2. This paper focus more on multimodal challenges: addresses unique issues in LMMs (e.g., visual hallucination) and T2I models (e.g., gender bias), reflecting the evolving multimodal AI landscape.
3. This survey highlights technical trade-offs (e.g., privacy vs. performance) and cross-domain links (e.g., security measures worsening bias), guiding integrated solutions for responsible AI.

Weaknesses:
1. The overview in Figure 1 is confusing. For example, models and tasks are not in the same dimensions and should not be put together.

2. The Large Multimodal model's abbreviation (LMM) is strange. Usually, we refer to this as a Multimodal Large Language Model (MLLM) instead of LMM, and it may need to be revised.

3. Adversarial Attacks of Security in Text-to-Image Models ignores some important works, such as [1][2].

4. This paper doesn't discuss the advantages and differences with other latest surveys [3][4] related to the security of foundation models.

5. It is unclear how adversarial training improves OOD detection.

[1] Content-based Unrestricted Adversarial Attack, NeurIPS 2023

[2] Diffusion Models for Imperceptible and Transferable Adversarial Attack, TPAMI 2024

[3] Safety at Scale: A Comprehensive Survey of Large Model Safety, SuperIntelligence-Robotics-Safety & Alignment 2025

[4] Benchmarking Trustworthiness of Multimodal Large Language Models: A Comprehensive Study, Arxiv 2024

---

> ### Author Response · Authors · 2025-08-08
> **Rebuttal to Reviewer 4vUV**
>
> We sincerely thank Reviewer 4vUV for the insightful and valuable feedback. Your comments have been instrumental in improving the quality of our work. In the revised manuscript (with changes highlighted in blue), we have carefully addressed all of your concerns. We hope these clarifications will address your concerns.
>
> ---
> > Q1: The overview in Figure 1 is confusing. For example, models and tasks are not in the same dimensions and should not be put together.
>
> A1: Thank you very much for the helpful suggestion. We have revised Figure 1 by removing the model part to avoid mixing elements from different dimensions. The updated figure now focuses solely on the nine core tasks/topics covered in our work. To maintain clarity, we have moved the model categorization into the caption. We believe this redesign more clearly separates models from tasks and improves the overall readability of the figure.
>
> ---
> >Q2: The Large Multimodal model's abbreviation (LMM) is strange. Usually, we refer to this as a Multimodal Large Language Model (MLLM) instead of LMM, and it may need to be revised.
>
> A2: Thank you for pointing this out. Following your suggestion, we reviewed recent multimodal literature and found that Multimodal Large Language Model (MLLM) has indeed become the standard term, rather than LMM. This terminology not only aligns better with current conventions but also more clearly emphasizes that these models are built upon LLMs, distinguishing them from other multimodal systems such as text-to-image or video generative models. Accordingly, we have replaced all instances of LMM with MLLM in the revised version.
>
> ---
> >Q3: Adversarial Attacks of Security in Text-to-Image Models ignores some important works, such as [1][2].
>
> A3: Thank you for the suggestion. We have incorporated both [1] and [2] into the section on Adversarial Attacks in Text-to-image models. We believe this addition helps make our coverage of the topic more comprehensive.
>
> ---
> >Q4: This paper doesn't discuss the advantages and differences with other latest surveys [3][4] related to the security of foundation models.
>
> A4: Thank you for the valuable feedback. We have included the latest surveys [3] and [4] in the updated version of our paper. Additionally, we have added a discussion of the differences and advantages of our work compared to these prior studies in the opening paragraph of the Security section.
>
> ---
> >Q5: It is unclear how adversarial training improves OOD detection.
>
> A5: Thank you for the feedback. We would like to clarify that we only mention the role of adversarial training in improving OOD generalization, not OOD detection. Additionally, we have added a dedicated paragraph in Section 9.3.2 that explains the underlying mechanism of adversarial training and discusses how it contributes to better generalization under distribution shifts.

---

### Review · Reviewer_qQuq · 2025-06-23

**Summary Of Contributions:**

This submission is a comprehensive survey for reliability and responsibility issues of foundation models (LLMs, LMMs, and generative diffusion models).  The coverage of literature is one of the strongest contributions of this work, which make exhaustive survey for a various aspects of reliability and responsibility, including bias and fairness, alignment, security, privacy, hallucination, uncertainty, distribution shift, explainability, and AGIC detection.

**Audience:**

Yes

**Broader Impact Concerns:**

There is no critical issue in this submission.

**Claims And Evidence:**

Yes

**Requested Changes:**

Please refer to the section of 'Strengths And Weaknesses'.

**Strengths And Weaknesses:**

<Weaknesses>
1. The title needs to include the term "Survey".

2. In each section, the list of literature should be better visually summarized; for example, citation trees, which are recently frequently used in various survey papers.

3. Many survey papers already exist for some topics of this paper.
For example, for the hallucination of LLMs, the following survey papers already exist.
- L. Huang et al., A Survey on Hallucination in Large Language Models: Principles, Taxonomy, Challenges, and Open Questions, ACM TOIS 2024.
- P. Sahoo et al., A Comprehensive Survey of Hallucination in Large Language, Image, Video and Audio Foundation Models, EMNLP findings 2024.
- Therefore, the novelty of this survey should be better justified by comparing with existing survey papers.
- Hallucination is only a single example. For each of the other topics, several survey papers can be easily found.

4. Moreover, it should be also verified why this kind of (somewhat too broad) survey paper is needed instead of these more-focused surveys for each of specific topics.

---

> ### Author Response · Authors · 2025-07-30
> **Rebuttal to Reviewer qQuq**
>
> We would like to sincerely thank the reviewer for the insightful and valuable comments. We have found the feedback to be extremely helpful in improving the quality of our paper. We have thoroughly revised the manuscript accordingly, with all major changes highlighted in blue text in the updated version. We sincerely apologize for submitting our rebuttal a little late.
>
> ---
>
> **Q1**: In each section, the list of literature should be better visually summarized; for example, citation trees, which are recently frequently used in various survey papers.
>
> **A1**: Thank you for your valuable suggestion. We completely agree that visual summaries like citation trees are crucial for enhancing the quality and impact of our survey. In our current version, we have integrated detailed citation trees for several key areas of our survey, specifically in Figure 4, Figure 9, Figure 11, Figure 16, and Figure 21. We believe these additions already substantially improve the paper's clarity and provide clear roadmaps for these topics. Due to the significant time required to design and create high-quality graphics for each section, we have not yet been able to implement this suggestion in the current revision for other sections. However, we want to assure the reviewer that we are fully committed to adding these citation trees for all key sections in the final camera-ready version of the paper, should it be accepted.
>
> ---
>
> **Q2**: Reference to existing survey papers
>
> **A2**: We appreciate the reviewer's guidance on this critical point, and we agree that a more explicit comparison with existing surveys is essential. In response, we have performed a comprehensive review of the latest survey papers in each topic and have integrated these citations into our work. More significantly, we have substantially revised each section to incorporate discussions on these papers (e.g., the introduction of Section 7). When citing these surveys, we now explicitly articulate our unique contribution. We believe these revisions now clearly establish the novelty and necessity of our work within the broader landscape of existing literature.
>
> ---
>
> **Q3**: Why is this kind of (somewhat too broad) survey paper needed instead of these more-focused surveys for each of the specific topics?
>
> **A3**: Thank you for this critical question, which addresses the core motivation of our paper. Our central thesis is that the key topics in reliable and responsible foundation models are deeply interconnected. We argue that treating them in isolation, as is common in other surveys, overlooks the significant emergent challenges that arise from their intersection. Therefore, a holistic, broad-scope survey is necessary to build bridges between these topics and guide research toward the development of truly responsible models. Our paper's structure is designed to prove this progressively. First, within each chapter's 'Limitations and Future Directions' section, we already begin to discuss the challenges of combining that topic with others. Then, the final 'Intersection and Conclusion' section provides a holistic, macro-level analysis of various possible combinations and their systemic implications. This design demonstrates that a broad-scope approach is essential, as the theme of intersection is woven into the very fabric of our analysis.

---

> > ### Comment · Action_Editor_ZZAe · 2025-08-05
> > **Rebutal incomlete**
> >
> > Dear authors,
> > Your rebutal is very incomplete. You have not reacted to the reviews of the other reviewers. If you still intend to do so, please respond and revise tge manuscript accordingly by Friday 8.8. We shall close the review process thereafter.
> > Best
> > Your AE

---

> > > ### Author Response · Authors · 2025-08-08
> > > **Response to AE and Reviewers**
> > >
> > > Dear AE and Reviewers,
> > >
> > > Thank you very much for your reminder and for the opportunity to improve our submission.
> > >
> > > We sincerely apologize for the delay in addressing the remaining reviewer comments. We truly appreciate all the feedback and are currently finalizing our responses and corresponding manuscript revisions. We will make sure to submit the updated rebuttal and revised manuscript by Friday, August 8 as requested.
> > >
> > > Thank you again for your patience and support.

---

### Comment · Reviewer_r6eT · 2025-05-18
**Review of "Reliable and Responsible Foundation Models"**

This is an impressively comprehensive survey paper on reliable and responsible foundation models. The authors have done an excellent job covering nine critical aspects (bias and fairness, alignment, security, privacy, hallucination, uncertainty, distribution shift, explainability, and AIGC detection) across multiple types of foundation models. I particularly appreciated the thoughtful organization and visual aids that help clarify complex concepts. The paper's strength lies in its systematic approach and the concluding discussion on intersections between different research areas.

While already strong, the paper could be improved in a few specific areas. The discussion on privacy techniques (Section 6) would benefit from more concrete examples of how these methods impact model performance in real deployments. The treatment of T2I and video models feels somewhat thinner than LLMs in several sections, particularly in explainability (Section 10). I was also curious about how the authors view the comparative urgency of these nine aspects in high-stakes domains - some prioritization guidance would be valuable for researchers. Finally, Section 9 on distribution shift could better compare the computational efficiency tradeoffs between different adaptation methods, which is crucial information for practical implementations.

---

> ### Author Response · Authors · 2025-06-10
> **Double-check the Section of the Review**
>
> Dear Reviewer r6eT,
>
> Thank you for taking the time to provide your review. I saw your comments in the official comment section and was wondering if you might have posted them there by accident, rather than through the review section.
>
> Thank you again for your time and effort.

---

### Decision · Action_Editor_ZZAe · 2025-08-24

**Recommendation:** Accept with minor revision

**Additional Comments:**

- In their response to reviewer qQuq authors acknowledge the benefit of the better visual summary of surveyed papers. They, however, postponed this to later revisions of the paper due to limited time within the rebuttal discussion. The low clarity of overview and organization
of the surveyed papers has been raised again as an issue in the final recommendations. Please improve the paper in this respect to help reader understand the scope of the review.
- The immense scope of the review is both a strength and weakness of the paper. To help the reader, please expand the discussion of existing surveys and contrast your paper to these in more detail. For example, an overview table summarizing the coverage of other surveys and comparing to yours would be of great help.
- Given the speed of developments in the area, a survey paper such as this risks becoming rapidly outdated. The authors should clearly state the time-relevance of their survey somewhere early in the manuscript to clarify the coverage.
- For the same reasons I recommend the authors to proceed with the above updates as fast as possible to allow for quick accept and release of the paper.

**Audience:**

Yes

**Audience Explanation:**

The paper is a comprehensive survey of foundation models focusing in their reliability and responsibility.
These are critical properties of the models for both research and practical deployment.
It covers a wide range of aspects - bias and fairness, security, privacy, uncertainty, alignment with human objectives, distribution shifts - for a broad set of models - large language, large multi-modal, text-to-image and video generative models.
In its scope it merges views for many relevant areas providing a unique crosscutting overview of the relevant topics.

**Claims And Evidence:**

Yes

**Claims Explanation:**

This is a survey paper where the claims are supported by the original papers cited within the survey.

---

> ### Author Response · Authors · 2025-09-25
> **Revision**
>
> Dear AE ZZAe,
>
> Thank you very much for your positive evaluation and the constructive feedback on our manuscript.
>
> We have addressed all of the points raised in our revision. Specifically, we have:
>
> 1: Added a visual summary of the surveyed papers for all sections.
>
> 2: Included a comparison table of existing surveys with necessary discussions (page 86).
>
> 3: Clarified the timeliness and relevance of our work in the introduction (page 4).
>
> The revised version incorporating these changes was submitted on September 24th.
>
> Thank you again for your guidance.
>
> Sincerely,
> The Authors

---

> > ### Comment · Action_Editor_ZZAe · 2025-09-28
> > **Please correct camera ready**
> >
> > Dear authors,
> >
> > in your latest pdf version of the paper the formatting of the list of authors is corrupted. Please correct it so that the paper can be released.
> >
> > Thanks
> > your AE

---

> > > ### Author Response · Authors · 2025-10-04
> > > **author list format revised**
> > >
> > > Dear AE,
> > >
> > > Many thanks for your notifications on this.
> > >
> > > We have uploaded a new version. Since we have many authors, we are following the format from some previous TMLR publications (like https://openreview.net/forum?id=NeS9Kj2JwF).
> > >
> > > Thank you once again for your guidance and help.
> > >
> > > Sincerely, The Authors